# Automatic Unsupervised Ensemble Outlier Model Selection

Hong-Phuc Phan [* 1]  Tuan-Anh Vu [* 2]  Tung Kieu [* 3]  Son Ha Xuan [4]  Bin Yang [5]  Christian S. Jensen [3]

## Abstract

Unsupervised outlier detection is attractive because it eliminates the need for labeled data. Moreover, forming multi-model ensembles can improve detection robustness. However, composing an ensemble without labeled data is challenging. Naively composed ensembles can suffer from ensemble saturation, where redundant or unreliable detection models degrade performance and incur unnecessary computation. We propose `MetaEns`, an automatic unsupervised framework for selecting ensembles of outlier detection models. Using labeled meta-datasets, `MetaEns` learns a model that predicts marginal ensemble gains, estimating the expected improvement from adding a candidate model to a partially constructed ensemble. At test time, this learned signal is combined with a submodular-inspired proxy objective that enforces diminishing returns through diversity-aware discounting and family-level risk regularization, thereby enabling greedy sequential selection with adaptive early stopping. As a result, `MetaEns` constructs compact, high-quality ensembles without access to ground-truth labels. Experiments on 39 real-world datasets show that `MetaEns` consistently outperforms state-of-the-art unsupervised selectors and ensemble baselines, achieving higher average precision while using fewer models.

## 1. Introduction

Outlier detection plays a critical role in applications like fraud detection, network security, medical diagnosis, and system monitoring (Chandola et al., 2009; Ruff et al., 2021). However, in many real-world settings, ground-truth labels are unavailable, and outlier detection must be performed in a fully unsupervised manner (Zimek et al., 2013). Further, contamination rates are often unknown, and data distributions vary widely across tasks (Han et al., 2022; Zhang et al., 2026). These factors make it difficult to detect anomalies and compare detection models (Marques et al., 2020).

Existing studies have proposed diverse unsupervised outlier detectors based on density estimation (Breunig et al., 2000), isolation mechanisms (Liu et al., 2008), reconstruction errors (Zong et al., 2018), and deep representations (Ruff et al., 2018). Despite their success in specific settings, no single detector performs reliably across diverse datasets. This observation has motivated the use of ensemble methods, which aim to improve robustness by aggregating multiple detectors (Campos et al., 2021; Kieu et al., 2019). In supervised learning, ensembles can be trained and validated using labeled data. In contrast, constructing effective ensembles for unsupervised outlier detection remains an open challenge (Zimek et al., 2013).

Selecting models without labels is a key challenge (Marques et al., 2020). Without ground-truth feedback, it is unclear which detectors are reliable on a given dataset or whether adding them will improve ensemble performance (Rayana et al., 2016; Aggarwal & Sathe, 2015). Consequently, many unsupervised methods rely on fixed aggregation strategies, such as averaging all detector scores or selecting a fixed number of top-ranked models (Zhao et al., 2019a). These strategies suffer from ensemble saturation: adding more detectors beyond a small size yields diminishing or negative returns due to redundancy, conflicting rankings, or poor models. Moreover, fixed-size ensembles are inflexible and cannot adapt to dataset-specific complexity.

Recent unsupervised model selection methods attempt to address these challenges by leveraging meta-learning across labeled auxiliary datasets (Hospedales et al., 2022). Notably, frameworks such as `MetaOD` (Zhao et al., 2021) and `ELECT` (Zhao et al., 2022) learn to recommend a single detector for a new unlabeled task based on task similarity or

---

[*]Equal contribution  [1]Department of Software Engineering, FPT University, Vietnam [2]Department of Information Technology, Can Tho University of Technology, Vietnam [3]Department of Computer Science, Aalborg University, Denmark [4]School of Business, RMIT University, Vietnam [5]School of Data Science and Engineering, East China Normal University, China. Correspondence to: Tung Kieu <tungkvt@cs.aau.dk>, Son Ha Xuan <Ha.Son@rmit.edu.vn>.

*Proceedings of the 43$^{rd}$ International Conference on Machine Learning*, Seoul, South Korea. PMLR 306, 2026. Copyright 2026 by the author(s).

historical performance patterns. While effective, these methods are limited to singleton selection and do not address the more general problem of adaptive ensemble construction, where multiple complementary detectors may be required to capture diverse outlier patterns (Cheng et al., 2020).

We address this gap by formulating unsupervised ensemble outlier model selection as a sequential decision problem. Our key insight is that although the true marginal benefit of adding a detector to an ensemble is unobservable at test time, its structure can be learned offline from labeled meta-datasets. Building on this idea, we propose `MetaEns`, a framework that learns to predict the marginal ensemble gain of candidate detectors conditioned on the ensemble state. At inference time, `MetaEns` greedily constructs an ensemble by maximizing a submodular-inspired proxy objective (Nemhauser et al., 1978) that integrates the predicted gain with mechanisms for diversity control and risk mitigation.

Specifically, `MetaEns` introduces two principles that are crucial for unsupervised ensemble construction. First, we enforce diminishing returns through similarity-based discounting that penalizes candidates that introduce redundancy with already selected detectors (Kulesza & Taskar, 2012). Second, we incorporate family-risk regularization, which discourages selecting multiple detectors from algorithmic families with a history of poor or unstable performance. Together, these components yield a proxy objective that favors compact, diverse ensembles and naturally supports adaptive early stopping when no further improvement is expected.

We evaluate `MetaEns` on a benchmark of 39 real-world anomaly detection datasets (Han et al., 2022) using a large pool of 297 candidate detectors spanning multiple algorithmic families. Extensive experiments show that `MetaEns` consistently outperforms strong unsupervised baselines and recent meta-learning approaches. Notably, `MetaEns` achieves higher detection accuracy while selecting fewer models than fixed-size ensembles, and it exhibits strong resilience by recovering performance even when an initially selected detector performs poorly.

In summary, our contributions are threefold:

- We formulate the problem of unsupervised ensemble outlier model selection and cast it as a sequential decision process without access to labels.

- We propose `MetaEns`, a meta-learning framework that predicts marginal ensemble gains and combines them with a diversity- and risk-aware proxy objective for adaptive ensemble construction with early stopping.

- We provide experimental results across 39 datasets, supported by ablation studies, showing that compact, adaptively sized ensembles can outperform larger fixed ensembles in fully unsupervised settings.

## 2. Preliminaries

**Definition 2.1** (Dataset). A dataset $\mathbf{X}$ is a finite collection of data instances (or data points) $\mathbf{X} = \{\mathbf{x}_1, \mathbf{x}_2, \ldots, \mathbf{x}_N\}$, where each instance $\mathbf{x}_i \in \mathbb{R}^d$ is a $d$-vector. We denote $|\mathbf{X}| = N$ as the cardinality of the dataset.

**Definition 2.2** (Unsupervised Outlier Detection). Given a dataset $\mathbf{X} \in \mathbb{R}^{N \times d}$, an unsupervised outlier detection model, or detector, learns a scoring function $f : \mathbb{R}^{N \times d} \rightarrow \mathbb{R}^N$ that assigns an outlier score $o_i = f(\mathbf{x}_i)$ to each instance $\mathbf{x}_i \in \mathbf{X}$, where larger values indicate a higher likelihood of being an anomaly. A decision function can be derived by thresholding the scores at a user-defined level $\tau$, yielding outlier labels $\hat{y}_i = \mathbb{I}(o_i > \tau)$. The resulting outlier set is defined as follows:

$$\mathbf{O} = \{\mathbf{x}_i \in \mathbf{X} \mid \hat{y}_i = 1\}$$

**Definition 2.3** (Unsupervised Outlier Model Selection). Let $\Omega = \{f_1, f_2, \ldots, f_K\}$ be the set of $K$ candidate outlier detection models. Each $f_i \in \Omega$ can be seen as a $(\texttt{detector}, \texttt{configuration})$ tuple, where the `configuration` denotes a set of hyperparameters of the `detector`. Let $\mathbf{X} = \{\mathbf{x}_1, \mathbf{x}_2, \ldots, \mathbf{x}_N\}$ be an unlabeled dataset. Each outlier detection model $f_i$ acts as an outlier scoring function $f_i : \mathbb{R}^{N \times d} \mapsto \mathbb{R}^N$, which assigns an outlier score to each instance in $\mathbf{X}$. The primary task of unsupervised outlier model selection is then to choose the optimal model $f^*$ as follows:

$$f^* = \arg\max_{f_i \in \Omega} \Gamma(f_i, \mathbf{X})$$

Here, $\Gamma(\cdot)$ is an unsupervised evaluation criterion that estimates the quality of model $f_i$ based solely on the distributional properties of its output scores without using labeled outlier or normal instances.

**Problem Definition: Unsupervised Ensemble Outlier Model Selection.** Let $\Omega = \{f_1, f_2, \ldots, f_K\}$ be a set of $K$ candidate outlier detection models. Each model $f_i \in \Omega$ can be seen as a $(\texttt{detector}, \texttt{configuration})$ tuple, where the `configuration` denotes a specific set of hyperparameters of the `detector`. Let $\mathbf{X} = \{\mathbf{x}_1, \mathbf{x}_2, \ldots, \mathbf{x}_N\}$ be an unlabeled dataset. Each outlier detection model $f_i \in \Omega$ acts as an outlier scoring function $f_i : \mathbb{R}^{N \times d} \mapsto \mathbb{R}^N$ that assigns an outlier score to each instance in $\mathbf{X}$. A candidate ensemble $P \subseteq \Omega$ is evaluated as a whole, not as an independent sum of member-model criteria. Specifically, we first aggregate member scores into a single ensemble score vector,

$$\mathbf{o}_P = \frac{1}{|P|} \sum_{f \in P} f(\mathbf{X}).$$

The task of unsupervised ensemble outlier model selection is to choose a set of models $P \subseteq \Omega$ as follows:

$$P^* = \arg\max_{P \subseteq \Omega} \psi(P; \mathbf{X}) \quad \text{s.t.} \quad |P| \leq \eta.$$

Here, the criterion $\psi(P; \mathbf{X})$ estimates the quality of the aggregated ensemble score vector $\mathbf{o}_P$ without using any ground-truth labels, and $\eta$ serves as a budget on the maximum allowable ensemble size.

**Definition 2.4** (Meta-datasets). We assume access to a collection of labeled meta-datasets $\mathcal{M} = \{(\mathbf{M}_1, \mathbf{y}_1), (\mathbf{M}_2, \mathbf{y}_2), \ldots, (\mathbf{M}_L, \mathbf{y}_L)\}$ where each $\mathbf{M}_i \in \mathbb{R}^{N_{M_i} \times d_{M_i}}$ is a dataset and $\mathbf{y}_i \in \{0, 1\}^{N_{M_i}}$ provides ground-truth anomaly labels. These meta-datasets are used exclusively for evaluating and selecting candidate outlier detection models. At test time, we are given an unlabeled dataset $\mathbf{X} = \{\mathbf{x}_1, \mathbf{x}_2, \ldots, \mathbf{x}_N\}$ that satisfies $\mathbf{X} \sim \mathbb{P}_{\mathbf{X}}$ and $\mathbb{P}_{\mathbf{X}} \neq \mathbb{P}_{\mathbf{M}_i}, \forall i$, i.e., the test distribution differs from the distributions underlying the meta-datasets.

## 3. Methodology

### 3.1. Framework Overview

`MetaEns` operates in two phases summarized in Fig. 1. **Offline** (Sec. 3.2): we run oracle-greedy rollouts on labeled meta-datasets to supervise a two-part predictor of the marginal gain of adding a candidate detector to a partial ensemble, and we estimate family-level risk priors from these trajectories. **Online** (Sec. 3.3): we initialize from an unsupervised primary detector and greedily expand the ensemble via a submodular-inspired proxy combining the predicted gain with redundancy and family-risk controls, terminating adaptively once no candidate offers positive utility.

### 3.2. Offline Meta-training

**Algorithmic families.** During offline meta-training, we do not merely evaluate individual models, but also track the historical behavior of their underlying algorithmic paradigms. Each detector $f \in \Omega$ belongs to exactly one of $|\mathcal{F}| = 8$ algorithmic families, $\mathcal{F} = \{\text{IForest}, \text{LOF}, \text{kNN}, \text{HBOS}, \text{OCSVM}, \text{LODA}, \text{ABOD}, \text{COF}\}$, where detectors within the same family share the underlying algorithm and differ only in hyperparameters (e.g., all `LOF` variants differing only in $k$ or distance metric). This mapping is fixed from metadata, denoted $\text{fam} : \Omega \to \mathcal{F}$, and lets us aggregate trajectory statistics at the family level to form risk priors used during online selection (sensitivity to family granularity in App. A.1).

Let $\mathcal{M} = \{(\mathbf{M}_\ell, \mathbf{y}_\ell)\}_{\ell=1}^L$ be the labeled meta-datasets, and let $\Omega = \{f_1, \ldots, f_K\}$ denote the candidate detector pool.

For a dataset $(\mathbf{M}, \mathbf{y}) \in \mathcal{M}$ and an ensemble (model set) $P \subseteq \Omega$, we define the ensemble score vector as the mean of the member scores:

$$\mathbf{o}_P = \frac{1}{|P|} \sum_{f \in P} f(\mathbf{M}), \qquad (1)$$

where $f(\mathbf{M}) \in \mathbb{R}^{|\mathbf{M}|}$ is the outlier-score vector produced by $f$ on $\mathbf{M}$. We default to mean aggregation to preserve continuous score information; median, max, and min variants are compared in App. A.2.

**Oracle marginal gain.** We define the true marginal gain of adding $f_i \in \Omega \setminus P$ to $P$ as the improvement in Average Precision:

$$G(f_i \mid P) = \text{AP}(\mathbf{o}_{P \cup \{f_i\}}, \mathbf{y}) - \text{AP}(\mathbf{o}_P, \mathbf{y}), \quad (2)$$

where $\text{AP}(\cdot, \cdot)$ is computed from the ensemble score vector and the ground-truth label vector. We use AP as it is threshold-independent and robust under severe class imbalance, which is common in outlier detection.

**Generating training trajectories.** To obtain informative supervision, we construct meta-training trajectories using an oracle greedy policy. For each meta-dataset, we initialize the ensemble with a primary model $f_1^*$ chosen to maximize $\text{AP}(f(\mathbf{M}), \mathbf{y})$. We then iteratively add models that maximize the true gain in Eq. 2. This strategy exposes the meta-model to high-quality partial ensembles, avoiding training dominated by arbitrary or low-signal states. We empirically verify that alternative strategies such as $\varepsilon$-greedy exploration consistently underperform oracle greedy rollouts, and that the full meta-training corpus is required to saturate this gain (App. A.3). At each step $i$, for every candidate $f \in \Omega \setminus P$, we compute a state representation $\phi(f, f_{i-1}^*, P)$ and its oracle gain $G(f \mid P)$, obtaining supervised pairs $\{(\phi(\cdot), G(\cdot))\}$ across multiple ensemble sizes and datasets to learn a predictor of marginal gains.

**State representation.** The state $\phi(f_i, f_{i-1}^*, P)$ encodes three factors: candidate-recent redundancy, candidate-ensemble compatibility, and the selection stage $|P|$. We deliberately restrict `MetaEns` to score-level summaries rather than raw inputs or detector internals. This score-only design provides three core advantages: (i) *Dimensionality independence*: A fixed-dimensional state derived from 1D scores applies uniformly across heterogeneous datasets of varying feature sizes (5–1,555); (ii) *Empirical grounding*: Distributional score properties (entropy, tail behavior) correlate reliably with true detection performance (Zhao et al., 2022); (iii) *Modality-agnosticism*: Operating solely on scores enables zero-shot transfer to image and text anomaly detection without retraining (Sec. 4.5).

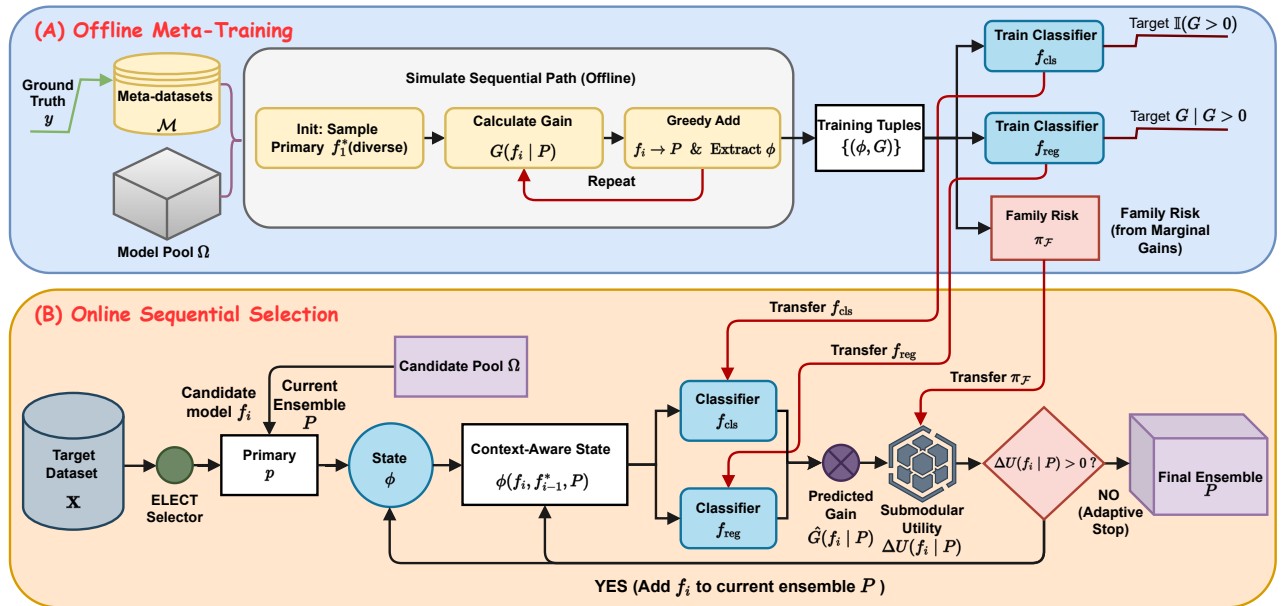

*Figure 1.* Overview of `MetaEns`. **(A) Offline:** Oracle-greedy rollouts on meta-datasets supervise a two-part gain predictor ($f_{\text{cls}}$, $f_{\text{reg}}$) and produce family-risk priors $\{\pi_F\}_{F \in \mathcal{F}}$. **(B) Online:** `MetaEns` greedily expands from a primary detector by maximizing a proxy combining predicted gain $\hat{G}$, redundancy discount $\gamma$, and risk penalty $\pi_{\text{fam}(\cdot)}$, stopping when no candidate has positive utility.

We extract a base feature vector $\phi_{\cdot,\cdot} \in \mathbb{R}^{d_{\text{base}}}$ from paired Min–Max normalized score vectors ($\mathbf{o} \in [0, 1]^N$). This extractor captures correlations (Spearman, cosine), distributional statistics (entropy, kurtosis), and top-ranked overlap (Jaccard); see App. A.4. Concretely, we compute pairwise features for $(f_i, f_{i-1}^*)$, $(f_i, P)$, and $(f_{i-1}^*, P)$ to form the final state:

$$\phi(f_i, f_{i-1}^*, P) = (\phi_{f_i, f_{i-1}^*}, \phi_{f_i, P}, \phi_{f_{i-1}^*, P}, |P|). \quad (3)$$

**Marginal gain modeling.** Positive marginal gains become sparse as the ensemble grows: most additional candidates have $G(f_i \mid P) \leq 0$, making the gain distribution zero-inflated. A single regressor on this distribution tends to predict small positive values to minimize squared error, triggering spurious additions. We therefore use a two-part hurdle-style model that separates whether a candidate improves the ensemble from how much:

$$\hat{G}(f_i \mid P) = f_{\text{cls}}(f_i \mid P) \cdot f_{\text{reg}}(f_i \mid P), \quad (4)$$

where $f_{\text{cls}}(f_i \mid P) = \mathbb{P}(G(f_i \mid P) > 0)$ predicts improvement probability and $f_{\text{reg}}(f_i \mid P) = \mathbb{E}[G(f_i \mid P) \mid G(f_i \mid P) > 0]$ predicts the positive gain magnitude. This decomposition lets the classifier gate sparse positive gains, while the regressor focuses on useful additions.

For each training pair $(\phi, G)$, $y_{\text{cls}} = \mathbb{I}(G > 0)$ and $y_{\text{reg}} = G$ is fit only when $G > 0$. Intuitively, $f_{\text{cls}}$ models membership of the positive-gain set while $f_{\text{reg}}$ models its magnitude. This prevents zero-inflated true gains from biasing a single

regressor into outputting spurious positive values. Table 2 validates this: a single predictor drops AP from $0.4308$ to $0.4133$ and worsens rank to $87$.

**Choice of meta-model.** We instantiate $f_{\text{cls}}$ and $f_{\text{reg}}$ with `ExtraTrees` (Geurts et al., 2006), preventing overfitting on dense, nonlinear state features. Compared to Random Forests (Breiman, 2001), its randomized thresholds provide stronger regularization. Unlike boosting (`XGBoost`/`LightGBM`), its bagging better supports zero-shot transfer without memorizing meta-training residuals. Empirically (App. A.5), `ExtraTrees` yields the best AP ($0.4308 \pm 0.0064$) and rank ($59.3 \pm 6.96$) among six families, outperforming alternatives while remaining efficient.

### 3.3. Online Model Selection

At test time, $G(\cdot)$ is unavailable. We initialize with a primary detector $f_1^*$ via an unsupervised selector $\mathcal{S}$ (defaulting to `ELECT` (Zhao et al., 2022)). Although this differs from the oracle-AP primary used in meta-training, $\phi$ depends only on score statistics, so $\hat{G}$ transfers robustly to suboptimal primary selectors (Sec. 4.6). We then expand via a proxy combining $\hat{G}$ with redundancy and risk controls, inducing submodular-like diminishing returns without requiring $\hat{G}$ to satisfy strict submodularity. Enforcing such a rigid constraint directly on the black-box predictor would unnecessarily compromise its estimation accuracy. Instead, our proxy relies on explicit penalty terms that naturally scale with the ensemble size.

**Proxy marginal utility.** Given a current ensemble $P$ (with the last selected model being $f_{i-1}^*$), the marginal utility of adding candidate $f_i$ is:

$$\Delta U(f_i \mid P) = \gamma(f_i, P) \cdot \left( \hat{G}(f_i \mid P) - \lambda_{\text{fam}} \pi_{\text{fam}(f_i)} \right), \quad (5)$$

where $\gamma(f_i, P) \in (0, 1]$ discounts redundant candidates, and $\lambda_{\text{fam}} \pi_{\text{fam}(f_i)} \geq 0$ penalizes historically risky families. The hyperparameters $\beta$ and $\lambda_{\text{fam}}$ control redundancy and risk, tuned solely on meta-training data (App. A.6). Note that $\beta$ is insensitive across the entire search range (Table A.7), so any value in $[2, 10]$ yields essentially the same performance.

**Redundancy discount.** We define the discount factor as

$$\gamma(f_i, P) = \frac{1}{1 + \beta \cdot \text{sim}_{\max}(f_i, P)}, \quad (6)$$

where $\text{sim}_{\max}(f_i, P) = \max_{f_j \in P} \text{sim}(f_i, f_j)$ (and 0 if $P = \emptyset$), and $\text{sim}(f_i, f_j)$ is the Jaccard similarity between the top-$k_{\text{top}}$ ranked instances of $f_i$ and $f_j$, with $k_{\text{top}} = \lceil 0.1 \, N \rceil$ (top 10% by default; see App. A.4). We use maximum rather than average similarity to prevent near-duplicates of existing members, which is particularly harmful without validation labels.

The discount $\gamma(\cdot)$ induces diminishing returns: since $\text{sim}_{\max}(f_i, P \cup \{f\}) \geq \text{sim}_{\max}(f_i, P)$ for any added model $f$, the redundancy penalty grows with the ensemble, dampening redundant candidates. We term this objective submodular-inspired: strict submodularity would require the black-box predictor $\hat{G}$ itself to be monotone non-increasing in $P$, i.e., $\hat{G}(f_i \mid P \cup \{f\}) \leq \hat{G}(f_i \mid P)$ for any added model $f$. Enforcing this rigid constraint would compromise prediction accuracy without yielding additional practical benefits (e.g., greedy efficiency, adaptive stopping).

**Family-risk regularization.** To reduce downside risk, we introduce a family-level prior that penalizes candidates from algorithmic families with negative lower-tail historical gains. Using the family map $\text{fam} : \Omega \to \mathcal{F}$ defined in Sec. 3.2, for a family $F \in \mathcal{F}$ we define the family risk as the 10th percentile of oracle gains observed during meta-training:

$$\text{Risk}_F(\mathcal{M}) = Q_{0.10} \left( \{ G(f \mid P) \mid f \in \mathcal{M}_F, |P| \geq 1 \} \right), \quad (7)$$

and convert it into a non-negative penalty:

$$\pi_F = \max \left( 0, -\text{Risk}_F(\mathcal{M}) \right). \quad (8)$$

We set $\pi_{\text{fam}(f_i)} \equiv \pi_F$ for the proxy utility (Eq. 5), where $\mathcal{M}_F$ collects candidates in family $F$ across all meta-datasets and ensemble states. This percentile is fixed globally on meta-training trajectories and never tuned on target datasets (sensitivity in App. A.1, A.6). Intuitively, if a family occasionally produces strongly negative marginal gains, it

receives a higher penalty, discouraging harmful additions in the absence of labels. For unseen families ($\mathcal{M}_F = \emptyset$), we set $\pi_F = 0$, treating them as neutral to support zero-shot plug-and-play extensions without unsubstantiated penalties. This family-level prior is orthogonal to the redundancy discount: $\gamma$ blocks near-duplicates of selected members while $\pi_F$ blocks systematically weak families, addressing failure modes that diversity alone cannot detect. Critically, both signals require no target labels—$\gamma$ uses only score similarities on the target and $\pi_F$ is fixed from meta-training—keeping the pipeline fully unsupervised at deployment.

**Acceptance gates.** Before evaluating $\Delta U(f_i \mid P)$, we apply a stage-dependent hard filter, discarding candidates with $\hat{G}(f_i \mid P) < \tau_s$. To facilitate initial expansion while maintaining rigor later, we use a permissive gate $\tau_1$ for the first partner ($|P|=1$) and a stricter gate $\tau_2$ ($\tau_2 > \tau_1$) for all subsequent additions ($|P| \geq 2$). Both thresholds are tuned purely on meta-training data (App. A.6). This gate complements the adaptive stop ($\max_{f_i} \Delta U(f_i \mid P) \leq 0$): while $\tau_s$ filters out statistically unreliable raw predictions, the $\Delta U \leq 0$ condition rejects candidates whose redundancy or family-risk penalties outweigh their predicted gain. Empirically, collapsing the gate (setting $\tau_1 = \tau_2 = -0.01$, effectively accepting all candidates) drops AP from 0.4308 to 0.4257 (Table A.10), confirming that hard gating and soft stopping address distinct failure modes. Selection terminates when no candidate passes the gate or yields positive proxy utility.

### 3.4. Framework Algorithm

Combining the components, `MetaEns` first selects the primary $f_1^*$ via $\mathcal{S}$, then iteratively expands the ensemble: at each step, surviving candidates (those passing the stage-dependent acceptance gate $\tau^{(s)}$) are scored by the proxy utility $\Delta U(f_i \mid P)$, and the candidate with the highest utility is added. Selection terminates when $\max_{f_i} \Delta U(f_i \mid P) \leq 0$ or when $|P| = \eta$.

Crucially, the adaptive stop renders the budget $\eta$ a loose upper bound rather than an active hyperparameter: `MetaEns` self-terminates at an average ensemble size of 2.2 on the 39-dataset benchmark (Sec. 4.8). This adaptivity is a key advantage over fixed-$k$ selectors such as `ELECT-Top-k`, which must commit to a single ensemble size across heterogeneous targets and are consistently outperformed in our experiments (Table 1). Complete pseudocode is in App. A.7.

### 3.5. Complexity Analysis

`MetaEns` runs in $O(|\Omega|^2 N + \eta \, |\Omega| \, C_{\text{inf}})$: a one-time precomputation of pairwise score statistics, then greedy expansion at $O(|\Omega| \, C_{\text{inf}})$ per step (independent of $N$, $\eta \ll |\Omega|$ via adaptive stopping; full breakdown in App. A.8).

# 4. Experiments

## 4.1. Experimental Settings

**Datasets.** We use the 39-dataset benchmark from ELECT (Zhao et al., 2022), comprising real-world tabular datasets from ODDS (Rayana, 2016) and the DAMI benchmark (Campos et al., 2016). Sample sizes range from 129 to 49,534 (median 1,600), dimensionality from 5 to 1,555 (median 18), and contamination rates from $0.20\%$ to $45.78\%$ (median $6.25\%$), spanning numeric, categorical, and mixed-type tabular data. Full details are in App. A.9.

**Candidate Model Pool.** We construct a pool of 297 unsupervised outlier detectors spanning 8 families: IForest, LOF, kNN, HBOS, OCSVM, LODA, ABOD, and COF, generated by systematically varying hyperparameters within each family (e.g., $k \in \{5, 10, \ldots, 100\}$ for kNN/LOF). Full specifications are in App. A.10.

**Baselines.** We compare MetaEns against a supervised greedy oracle (using ground-truth labels for upper-bound iterative selection) and 19 unsupervised baselines. For fixed-size ensembles, we report the best oracle size $k$ by average AP, giving them a competitive advantage. The unsupervised baselines span: *single models and naïve/random ensembles* (Singleton, LOF, Global Best, IForest, RandNet (Chen et al., 2017), Random Ensemble, Mega Ensemble); *deep detectors* (RDA (Zhou & Paffenroth, 2017), DAGMM (Zong et al., 2018), DeepSVDD (Ruff et al., 2018), ROBOD (Ding et al., 2022), LUNAR (Goodge et al., 2022), DTE-C (Livernoche et al., 2024), TCCM (Li et al., 2025)); and *meta-learning selectors* (MetaOD (Zhao et al., 2021), LSCP (Zhao et al., 2019a), ELECT (Zhao et al., 2022), plus top-$k$ and random expansion ELECT variants). Deep methods follow the same unsupervised protocol. Full details are in App. A.11.

**Hyperparameter & Implementation Details.** No method uses target labels for tuning. Baselines use fixed defaults (PyOD/Scikit-learn) or the best fixed ensemble size by benchmark AP, while MetaEns tunes only on labeled meta-training data via leave-one-dataset-out validation. Detailed hyperparameter settings are provided in App. A.6, and implementation details are included in App. A.12.

**Metrics.** We evaluate all methods using five metrics: Average Precision (AP), AP Rank, ROC-AUC, Precision@$\pi$, and Max F1-score. Details of metrics are provided in App. A.13.

## 4.2. Main Results

Table 1 compares MetaEns against all baselines across 39 datasets (details in App. A.14). MetaEns achieves the best performance across all metrics. Notably, it outperforms

ELECT despite sharing the same primary detector, demonstrating the value of sequential partner selection.

Naïve strategies perform poorly: averaging all detectors or random formation fails to filter weak models, while static selectors (Global Best) trail adaptive methods.

Deep learning baselines show limited effectiveness on fully unsupervised tabular tasks (Grinsztajn et al., 2022; Shwartz-Ziv & Armon, 2022; Han et al., 2022), reflecting the difficulty of label-free tuning under class imbalance and mixed feature types rather than a general limitation of deep anomaly detection. Recent methods (ROBOD, LUNAR, DTE-C, TCCM) improve over older deep baselines but trail MetaEns, indicating cross-family selection is more robust than committing to a single neural family.

ELECT ensemble variants yield marginal gains over single-model selection, whereas MetaEns's context-aware expansion yields clear improvements, demonstrating effectiveness and efficiency via compact, adaptive ensembles.

## 4.3. Ablation Study

Table 2 ablates MetaEns's key mechanisms. Disabling diversity discounting ($\beta = 0$) degrades performance, while removing family-risk regularization ($\lambda_{\mathrm{fam}} = 0$) causes the largest drop, confirming its necessity for robust ensembles. Replacing the two-part meta-model with a single gain predictor also yields moderate degradation.

Sensitivity analyses for hyperparameters ($\beta$, $\lambda_{\mathrm{fam}}$, $\tau_1$, $\tau_2$) and family configurations are in App.A.1, A.6, and A.15.

## 4.4. Statistical Significance

We conduct paired one-sided Wilcoxon signed-rank tests across the 39 datasets. MetaEns achieves statistically significant AP improvements over all baselines, including recent deep detectors. The rank-based view is important as the benchmark is heterogeneous: average AP can be dominated by easier datasets, whereas rank captures consistent competitiveness across varying contamination rates and feature types. Full $p$-values and win rates are in App. A.16.

## 4.5. Modality Transfer to Image and Text Anomaly Detection

Because MetaEns uses only normalized score vectors, it transfers beyond tabular data. On 20 ADBench image/text extracted-feature datasets (15 MVTec-AD, 5 text), MetaEns outperforms the strongest baselines overall ($+0.0197$ AP vs. LUNAR) and on images ($+0.0257$ AP vs. ELECT-1); see App. A.17. This validates the score-only design: the meta-model is independent of raw dimensionality, modality, or detector internals, reusing the same state representation for image, text, and tabular scores.

*Table 1.* Main performance comparison on the 39 benchmark datasets. We report Average Precision (AP), AP Rank (median across datasets, lower is better), ROC-AUC, Precision@$\pi$ where $\pi$ is the number of true anomalies, Max-F1 over all thresholds, and the average selected ensemble size. Best results are in **bold**; second-best are underlined. Method abbreviations: `Singleton` is random single-model selection, `Mega Ensemble` averages all 297 detectors, and `ELECT Top-k` aggregates the top-$k$ models selected by ELECT.

| Method | AP ↑ | Rank ↓ | ROC-AUC ↑ | Prec@$\pi$ ↑ | Max-F1 ↑ | Ens Size |
|---|---|---|---|---|---|---|
| *Theoretical Upper Bound* | | | | | | |
| Greedy Oracle | 0.6877 | 1.0 | 0.8968 | 0.6504 | 0.6906 | 10 |
| *Single Model Baselines* | | | | | | |
| Singleton | $0.3495 \pm 0.0255$ | $147.9 \pm 22.0578$ | $0.7081 \pm 0.0248$ | $0.3304 \pm 0.0283$ | $0.4075 \pm 0.0190$ | 1 |
| LOF | $0.3513 \pm 0.0038$ | $120.1 \pm 2.2336$ | $0.7439 \pm 0.0063$ | $0.3297 \pm 0.0037$ | $0.4169 \pm 0.0051$ | 1 |
| Global Best | $0.3787 \pm 0.0074$ | $122.7 \pm 9.0314$ | $0.7583 \pm 0.0049$ | $0.3593 \pm 0.0093$ | $0.4266 \pm 0.0075$ | 1 |
| *Naïve & Random Ensembles* | | | | | | |
| IForest | $0.3858 \pm 0.0016$ | $117.0 \pm 5.8119$ | $0.7699 \pm 0.0018$ | $0.3619 \pm 0.0033$ | $0.4337 \pm 0.0022$ | 200 |
| RandNet | $0.3460 \pm 0.0018$ | $170.6 \pm 2.9136$ | $0.6865 \pm 0.0029$ | $0.3189 \pm 0.0034$ | $0.3927 \pm 0.0027$ | 20 |
| Mega Ensemble | $0.3970 \pm 0.0000$ | $100.0 \pm 0.0000$ | $0.7737 \pm 0.0000$ | $0.3782 \pm 0.0000$ | $0.4443 \pm 0.0000$ | 297 |
| Random Ensemble | $0.3759 \pm 0.0175$ | $124.3 \pm 13.6874$ | $0.7477 \pm 0.0106$ | $0.3608 \pm 0.0172$ | $0.4290 \pm 0.0140$ | 3 |
| *Deep Learning Baselines* | | | | | | |
| RDA | $0.2742 \pm 0.0065$ | $211.9 \pm 11.4741$ | $0.7063 \pm 0.0064$ | $0.2721 \pm 0.0090$ | $0.3515 \pm 0.0068$ | 1 |
| DAGMM | $0.2958 \pm 0.0124$ | $221.6 \pm 6.8508$ | $0.6676 \pm 0.0134$ | $0.3013 \pm 0.0135$ | $0.3681 \pm 0.0129$ | 1 |
| DeepSVDD | $0.2073 \pm 0.0115$ | $247.5 \pm 6.8516$ | $0.5905 \pm 0.0164$ | $0.2116 \pm 0.0147$ | $0.2968 \pm 0.0073$ | 1 |
| ROBOD | $0.3135 \pm 0.0026$ | $208.0 \pm 2.1602$ | $0.6665 \pm 0.0001$ | $0.2973 \pm 0.0024$ | $0.3621 \pm 0.0002$ | 1 |
| LUNAR | $0.3024 \pm 0.0045$ | $172.0 \pm 18.6815$ | $0.6566 \pm 0.0062$ | $0.3026 \pm 0.0098$ | $0.3745 \pm 0.0074$ | 1 |
| DTE-C | $0.3144 \pm 0.0012$ | $199.7 \pm 9.2376$ | $0.7469 \pm 0.0087$ | $0.3083 \pm 0.0033$ | $0.3971 \pm 0.0016$ | 1 |
| TCCM | $0.2929 \pm 0.0144$ | $188.3 \pm 28.2902$ | $0.6745 \pm 0.0126$ | $0.2834 \pm 0.0226$ | $0.3690 \pm 0.0111$ | 1 |
| *Meta-Learning & Ensemble Methods* | | | | | | |
| LSCP | $0.3484 \pm 0.0173$ | $124.3 \pm 11.1161$ | $0.7560 \pm 0.0096$ | $0.3441 \pm 0.0208$ | $0.4251 \pm 0.0123$ | 1 |
| MetaOD | $0.3989 \pm 0.0024$ | $101.0 \pm 7.6594$ | $0.7547 \pm 0.0014$ | $0.3746 \pm 0.0032$ | $0.4392 \pm 0.0022$ | 1 |
| ELECT+Random | $0.3981 \pm 0.0060$ | $102.2 \pm 9.2232$ | $0.7719 \pm 0.0068$ | $0.3778 \pm 0.0103$ | $0.4449 \pm 0.0067$ | 10 |
| ELECT (Top-1) | $0.4069 \pm 0.0063$ | $85.8 \pm 7.8712$ | $0.7734 \pm 0.0046$ | $\underline{0.3861} \pm 0.0059$ | $0.4519 \pm 0.0051$ | 1 |
| ELECT (Top-10) | $\underline{0.4117} \pm 0.0050$ | $\underline{83.2} \pm 6.8118$ | $\underline{0.7785} \pm 0.0038$ | $0.3856 \pm 0.0055$ | $\underline{0.4546} \pm 0.0043$ | 10 |
| MetaEns (**Ours**) | $\mathbf{0.4308} \pm 0.0064$ | $\mathbf{59.3} \pm 6.9610$ | $\mathbf{0.7867} \pm 0.0045$ | $\mathbf{0.4042} \pm 0.0063$ | $\mathbf{0.4681} \pm 0.0069$ | 2.2 |

*Table 2.* Ablation study on the 39 benchmark datasets. $\Delta$AP measures the degradation relative to the full model. Setting $\beta = 0$ removes the redundancy/diversity discount, setting $\lambda_{\mathrm{fam}} = 0$ removes family-risk regularization, and the single-part gain model replaces the two-part classifier–regressor architecture with one direct gain predictor.

| Variant | AP ↑ | AP Rank ↓ | $\Delta$AP |
|---|---|---|---|
| w/o diversity ($\beta$=0) | 0.4185 | 77.0 | $-0.0123$ |
| w/o family-risk ($\lambda_{\mathrm{fam}}$=0) | 0.3995 | 72.0 | $-0.0313$ |
| single-part gain model | 0.4133 | 87.0 | $-0.0175$ |
| MetaEns | **0.4308** | **59.3** | — |

### 4.6. Robustness to Initialization

A central question is whether `MetaEns`'s gains depend on specific primary selectors or strong initial models. We pair `MetaEns` with ELECT, LOF, IForest, and a random singleton, comparing initial and final AP (Fig. 2).

Across all configurations, `MetaEns` consistently improves over the starting model, with pronounced gains especially when the primary model underperforms. Rather than prop-agating initial errors, `MetaEns` recovers performance by selecting complementary detectors from diverse families, confirming that gains stem from diversity-aware partner selection rather than reliance on primary model quality.

### 4.7. Failure Mode Analysis

While `MetaEns` consistently outperforms baselines on average, it underperforms in two regimes: (i) very low-dimensional datasets (e.g., `Glass`, `Wilt`; $d \leq 9$), where classical detectors with strong inductive biases are already near-optimal—a known ensemble limitation (Zimek et al., 2013); and (ii) datasets where a single specialized detector achieves disproportionately high AP (e.g., `lympho`, `mnist`), causing mean-aggregation to dilute the dominant signal. See App. A.9 for dataset details.

Fig. 2 highlights a *Rescue Zone* (primary AP $< 0.4$): when all candidate models perform poorly, no complementary partner exists, and adaptive early stopping prevents further degradation. Per-dataset results are in App. A.14.

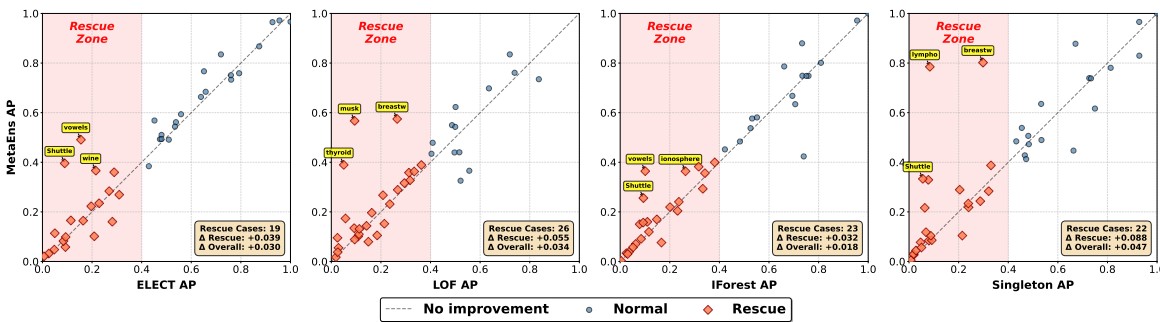

*Figure 2.* Robustness analysis across four different primary selectors: `ELECT`, `LOF`, `IForest`, and `Singleton` (random single-model selection). Each panel compares the primary model's performance ($x$-axis) against the final `MetaEns` ensemble ($y$-axis). Points above the diagonal indicate improvement. The shaded red "Rescue Zone" highlights where the primary model fails (AP < 0.4). `MetaEns` consistently rescues performance in these failure modes across all primary selectors, demonstrating that its diverse partner selection logic is robust and selector-agnostic.

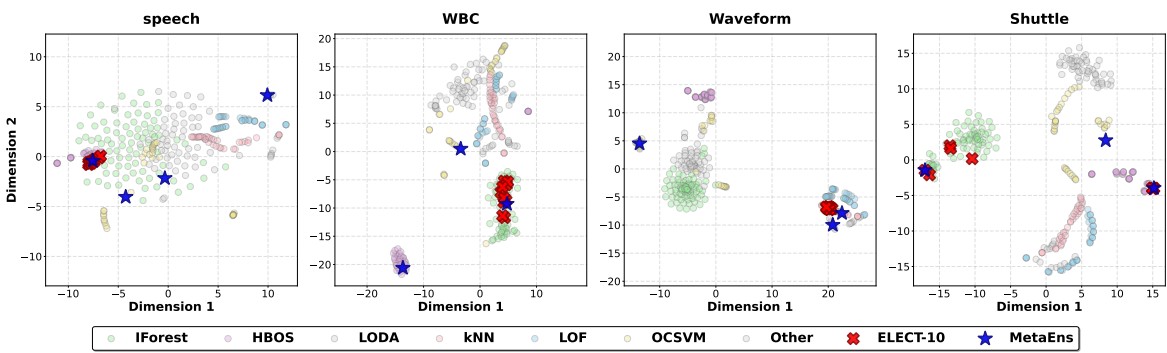

*Figure 3.* Model diversity visualization using `t-SNE` projection across four datasets. `ELECT-10` selections tend to cluster within a single family, whereas `MetaEns` selects models spanning multiple families, indicating greater ensemble diversity.

## 4.8. Model Diversity Analysis

To illustrate diversity differences, we project the 297-model pool into 2D via `t-SNE` (van der Maaten & Hinton, 2008) based on prediction-correlation distance (Fig. 3). Models cluster by family, reflecting similar outlier rankings within families and reinforcing that detectors from the same algorithmic paradigm often behave similarly. `ELECT-10` concentrates selections within a single cluster (e.g., `HBOS` for **Speech**, `IForest` for **WBC**), whereas `MetaEns` spans multiple clusters—four families on **Speech**, three on **WBC**—driven by family-risk regularization. This cross-family spread is especially valuable without validation labels and helps capture complementary decision patterns across diverse datasets. Quantitative diversity metrics are in App. A.18.

## 4.9. Pool Size Analysis

A detailed analysis of the effect of candidate pool size is provided in App. A.15. The study shows how `MetaEns`'s performance evolves as the number of available detectors increases, demonstrating that the method scales well and remains stable while benefiting from greater model diversity.

## 4.10. Effects of Meta-model Architectures

We study the choice of meta-model architectures in App. A.5. The comparison evaluates several alternative learning families for marginal gain prediction and candidate filtering, showing that the default `ExtraTrees` configuration achieves the best balance between predictive performance and computational efficiency.

## 4.11. Effect of Expanded Model Pools

To evaluate `MetaEns`'s scalability and robustness to model pool composition, we conduct additional experiments with an expanded pool of 310 models that incorporates neural networks. A detailed analysis of the expanded model pool is provided in App. A.19.

## 4.12. Effect of Ensemble Size

To understand how ensemble size affects performance, we analyze the relationship between the number of models and detection quality. A detailed analysis of the effect of ensemble size is provided in App. A.20.

# 5. Related Work

## 5.1. Unsupervised Outlier Detection

Unsupervised outlier detection spans statistical, density-based, tree-based, distance-based, one-class, subspace, and deep paradigms. Statistical methods such as HBOS (Goldstein & Dengel, 2012) identify anomalies in low-density regions under feature independence assumptions. Density- and distribution-based methods, including LOF (Breunig et al., 2000), COPOD (Li et al., 2020), and ECOD (Li et al., 2023), quantify deviations using local density or tail probabilities. Tree-based methods such as IForest (Liu et al., 2008) isolate anomalies via random partitioning, while distance-based methods including kNN (Chehreghani, 2016) and ODIN (Hautamäki et al., 2004) rely on spatial isolation. One-class classifiers such as OCSVM (Schölkopf et al., 1999) and SVDD (Tax & Duin, 2004) learn decision boundaries around normal data, whereas subspace-based methods like PCA (Chapel & Friguet, 2014) use reconstruction errors.

More recently, deep methods learn expressive normality representations through autoencoders (AEs) (Goodge et al., 2020), variational autoencoders (VAEs) (Xu et al., 2018), and generative adversarial networks (GANs) (Lim et al., 2018). Extensions such as Deep SVDD (Ruff et al., 2018), DAGMM (Zong et al., 2018), and RDA (Zhou & Paffenroth, 2017) combine representation learning with one-class objectives or density estimation. Recent methods further explore graph neural representations (LUNAR) (Goodge et al., 2022), diffusion-based anomaly scoring (DTE-C) (Livernoche et al., 2024), flow matching (TCCM) (Li et al., 2025), and transformers (Kim et al., 2024). Despite their effectiveness, these methods remain sensitive to architecture and hyperparameters, motivating ensemble-based strategies.

## 5.2. Ensembles for Outlier Detection

Ensemble methods improve robustness and stability by combining multiple detectors (Aggarwal & Sathe, 2015). Common aggregation strategies include score averaging, ranking, and voting, but naively combining all detectors can incur high cost, weak-model contamination, and limited adaptivity. Feature-bagging ensembles promote diversity through random subspaces (Noto et al., 2010; Lazarevic & Kumar, 2005), while tree-based ensembles such as IForest (Liu et al., 2008) embed diversity via randomized construction. Sequential ensembles refine performance by reweighting instances across rounds, as exemplified by XGBOD (Zhao & Hryniewicki, 2018). Stacking-based approaches combine heterogeneous detectors, including autoencoder (Chen et al., 2017) and GAN ensembles (Han et al., 2021), to preserve complementary behaviors. ROBOD (Ding et al., 2022) addresses hyperparameter sensitivity in deep outlier detection by constructing a scalable homogeneous hyper-ensemble. Yet, most existing methods rely on fixed aggregation schemes, a single architecture family, or upfront full-ensemble construction, limiting adaptivity in unsupervised settings. Unlike these approaches, MetaEns focuses on label-free model selection over heterogeneous detector families rather than score reweighting within a fixed architecture.

## 5.3. Model Selection

Model selection is well established in supervised learning via information-theoretic criteria such as AIC and BIC (Schwarz, 1978), cross-validation (Stone, 1974), structural risk minimization (Vapnik, 1998), and modern hyperparameter optimization methods such as Bayesian optimization (Snoek et al., 2012) and automated machine learning (Feurer et al., 2015). In contrast, unsupervised outlier model selection is harder due to missing labels, extreme class imbalance, and heterogeneous anomaly patterns. Early approaches therefore rely on internal validation heuristics from score distributions or stability under perturbations (Goix, 2016; Marques et al., 2020), which often correlate weakly with true detection performance.

Recent meta-learning approaches (Vilalta & Drissi, 2002; Hospedales et al., 2022) use labeled meta-datasets to generalize model selection across tasks. Representative methods include MetaOD (Zhao et al., 2021) and ELECT (Zhao et al., 2022), which mainly select individual detectors. Ensemble-based selection remains underexplored: LSCP (Zhao et al., 2019a) performs instance-wise detector selection but assumes local data consistency and requires the full ensemble in advance. In contrast, MetaEns performs sequential ensemble selection without local consistency assumptions and supports adaptive early stopping, enabling compact, high-performing ensembles in fully unsupervised settings.

# 6. Conclusion

We address the problem of unsupervised ensemble outlier model selection, where ensembles must be constructed without access to labels. We propose MetaEns, a meta-learned framework that predicts marginal ensemble gains and combines these with a submodular-inspired proxy objective to guide adaptive ensemble construction with early stopping. Extensive experiments on 39 real-world datasets show that MetaEns consistently outperforms strong unsupervised baselines while selecting substantially smaller ensembles, highlighting the importance of family-level diversification and risk control in unsupervised settings.

Future work includes exploring richer meta-representations to improve gain prediction under distribution shift, extending to streaming or non-stationary data, and developing theoretical or uncertainty-aware variants of the proxy objective to better understand unsupervised ensemble selection.

## Impact Statement

Unsupervised outlier detection is used widely in high-impact domains such as fraud detection, cybersecurity, healthcare monitoring, and scientific data analysis. By enabling adaptive and data-driven ensemble construction without requiring labeled data, `MetaEns` has the potential to improve the robustness and reliability of outlier detection systems deployed in practice. In particular, the ability to construct compact ensembles can reduce computational cost and energy consumption, which is beneficial for large-scale or resource-constrained settings.

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

# A. Appendix

## A.1. Ablation on Family Definition

To assess the sensitivity of `MetaEns` to the granularity of the family definition, we compare three variants: (1) no family-risk penalty ($\lambda_{\text{fam}} = 0$), (2) a *coarse-grained* definition mapping the 8 base algorithms into 3 broad super-families, and (3) our default *fine-grained* definition using all 8 base families independently.

The coarse-grained grouping maps algorithms as follows: *Density/Proximity* (`LOF`, `kNN`, `ABOD`, `COF`), *Isolation/Tree* (`IForest`), and *Linear/Probabilistic* (`HBOS`, `OCSVM`, `LODA`).

*Table A.1.* Ablation on family definition granularity. $\Delta$AP measures performance degradation relative to the full model.

| Variant | AP $\uparrow$ | Rank $\downarrow$ | $\Delta$AP |
|---|---|---|---|
| w/o family-risk | 0.3995 | 72.0 | −0.0313 |
| Coarse-grained | 0.4196 | 65.0 | −0.0112 |
| `MetaEns` Fine-grained | **0.4308** | **59.3** | — |

Table A.1 shows that fine-grained family definitions yield the strongest performance. Even the coarse-grained variant substantially outperforms the no-penalty baseline ($\Delta$AP $= -0.0112$ vs. $-0.0313$), demonstrating that `MetaEns` does not require a perfectly optimal taxonomy. However, broad groupings can cause *collateral penalties*: for instance, a strong `kNN` candidate may be suppressed because a `LOF` model underperformed, despite their differing sensitivity to local versus global anomaly patterns. The fine-grained definition ensures that a family's historical failure penalizes only its own hyperparameter variants, without unfairly suppressing distinct but conceptually related algorithms.

## A.2. Ablation on Ensemble Score Aggregation

Eq. 1 defines the ensemble score as the mean of member detector scores. We use the mean because it is the standard and most stable aggregation rule in unsupervised outlier ensembles: it preserves the continuous ranking signal contributed by all selected detectors while reducing dependence on any single base model. In contrast, max aggregation is highly sensitive to isolated false positives from a poorly calibrated detector, while min aggregation requires near-unanimous agreement and is therefore sensitive to false negatives. Median aggregation is more robust to extreme scores, but can discard fine-grained score variation from competent detectors.

To verify this design choice, we conduct a focused aggregation ablation across three random seeds on the 39 benchmark datasets, changing only the operator used to combine selected detector scores. Table A.2 shows that mean aggregation achieves the best AP. It significantly outperforms max and min aggregation under a one-sided Wilcoxon test

at $\alpha = 0.05$. The gap between mean and median is positive but not statistically significant, indicating that the median is a reasonable robust alternative but not a better default in our setting.

*Table A.2.* Ablation on ensemble score aggregation. $p$-values are from one-sided Wilcoxon tests against mean aggregation; $^{*}$ indicates significance at $\alpha = 0.05$.

| Combiner | AP (Mean $\pm$ Std) $\uparrow$ | $\Delta$AP | $p$-value |
|---|---|---|---|
| Mean (default) | **0.4308**±0.0064 | — | — |
| Median | 0.4182±0.0028 | −0.0126 | 0.377 |
| Max | 0.3778±0.0251 | −0.0530 | 0.049$^{*}$ |
| Min | 0.3736±0.0057 | −0.0572 | 0.022$^{*}$ |

These results support mean aggregation as the most stable default for `MetaEns`. This focused ablation isolates the score-combining operator and is therefore complementary to the main benchmark results in Table 1, where `MetaEns` achieves AP $0.4308 \pm 0.0064$ and AP rank $59.3 \pm 6.96$.

## A.3. Ablation on Training Trajectory Strategies

We evaluate two alternative offline training designs against the default oracle greedy rollout: (A) $\varepsilon$-*greedy extension*, where the greedy argmax is replaced by random sampling with probability $\varepsilon$; and (B) *reduced meta-dataset coverage*, using 25/50/75% of labeled meta-datasets. Results are averaged over 3 seeds on the 39-dataset benchmark.

*Table A.3.* Ablation on training trajectory strategies.

| Strategy | Mean AP $\uparrow$ | Avg. Rank $\downarrow$ |
|---|---|---|
| Greedy (ours, $\varepsilon$=0) | **0.4308**±0.0064 | **59.3**±6.96 |
| $\varepsilon$-greedy ($\varepsilon$=0.1) | 0.4156±0.0069 | 70.7±10.8 |
| $\varepsilon$-greedy ($\varepsilon$=0.2) | 0.4162±0.0036 | 77.7±1.2 |
| 25% meta-datasets | 0.3358±0.0079 | 109.0±0.0 |
| 50% meta-datasets | 0.3743±0.0053 | 96.0±3.5 |
| 75% meta-datasets | 0.4122±0.0013 | 92.0±14.0 |
| 100% (ours) | **0.4308**±0.0064 | **59.3**±6.96 |

Table A.3 shows that adding exploration ($\varepsilon > 0$) degrades both AP and rank, as random steps create a train/test distribution mismatch—the deployed policy is purely greedy, so training should match this structure. Meta-dataset coverage improves monotonically, confirming that diversity across meta-tasks is critical for transferable gain prediction.

## A.4. State Representation Features

**State Representation.** The state representation $\phi(f_i, f^*_{i-1}, P)$ is a fixed 61-dimensional feature vector that encodes the interaction between a candidate model $f_i$, the most recently selected detector $f^*_{i-1}$, and the current ensemble $P$. It consists of three 20-dimensional feature

blocks and one scalar feature, matching the decomposition in Eq. 3: (i) $\phi_{f_i, f_{i-1}^*}$, capturing pairwise interactions between the candidate and the last selected model; (ii) $\phi_{f_i, P}$, summarizing interactions between the candidate and the existing ensemble; (iii) $\phi_{f_{i-1}^*, P}$, summarizing interactions between the last selected model and the ensemble; and (iv) $|P|$, the current ensemble cardinality.

Each 20-dimensional block is composed of 15 base pairwise features and 5 context features, computed from normalized outlier score vectors $\mathbf{o}_f \in [0, 1]^N$, where scores are Min–Max normalized independently for each detector and dataset. Table A.4 provides the complete specification of all feature dimensions.

**Feature Aggregation for Ensemble Context.** For ensembles with $|P| > 1$, set-level context features are computed via mean pooling to ensure permutation invariance:

$$\phi_{f_{i-1}^*, P} = \frac{1}{|P| - 1} \sum_{f \in P \setminus \{f_{i-1}^*\}} \phi_{f_{i-1}^*, f}$$

$$\phi_{f_i, P} = \frac{1}{|P|} \sum_{f \in P} \phi_{f, f_i}$$

Here, the first equation measures the variance between the last selected model and the current ensemble. The second equation measures the variance between the candidate model and the current ensemble. When $|P| = 1$ (i.e., only the primary detector has been selected), $\phi_{f_{i-1}^*, P}$ is zero-padded, while $\phi_{f_i, P}$ is computed directly from the primary detector. This design maintains a fixed 61-dimensional representation across all selection stages and allows the meta-model to distinguish early-stage expansion from later iterative selection.

### A.5. Effects of Meta-Model Architectures

The effectiveness of `MetaEns` depends on the underlying models used to predict marginal gains ($f_{\text{reg}}$) and filter candidates ($f_{\text{cls}}$). In this section, we justify our default choice of `ExtraTrees` by comparing with five alternative learning families: `Random Forest` (Breiman, 2001), `XGBoost` (Chen & Guestrin, 2016), `LightGBM` (Ke et al., 2017), `Multi-Layer Perceptrons` (Goodfellow et al., 2016), and `Linear Models` (James et al., 2013). All methods use the same meta-features and training protocol.

To ensure a fair comparison, we employ strong, standardized configurations across baselines. Tree ensembles (`Random Forest`, `ExtraTrees`, `XGBoost`, `LightGBM`) use 500 estimators for classification tasks and 800 for regression tasks. Gradient boosting models (`XGBoost`, `LightGBM`) adopt a subsample ratio of 0.8 to improve generalization. The `Multi-Layer Perceptrons` model consists of two hidden layers with sizes $(100, 50)$, trained for 500 epochs.

`Linear Models` use Logistic Regression and Ridge Regression with default regularization.

Table A.5 summarizes the results. *Train (s)* denotes the average wall-clock time to train the meta-model on the meta-training set (leave-one-out protocol), while *Infer (s)* measures the time required to select models for a new dataset. `ExtraTrees` consistently achieves the best predictive performance, with the highest AP (0.4308) and best AP rank (59.3), while maintaining competitive training (4.15s) and inference (1.68s). `XGBoost` provides faster training but trails in AP (0.4110) and rank (108.0), suggesting overfitting under the relatively small meta-training regime. `Linear Models` offer the fastest inference (1.03s) but cannot capture the non-linear relationships required for accurate selection, resulting in substantially worse ranking performance. `Random Forest` is competitive in rank but is slower to train and less accurate than `ExtraTrees`; this supports the benefit of fully randomized splits as additional regularization. `Multi-Layer Perceptrons` perform poorly both in accuracy (lowest AP 0.3711) and computational efficiency, indicating that neural architectures are less suitable for this structured meta-learning task. Overall, the comparison shows that `ExtraTrees` provides the best accuracy–efficiency tradeoff for our two-part marginal-gain predictor, making it the most practical choice in our setting.

### A.6. Hyperparameter Settings

**Baseline tuning protocol.** To preserve the fully unsupervised evaluation setting, target-dataset labels are never used to tune any baseline. Single-model baselines are evaluated with fixed default configurations from standard implementations (e.g., PyOD and Scikit-learn) across all datasets. For baselines with an ensemble-size parameter $k$, we report the best fixed $k$ according to average AP over the benchmark, giving these baselines an oracle advantage over normal unsupervised deployment. The supervised greedy oracle is reported only as an upper bound and is not a deployable unsupervised method.

**`MetaEns` hyperparameters.** We configure the following hyperparameters, which remain fixed across all 39 test datasets to ensure fair comparison. For primary model selection via `ELECT` (Zhao et al., 2022), we adopt the default configuration. For marginal gain prediction, we train a two-part meta-model using `ExtraTrees`: (i) a classifier component for predicting improvement probability ($y_p = \mathbb{I}(g > 0)$) with balanced class weights, and (ii) a regression component for predicting improvement magnitude ($y_m = \max(0, g)$) optimized for MAE loss. For sequential selection, we set acceptance thresholds $\tau_1$ and $\tau_2$ for first-partner selection and iterative expansion, respectively. For diversity modulation, we configure discount strength $\beta$, family-risk weight $\lambda_{\text{fam}}$, and Jaccard similarity top-$k$ threshold $k_{\text{top}}$. The ensemble

*Table A.4.* Feature specification for pairwise model comparison. All features are computed from normalized outlier score vectors $\mathbf{o}_p, \mathbf{o}_q \in [0,1]^N$.

| Feature | Type | Mathematical Definition |
|---|---|---|
| *Base Pairwise Features (15 dimensions)* | | |
| pearson | Correlation | Pearson correlation: $\rho(\mathbf{o}_p, \mathbf{o}_q) = \frac{\text{cov}(\mathbf{o}_p, \mathbf{o}_q)}{\sigma_p \sigma_q}$ |
| tail_pearson | Correlation | Pearson correlation on $\mathcal{U} = \text{top-}k_p \cup \text{top-}k_q$ where $k = \lceil 0.1N \rceil$ |
| jaccard | Set similarity | Jaccard index: $J = |\text{top-}k_p \cap \text{top-}k_q| / |\text{top-}k_p \cup \text{top-}k_q|$ |
| rel_kurtosis | Moment ratio | Relative kurtosis: $\text{kurt}(\mathbf{o}_q) / \max(\text{kurt}(\mathbf{o}_p), 10^{-3})$ |
| tail_pos_disagreement | Distributional | Mean positive difference: $\mathbb{E}_{i \in \mathcal{U}}[\max(0, \mathbf{o}_{q,i} - \mathbf{o}_{p,i})]$ |
| centrality | Distributional | Correlation with pool mean: $\rho(\mathbf{o}_q, \bar{\mathbf{o}})$ where $\bar{\mathbf{o}} = \frac{1}{K} \sum_{j=1}^{K} \mathbf{o}_j$ |
| cand_std | Distributional | Standard deviation: $\sigma(\mathbf{o}_q)$ |
| pseudo_ap | Pseudo-label | Average Precision treating top-$k_p$ as positive class, $\mathbf{o}_q$ as predictions |
| pseudo_roc | Pseudo-label | ROC AUC treating top-$k_p$ as positive class, $\mathbf{o}_q$ as predictions |
| tail_entropy | Distributional | Shannon entropy: $H(\mathbf{o}_q) = -\sum_{b=1}^{10} p_b \log p_b$ where $p_b$ is 10-bin histogram |
| score_dist_l2 | Distance | $L_2$ distance of sorted vectors: $\|\text{sort}(\mathbf{o}_p) - \text{sort}(\mathbf{o}_q)\|_2$ |
| cosine_dist | Distance | Cosine distance: $1 - \frac{\mathbf{o}_p \cdot \mathbf{o}_q}{\|\mathbf{o}_p\|_2 \|\mathbf{o}_q\|_2}$ |
| tail_divergence | Distance | Mean absolute difference: $\mathbb{E}_{i \in \mathcal{U}}[|\mathbf{o}_{q,i} - \mathbf{o}_{p,i}|]$ |
| same_family | Categorical | Binary indicator: $\mathbb{I}[\text{fam}(p) = \text{fam}(q)]$ where $\text{fam} : \Omega \to \mathcal{F}$ maps models to families |
| same_as_ensemble_count | Set overlap | Count of partners $f \in P$ with $J(q, f) > 0.5$; equals 0 when $P = \emptyset$ |
| *Primary-Context Features (5 dimensions)* | | |
| prim_std | Distributional | Standard deviation of primary: $\sigma(\mathbf{o}_p)$ |
| prim_entropy | Distributional | Shannon entropy of primary score distribution: $H(\mathbf{o}_p)$ |
| prim_centrality | Distributional | Centrality of primary: $\rho(\mathbf{o}_p, \bar{\mathbf{o}})$ |
| prim_kurtosis | Moment | Excess kurtosis of primary: $\text{kurt}(\mathbf{o}_p)$ |
| prim_skewness | Moment | Skewness of primary: $\text{skew}(\mathbf{o}_p)$ |
| *Set-Level Scalar Features (1 dimension)* | | |
| $|P|$ | Cardinality | Number of partners: $|P| \in \{0, 1, 2, \ldots\}$ |

*Table A.5.* Comparison of meta-model architectures for `MetaEns` selection quality.

| Method | AP (Mean $\pm$ Std) $\uparrow$ | Rank $\downarrow$ | ROC-AUC $\uparrow$ | Train (s) $\downarrow$ | Infer (s) $\downarrow$ |
|---|---|---|---|---|---|
| ExtraTrees (**Ours**) | **0.4308** $\pm$ 0.0064 | **59.3** $\pm$ 6.9610 | **0.7867** $\pm$ 0.0045 | 4.1518 | 1.6839 |
| Linear Models | 0.4056 $\pm$ 0.0015 | 105.2 $\pm$ 2.5000 | 0.7737 $\pm$ 0.0010 | 6.9150 | **1.0327** |
| XGBoost | 0.4110 $\pm$ 0.0055 | 108.0 $\pm$ 8.2000 | 0.7664 $\pm$ 0.0040 | **1.7408** | 1.4910 |
| Random Forest | 0.4043 $\pm$ 0.0062 | 77.0 $\pm$ 7.1000 | 0.7617 $\pm$ 0.0050 | 15.5561 | 1.5200 |
| LightGBM | 0.3899 $\pm$ 0.0058 | 99.0 $\pm$ 8.5000 | 0.7600 $\pm$ 0.0042 | 1.9412 | 1.3405 |
| Multi-Layer Perceptrons | 0.3711 $\pm$ 0.0115 | 110.0 $\pm$ 12.5000 | 0.7607 $\pm$ 0.0090 | 35.6667 | 10.7667 |

size budget is $\eta$ with adaptive early stopping. All hyperparameters are tuned via leave-one-out cross-validation on meta-training data. Table A.6 lists the search grids for all tunable proxy objective hyperparameters.

**Sensitivity analysis.** Table A.6 summarizes the search grids used. Tables A.7, A.8, A.9, and A.10 report sensitivity across the 39 benchmark datasets, with each row averaged over 3 seeds. The sensitivity tables explore the operational range of each hyperparameter, while the main ablation isolates the *structural effect* of fully disabling a component. For $\lambda_{\text{fam}}$, the default value 0.2 achieves peak performance: $\lambda_{\text{fam}}=0$ disables family-risk regularization (matching the "w/o family-risk" row in Table 2), while larger values ($\lambda_{\text{fam}} \geq 0.4$) over-regularize and saturate at a stable plateau (0.4231 AP) where the penalty consistently rejects the same candidate set across all tested values up to 1.5. Setting $\tau$ too high causes premature early stopping; the default small positive threshold (0.001/0.005) does not

require fine-tuning. For the family-risk percentile, the 10th percentile is a fixed global lower-tail choice rather than a target-dataset-tuned parameter. It penalizes families with occasional catastrophic negative gains while preserving families whose average behavior may still contribute orthogonal diversity, thereby offering greater robustness.

Table A.8 shows that `MetaEns` is robust from $0\%$ to $10\%$, so its performance does not hinge on a finely tuned percentile. By contrast, more aggressive thresholds like $25\%$ and $50\%$ noticeably hurt AP and rank by over-penalizing candidate families and suppressing useful diversity. The 10th percentile therefore strikes a conservative balance between downside risk and cross-family complementarity.

### A.7. Framework Algorithm

**Offline Meta-training Procedure.** Algorithm 1 summarizes the offline meta-training phase of `MetaEns`. Given labeled meta-datasets $\mathcal{M} = \{(\mathbf{M}_\ell, \mathbf{y}_\ell)\}$ and a detector pool

*Table A.6.* Hyperparameter search grids.

| Hyperparameter | Role | Search Grid |
|---|---|---|
| $\tau_1, \tau_2$ | Acceptance thresholds | $\{-0.01, 0, 0.001/0.005, 0.01, 0.05, 0.1\}$ |
| $\beta$ | Diversity discount strength | $\{0.5, 1, 2, 3, 5, 10\}$ |
| $\lambda_{\text{fam}}$ | Family-risk weight | $\{0, 0.2, 0.4, 0.6, 0.8, 1.0, 1.5\}$ |

*Table A.7.* Sensitivity to $\beta$ (diversity discount strength).

| $\beta$ | AP (Mean $\pm$ Std) $\uparrow$ | ROC-AUC $\uparrow$ | Rank $\downarrow$ |
|---|---|---|---|
| 0.5 | $0.4304 \pm 0.0057$ | 0.7842 | 60.2 |
| 1 | $0.4304 \pm 0.0057$ | 0.7842 | 60.2 |
| 2 | $0.4308 \pm 0.0064$ | 0.7867 | 59.3 |
| 3 (def.) | $\mathbf{0.4308 \pm 0.0064}$ | **0.7867** | **59.3** |
| 5 | $0.4306 \pm 0.0058$ | 0.7841 | 59.8 |
| 10 | $0.4308 \pm 0.0058$ | 0.7841 | 59.8 |

*Table A.8.* Sensitivity to the family-risk percentile $\tau_{\text{risk}}$ used in Eq. 7. The default 10% threshold matches the main result reported in Table 1.

| $\tau_{\text{risk}}$ | AP (Mean $\pm$ Std) $\uparrow$ | AP Rank $\downarrow$ |
|---|---|---|
| 0% (no risk penalty) | $0.4269 \pm 0.0066$ | 67.7 |
| 5% | $0.4272 \pm 0.0066$ | 67.7 |
| 10% (def.) | $\mathbf{0.4308 \pm 0.0064}$ | **59.3** |
| 25% | $0.3368 \pm 0.0048$ | 143.3 |
| 50% (median) | $0.3142 \pm 0.0087$ | 170.0 |

*Table A.9.* Sensitivity to $\lambda_{\text{fam}}$ (family-risk weight).

| $\lambda_{\text{fam}}$ | AP (Mean $\pm$ Std) $\uparrow$ | ROC-AUC $\uparrow$ | Rank $\downarrow$ |
|---|---|---|---|
| 0 | $0.3995 \pm 0.0041$ | 0.7111 | 72.0 |
| 0.2 (def.) | $\mathbf{0.4308 \pm 0.0064}$ | **0.7867** | **59.3** |
| 0.4 | $0.4231 \pm 0.0065$ | 0.7802 | 68.3 |
| 0.6 | $0.4231 \pm 0.0065$ | 0.7802 | 68.3 |
| 0.8 | $0.4231 \pm 0.0065$ | 0.7802 | 68.3 |
| 1.0 | $0.4231 \pm 0.0065$ | 0.7802 | 68.3 |
| 1.5 | $0.4231 \pm 0.0065$ | 0.7802 | 68.3 |

*Table A.10.* Sensitivity to $\tau_1, \tau_2$ (acceptance thresholds).

| $\tau_1, \tau_2$ | AP (Mean $\pm$ Std) $\uparrow$ | ROC-AUC $\uparrow$ | Rank $\downarrow$ |
|---|---|---|---|
| $-0.01, -0.01$ | $0.4257 \pm 0.0060$ | 0.7836 | 68.7 |
| $0, 0$ | $0.4257 \pm 0.0060$ | 0.7836 | 68.7 |
| $0.001, 0.005$ (def.) | $\mathbf{0.4308 \pm 0.0064}$ | **0.7867** | **59.3** |
| $0.01, 0.01$ | $0.4041 \pm 0.0052$ | 0.7461 | 86.8 |
| $0.05, 0.05$ | $0.4133 \pm 0.0000$ | 0.7792 | 79.1 |
| $0.1, 0.1$ | $0.4133 \pm 0.0000$ | 0.7792 | 79.1 |

$\Omega$, we simulate oracle greedy ensemble construction by iteratively expanding an ensemble $P \subseteq \Omega$. Starting from an oracle-selected primary detector $f_1^*$, at each step we evaluate every candidate $f_i \in \Omega \setminus P$ using its state representation $\phi(f_i, f_{i-1}^*, P)$ and compute the true marginal gain $G(f_i \mid P)$ in Average Precision. These state–gain pairs are used to train a two-part meta-model that estimates the probability and magnitude of marginal improvement, yielding a predictor $\hat{G}(f_i \mid P)$ that is later used to guide label-free online ensemble selection.

**Online Model Selection Procedure.** Algorithm 2 describes the online ensemble selection procedure of `MetaEns` for a new unlabeled dataset. Starting from a primary detector $f_1^*$ selected by an unsupervised criterion, we iteratively construct an ensemble $P$ by adding candidates $f_i \in \Omega \setminus P$ that maximize the proxy marginal utility $\Delta U(f_i \mid P)$. The proxy utility combines the predicted marginal gain $\hat{G}(f_i \mid P)$ from the meta-model with redundancy discounting and family-risk regularization, thereby enforcing diminishing returns as the ensemble grows. The selection process terminates automatically when $\max_{f_i} \Delta U(f_i \mid P) \leq 0$ or when the budget $|P| = \eta$ is reached, yielding compact, dataset-adaptive ensembles without access to labels.

## A.8. Complexity Analysis

We conduct a computational complexity analysis to clarify how the proposed framework scales with the size of the candidate model pool and dataset size.

`MetaEns` operates in $O(|\Omega|^2 \cdot N + \eta|\Omega| \cdot C_{\text{inf}})$ time, where $|\Omega|$ is the size of the candidate model pool, $N$ is the number of samples in the target dataset, $\eta$ is the ensemble budget, and $C_{\text{inf}}$ is the meta-model inference cost. The $O(|\Omega|^2 \cdot N)$ term corresponds to one-time feature pre-computation of similarity statistics for the target dataset: correlation-based features require $O(|\Omega|^2 \cdot N)$ operations, while Jaccard similarity over top-$k_{\text{top}}$ instances requires $O(|\Omega| \cdot N)$ time for extracting top-$k_{\text{top}}$ indices and $O(|\Omega|^2 \cdot k_{\text{top}})$ time for set intersections, becoming independent of $N$ after ranking. The interactive selection phase has complexity $O(\eta|\Omega| \cdot C_{\text{inf}})$ and is independent of $N$. For tree-based meta-models, $C_{\text{inf}} = O(n_{\text{trees}} \cdot d_\phi)$, where $n_{\text{trees}}$ is the number of trees and $d_\phi$ is the feature dimensionality.

## A.9. Datasets

We conduct experiments on the 39-dataset benchmark introduced in `ELECT` (Zhao et al., 2022), which comprises independent, real-world tabular anomaly detection tasks sourced from ODDS (Rayana, 2016) and the DAMI benchmark (Campos et al., 2016). Detailed characteristics of

**Algorithm 1** `MetaEns` Offline Meta-training: Oracle Rollouts for Marginal-Gain Supervision

**Require:** Labeled meta-datasets $\mathcal{M} = \{(\mathbf{M}_\ell, \mathbf{y}_\ell)\}_{\ell=1}^{L}$; detector pool $\Omega$; budget $\eta$; rollout length $T \leq \eta$; state function $\phi(f_i, f_{i-1}^*, P)$

**Ensure:** Trained meta-models $(f_{\text{cls}}, f_{\text{reg}})$

1: Initialize $\mathcal{D}_{\text{cls}} \leftarrow \emptyset, \mathcal{D}_{\text{reg}} \leftarrow \emptyset$
2: **for all** $(\mathbf{M}, \mathbf{y}) \in \mathcal{M}$ **do**
3:    (**Oracle primary**)
4:    $f_1^* \leftarrow \arg\max_{f \in \Omega} \text{AP}(f(\mathbf{M}), \mathbf{y})$
5:    $P \leftarrow \{f_1^*\}$
6:    $f_{\text{last}} \leftarrow f_1^*$
7:    **for** $t = 2$ to $T$ **do**
8:      **for all** $f_i \in \Omega \setminus P$ **do**
9:        $\phi_i \leftarrow \phi(f_i, f_{\text{last}}, P)$
10:       $G(f_i \mid P) \leftarrow \text{AP}(\mathbf{o}_{P \cup \{f_i\}}, \mathbf{y}) - \text{AP}(\mathbf{o}_P, \mathbf{y})$
11:       $y_{\text{cls}} \leftarrow \mathbb{I}(G(f_i \mid P) > 0)$
12:       $\mathcal{D}_{\text{cls}} \leftarrow \mathcal{D}_{\text{cls}} \cup \{(\phi_i, y_{\text{cls}})\}$
13:       **if** $y_{\text{cls}} = 1$ **then**
14:         $y_{\text{reg}} \leftarrow G(f_i \mid P)$
15:         $\mathcal{D}_{\text{reg}} \leftarrow \mathcal{D}_{\text{reg}} \cup \{(\phi_i, y_{\text{reg}})\}$
16:       **end if**
17:      **end for**
18:      $f^* \leftarrow \arg\max_{f_i \in \Omega \setminus P} G(f_i \mid P)$
19:      **if** $G(f^* \mid P) \leq 0$ **then**
20:        **break**
21:      **end if**
22:      $P \leftarrow P \cup \{f^*\}$; $f_{\text{last}} \leftarrow f^*$
23:      **if** $|P| \geq \eta$ **then**
24:        **break**
25:      **end if**
26:    **end for**
27: **end for**
28: Train classifier $f_{\text{cls}}$ on $\mathcal{D}_{\text{cls}}$ (balanced weights)
29: Train regressor $f_{\text{reg}}$ on $\mathcal{D}_{\text{reg}}$ (positives only)
30: **return** $(f_{\text{cls}}, f_{\text{reg}})$

**Algorithm 2** `MetaEns`: Online Selection via Submodular-Inspired Proxy Maximization

**Require:** dataset $\mathbf{X}$, candidate pool $\Omega$, meta-model $M(\cdot)$ producing $\hat{G}$, state representation $\phi(\cdot)$, family mapping fam : $\Omega \to \mathcal{F}$, thresholds $\tau_1, \tau_2$, budget $\eta \geq 1$, diversity $\beta$, risk $\lambda_{\text{fam}}$, $k_{\text{top}}$

**Ensure:** Ensemble $P$

1: *Primary model selection using* `ELECT` *as* $\mathcal{S}(\cdot)$
2: $f_1^* \leftarrow \mathcal{S}(\mathbf{X}, \Omega)$
3: $P \leftarrow \{f_1^*\}$
4: $\Omega_{\text{pool}} \leftarrow \Omega \setminus \{f_1^*\}$
5: *Stage 1: Select first expansion model*
6: **for all** $f_i \in \Omega_{\text{pool}}$ **do**
7:    $\hat{G}(f_i \mid P) \leftarrow M.\text{predict}(\phi(f_i, f_1^*, P))$
8: **end for**
9: $f_2^* \leftarrow \arg\max_{f_i \in \Omega_{\text{pool}}} \hat{G}(f_i \mid P)$
10: **if** $\hat{G}(f_2^* \mid P) < \tau_1$ **then**
11:    **return** $P$
12: **end if**
13: $P \leftarrow P \cup \{f_2^*\}$
14: *Stage 2: Iterative expansion*
15: **while** $|P| < \eta$ **do**
16:    $u_{\text{best}} \leftarrow 0$
17:    **for all** $f_i \in \Omega_{\text{pool}} \setminus P$ **do**
18:      $\hat{G}(f_i \mid P) \leftarrow M.\text{predict}\big(\phi(f_i, f_{|P|-1}^*, P)\big)$
19:      **if** $\hat{G}(f_i \mid P) < \tau_2$ **then**
20:        **continue**
21:      **end if**
22:      Compute $\Delta U(f_i \mid P)$
23:      **if** $\Delta U(f_i \mid P) > u_{\text{best}}$ **then**
24:        $f^* \leftarrow f_i$
25:        $u_{\text{best}} \leftarrow \Delta U(f_i \mid P)$
26:      **end if**
27:    **end for**
28:    **if** $u_{\text{best}} \leq 0$ **then**
29:      **break**
30:    **end if**
31:    $P \leftarrow P \cup \{f^*\}$
32: **end while**
33: **return** $P$

all datasets used in our experiments are summarized in Table A.11.

## A.10. Candidate Model Pool

We construct a diverse candidate pool of $M = 297$ unsupervised outlier detection models by systematically varying hyperparameters within 8 widely-used algorithmic families. For each dataset, all 297 models are pre-fitted and their anomaly scores cached for efficient meta-learning experiments. Table A.12 summarizes the configuration.

## A.11. Baselines

We compare the proposed `MetaEns` framework against 19 unsupervised baselines categorized into four groups. For ensemble baselines dependent on a size parameter $k$, we report the best $k$ performance—the fixed ensemble size that yields the highest average AP across the benchmark—to ensure a strong competitive baseline.

**Single Baselines:** (1) `Singleton`: Selects a single model uniformly at random from the candidate pool; (2) `LOF` (Breunig et al., 2000); (3) `Global Best`: A static meta-learner selecting the single model with the highest average performance across all training datasets;

**Naïve & Random Ensemble Baselines:** (4) `IForest` (Liu et al., 2008): A tree-based ensemble that isolates anomalies using random partitions; (5) `RandNet` (Chen et al., 2017): An ensemble of autoencoders with randomized connectivity to mitigate overfitting and enhance diversity; (6) `Random Ensemble`: An ensemble of $k$ randomly selected models; (7) `Mega Ensemble`: A naïve ensemble that averages outlier scores from all 297 models in the pool.

**Deep Learning Baselines:** (8) `RDA` (Zhou & Paffenroth, 2017): A robust deep autoencoder that decomposes input into low-dimensional manifold and sparse noise components; (9) `DAGMM` (Zong et al., 2018): An end-to-end framework jointly optimizing a compression network and

*Table A.11.* Summary of benchmark datasets.

| No. | Dataset | #Samples | Dimensionality | Contamination (%) | Type |
|-----|---------|----------|----------------|-------------------|------|
| *ODDS Library (21 datasets)* | | | | | |
| 1 | annthyroid | 7,200 | 6 | 7.42 | Mixed |
| 2 | arrhythmia | 452 | 274 | 14.60 | Mixed |
| 3 | breastw | 683 | 9 | 34.99 | Numeric |
| 4 | glass | 214 | 9 | 4.21 | Numeric |
| 5 | ionosphere | 351 | 33 | 35.90 | Numeric |
| 6 | letter | 1,600 | 32 | 6.25 | Numeric |
| 7 | lympho | 148 | 18 | 4.05 | Categorical |
| 8 | mammography | 11,183 | 6 | 2.32 | Numeric |
| 9 | mnist | 7,603 | 100 | 9.21 | Numeric |
| 10 | musk | 3,062 | 166 | 3.17 | Numeric |
| 11 | optdigits | 5,216 | 64 | 2.88 | Numeric |
| 12 | pendigits | 6,870 | 16 | 2.27 | Numeric |
| 13 | pima | 768 | 8 | 34.90 | Numeric |
| 14 | satellite | 6,435 | 36 | 31.64 | Numeric |
| 15 | satimage-2 | 5,803 | 36 | 1.22 | Numeric |
| 16 | speech | 3,686 | 400 | 1.65 | Numeric |
| 17 | thyroid | 3,772 | 6 | 2.47 | Mixed |
| 18 | vertebral | 240 | 6 | 12.50 | Numeric |
| 19 | vowels | 1,456 | 12 | 3.43 | Numeric |
| 20 | wbc | 378 | 30 | 5.56 | Categorical |
| 21 | wine | 129 | 13 | 7.75 | Numeric |
| *DAMI Benchmark (18 datasets)* | | | | | |
| 22 | ALOI | 49,534 | 27 | 3.04 | Numeric |
| 23 | Annthyroid | 7,129 | 21 | 7.49 | Mixed |
| 24 | Arrhythmia | 450 | 259 | 45.78 | Mixed |
| 25 | Cardiotocography | 2,114 | 21 | 22.04 | Numeric |
| 26 | Glass | 214 | 7 | 4.21 | Numeric |
| 27 | HeartDisease | 270 | 13 | 44.44 | Mixed |
| 28 | InternetAds | 1,966 | 1,555 | 18.72 | Mixed |
| 29 | PageBlocks | 5,393 | 10 | 9.46 | Numeric |
| 30 | PenDigits | 9,868 | 16 | 0.20 | Numeric |
| 31 | Pima | 768 | 8 | 34.90 | Numeric |
| 32 | Shuttle | 1,013 | 9 | 1.28 | Numeric |
| 33 | SpamBase | 4,207 | 57 | 39.91 | Numeric |
| 34 | Stamps | 340 | 9 | 9.12 | Numeric |
| 35 | WBC | 223 | 9 | 4.48 | Numeric |
| 36 | WDBC | 367 | 30 | 2.72 | Categorical |
| 37 | WPBC | 198 | 33 | 23.74 | Numeric |
| 38 | Waveform | 3,443 | 21 | 2.90 | Numeric |
| 39 | Wilt | 4,819 | 5 | 5.33 | Numeric |

a Gaussian Mixture Model for density estimation; (10) DeepSVDD (Ruff et al., 2018): Maps data into a minimum volume hypersphere to extract common factors of variation; (11) ROBOD (Ding et al., 2022): A deep hyper-ensemble that aggregates multiple MLP detectors trained under varied hyperparameters and unsupervised contamination assumptions; (12) LUNAR (Goodge et al., 2022): A graph-based contrastive representation learner for tabular anomaly detection; (13) DTE-C (Livernoche et al., 2024): A diffusion-based generative model for tabular anomaly detection with a transductive scoring protocol aligned to our benchmark; (14) TCCM (Li et al., 2025): A flow-matching generative model for tabular anomaly detection, evaluated with the same protocol as DTE-C.

**Meta-Learning and Ensemble Baselines:** (15) MetaOD (Zhao et al., 2021): A model selection as a cold-start recommendation problem using matrix factorization on historical meta-features; (16) LSCP (Zhao et al., 2019a): An ensemble framework that selects competent base detectors for local regions of test instances; (17)

ELECT (Zhao et al., 2022): The current state-of-the-art model selector, which identifies the single best model based on performance-driven task similarity. (18) ELECT-$k$ (equivalently ELECT (Top-$k$); specific instances ELECT-1 and ELECT-10): An ensemble baseline that aggregates the top-$k$ models ranked by ELECT. This tests whether simple rank-based selection is sufficient compared to our context-aware approach. (19) ELECT+Random (abbreviated ELECT+R in tables): A hybrid baseline that initializes with the high-quality primary model selected by ELECT but expands the ensemble with random partners.

**Theoretical Upper Bound:** (20) Greedy Oracle: A fully supervised upper bound that utilizes ground-truth labels to iteratively select the candidate maximizing marginal AP at each step. This baseline quantifies the maximum potential performance achievable by a greedy sequential selection strategy under ideal supervision.

*Table A.12.* Candidate model pool specification ($M = 297$ total models).

| Family | Count | Hyperparameter Grid |
|---|---|---|
| kNN (Chehreghani, 2016) | 36 | method $\in$ {largest, mean, median}; $k \in \{1, 5, 10, 15, 20, 25, 50, 60, 70, 80, 90, 100\}$ |
| LOF (Breunig et al., 2000) | 36 | metric $\in$ {euclidean, manhattan, minkowski}; $k \in \{1, 5, 10, 15, 20, 25, 50, 60, 70, 80, 90, 100\}$ |
| IForest (Liu et al., 2008) | 81 | $n_{\text{estimators}} \in \{10, 20, 30, 40, 50, 75, 100, 150, 200\}$; max_samples $\in \{0.1, 0.2, \dots, 0.9\}$ |
| HBOS (Goldstein & Dengel, 2012) | 40 | $n_{\text{bins}} \in \{5, 10, 20, 30, 40, 50, 75, 100\}$; tolerance $\in \{0.1, 0.2, 0.3, 0.4, 0.5\}$ |
| OCSVM (Schölkopf et al., 1999) | 36 | kernel $\in$ {linear, poly, rbf, sigmoid}; $\nu \in \{0.1, 0.2, \dots, 0.9\}$ |
| LODA (Pevný, 2016) | 54 | $n_{\text{bins}} \in \{5, 10, 15, 20, 25, 30\}$; $n_{\text{cuts}} \in \{10, 20, 30, 40, 50, 75, 100, 150, 200\}$ |
| ABOD (Kriegel et al., 2008) | 7 | $n_{\text{neighbors}} \in \{3, 5, 10, 15, 20, 25, 50\}$ |
| COF (Tang et al., 2002) | 7 | $n_{\text{neighbors}} \in \{3, 5, 10, 15, 20, 25, 50\}$ |
| **Total** | **297** | |

## A.12. Implementation Details

We implement the proposed framework using PyTorch 1.13 (Paszke et al., 2019) for neural components and Scikit-learn 1.2 (Pedregosa et al., 2011) and PyOD 1.0 (Zhao et al., 2019b) for base outlier detectors in Python 3.9. All experiments were executed on a high-performance computing cluster equipped with dual NVIDIA RTX 3090 GPUs (24GB VRAM), AMD EPYC 7742 64-Core Processors, and 512GB RAM. The source code is available at `https://github.com/ph-phuc/MetaEns`. For preprocessing, numeric features are scaled using RobustScaler (median removal and scaling according to the interquartile range) to preserve the integrity of global outliers. Categorical features are transformed via target encoding with leave-one-out regularization ($\alpha = 10$) to prevent data leakage. We add binary missingness indicators and impute missing values with median/mode. Since raw anomaly scores from different algorithms operate on vastly different scales, we strictly apply Min-Max Normalization per model per dataset to all model outputs before aggregation. We set the default random seed to 42. Aggregate results are reported as mean $\pm$ std over multiple seeds, as noted in each section, while per-dataset tables use seed 42.

## A.13. Evaluation Metrics

Let $\{(\mathbf{x}_i, y_i)\}_{i=1}^{N}$ denote an evaluation dataset, where $y_i \in \{0, 1\}$ indicates whether instance $\mathbf{x}_i$ is anomalous ($y_i = 1$) or normal ($y_i = 0$). Let $\mathbf{o} = (o_1, \dots, o_N)$ denote the outlier score vector produced by a method, where larger values indicate higher anomaly likelihood. Let $\pi = \sum_{i=1}^{N} y_i$ denote the number of anomalies in the dataset. All ranking-based metrics are computed by sorting instances in descending

order of $o_i$.

**Average Precision (AP).** Average Precision summarizes the area under the precision–recall curve:

$$\text{AP} = \frac{1}{\pi} \sum_{k=1}^{N} \mathbb{I}(y_{(k)} = 1) \, P(k), \qquad (9)$$

where $y_{(k)}$ is the label of the instance ranked at position $k$, and $P(k)$ is the precision at cutoff $k$:

$$P(k) = \frac{1}{k} \sum_{j=1}^{k} \mathbb{I}(y_{(j)} = 1) \qquad (10)$$

**AP Rank.** Let $r_{f_i, \mathbf{X}_j}$ denote the rank of method $f_i$ on dataset $\mathbf{X}_j$ according to Average Precision, where rank 1 indicates the best-performing method. The AP Rank of method $f_i$ is defined as the median of per-dataset ranks:

$$\text{Rank}_{f_i} = \text{median}\big(\{r_{f_i, \mathbf{X}_j} : \mathbf{X}_j \in \mathcal{D}\}\big), \qquad (11)$$

where $\mathcal{D}$ denotes the set of benchmark datasets. We use the median to be robust to per-dataset outliers in rank position. Lower values indicate better overall performance.

**ROC-AUC.** The ROC-AUC measures the probability that an anomalous instance receives a higher outlier score than a normal instance:

$$\text{AUC} = \mathbb{P}(o^+ > o^-) = \frac{\sum_{i:y_i=1} \sum_{j:y_j=0} \mathbb{I}(o_i > o_j)}{\pi(N - \pi)}, \qquad (12)$$

where $o^+$ and $o^-$ denote outlier scores of anomalous and normal instances, respectively.

**Precision@$\pi$.** Precision@$\pi$ evaluates the accuracy of the top-ranked predictions at the exact number of ground-truth anomalies and is defined as

$$\text{Precision@}\pi = \frac{1}{\pi} \sum_{k=1}^{\pi} \mathbb{I}\big(y_{(k)} = 1\big), \qquad (13)$$

where $\pi = \sum_{i=1}^{N} y_i$ is the number of anomalous instances in the dataset and $y_{(k)}$ is the label of the instance ranked at position $k$. This metric reflects practical inspection scenarios where only the top-ranked instances are examined.

**Max F1-score.** For a decision threshold $\tau$ applied to outlier scores, predicted labels are defined as follows.

$$\hat{y}_i(\tau) = \mathbb{I}(o_i \geq \tau) \qquad (14)$$

Precision and recall at threshold $\tau$ are given as follows.

$$\begin{aligned}
\text{Prec}(\tau) &= \frac{\sum_i \mathbb{I}(\hat{y}_i(\tau) = 1 \wedge y_i = 1)}{\sum_i \mathbb{I}(\hat{y}_i(\tau) = 1)} \\
\text{Rec}(\tau) &= \frac{\sum_i \mathbb{I}(\hat{y}_i(\tau) = 1 \wedge y_i = 1)}{\sum_i \mathbb{I}(y_i = 1)}
\end{aligned} \qquad (15)$$

The F1-score at threshold $\tau$ is defined as follows.

$$\text{F1}(\tau) = \frac{2 \cdot \text{Prec}(\tau) \cdot \text{Rec}(\tau)}{\text{Prec}(\tau) + \text{Rec}(\tau)} \qquad (16)$$

The Max F1-score is defined as follows.

$$\text{MaxF1} = \max_{\tau} \text{F1}(\tau) \qquad (17)$$

### A.14. Detailed Experimental Results

Table A.13 reports the detailed performance of `MetaEns` and all baselines on each of the 39 benchmark datasets. For each dataset, we report Average Precision (AP) scores computed under the same evaluation protocol described in Sec. 4, ensuring a consistent comparison across methods. This table complements the aggregated results in the main paper by providing a per-dataset view of method behavior, enabling fine-grained comparison and reproducibility analysis.

### A.15. Pool Size Analysis

We study the performance and robustness of `MetaEns` by varying the size of the candidate model pool from 10 to 297 detectors. For each pool size, we randomly sample the specified number of models and apply the full `MetaEns` selection procedure. To account for variability introduced by sub-pool sampling, experiments are repeated with 10 random seeds and results are averaged.

Fig. A.1 shows the relationship between pool size and final ensemble performance measured by AP. The blue curve

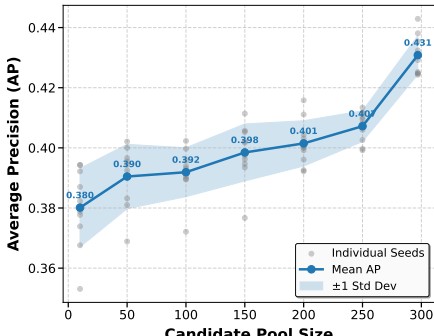

*Figure A.1.* Impact of candidate pool size on `MetaEns` performance. Results are averaged over 10 random seeds for each pool size. The shaded area represents $\pm 1$ standard deviation.

denotes the average AP across the 10 runs, and the shaded region indicates $\pm 1$ standard deviation.

Performance improves steadily as the pool size increases, indicating that `MetaEns` effectively leverages a richer set of candidate models. Gains begin to plateau around 250 models, suggesting diminishing returns beyond this point. This trend indicates that the full pool of 297 models is sufficient to capture the diversity needed for robust anomaly detection across the benchmark datasets.

To separate algorithm diversity from hyperparameter selection, we further run a single-family pool analysis. In this setting, `MetaEns` may still select and combine multiple hyperparameter variants, but all candidates come from the same algorithmic family. Table A.14 shows that performance falls substantially for IForest-only and LOF-only pools, while the average ensemble size remains close to the full setting. Thus, the loss is mainly due to missing cross-family candidates rather than an inability to form ensembles, indicating that algorithm diversity is the primary driver and hyperparameter selection provides complementary gains.

### A.16. Statistical Significance Analysis

As discussed in Sec. 4.4, we use paired Wilcoxon signed-rank tests to compare `MetaEns` against each baseline across the 39 benchmark datasets. The test is non-parametric and is therefore suitable for paired comparisons across heterogeneous datasets without assuming normally distributed AP differences. We use a one-sided alternative hypothesis that `MetaEns` achieves higher AP than the baseline and set $\alpha = 0.05$. Table A.15 reports the corresponding $p$-values, mean AP differences, and dataset-level win rates.

### A.17. Modality Transfer Experiments

We further evaluate whether `MetaEns` can be used as a plug-and-play selector outside tabular anomaly detection. Since the state representation depends only

*Table A.13.* Detailed Performance Comparison Across 39 Benchmark Datasets. For each method, we report Average Precision (AP) and rank in parentheses (lower rank is better, 1–19, among compared methods). Best results in **bold**. Per-dataset values are from a single representative seed (seed=42); the 3-seed averaged AP for `MetaEns` is **0.4308** (Table 1). Abbreviations: `Sgl` = `Random Selection`, RE = `Random Ensemble` (k=3), ELECT = ELECT (Top-1), ELECT-10 = ELECT (Top-10).

| Dataset | Sgl | IForest | LOF | GB | ME | RE | RDA | DAGMM | DeepSVDD | RandNet | ROBOD | LUNAR | DTE-C | TCCM | LSCP | MetaOD | ELECT-1 | ELECT-10 | MetaEns |
|---|---|---|---|---|---|---|---|---|---|---|---|---|---|---|---|---|---|---|---|
| ALOI | 0.039 (11) | 0.034 (16) | 0.074 (6) | 0.032 (19) | 0.036 (14) | 0.035 (15) | 0.039 (10) | 0.038 (12) | 0.049 (7) | 0.040 (9) | 0.042 (8) | 0.142 (4) | 0.033 (18) | 0.038 (13) | 0.077 (5) | 0.033 (17) | 0.164 (2) | 0.158 (3) | **0.164 (1)** |
| Annthyroid | 0.104 (12) | 0.113 (10) | 0.129 (8) | 0.139 (6) | 0.135 (7) | 0.129 (9) | 0.083 (17) | 0.090 (15) | 0.109 (11) | 0.066 (19) | 0.068 (18) | 0.093 (14) | 0.102 (13) | 0.171 (4) | 0.083 (16) | 0.155 (5) | 0.197 (3) | 0.218 (2) | **0.223 (1)** |
| Arrhythmia | 0.675 (14) | 0.765 (2) | 0.755 (6) | 0.751 (7) | 0.748 (9) | 0.739 (12) | 0.669 (15) | 0.637 (16) | 0.501 (19) | 0.747 (10) | 0.737 (13) | 0.750 (8) | 0.607 (17) | 0.591 (18) | 0.746 (11) | **0.768 (1)** | 0.761 (4) | 0.764 (3) | 0.757 (5) |
| Cardiotocography | 0.412 (8) | 0.437 (6) | 0.302 (16) | 0.473 (3) | 0.408 (9) | 0.398 (10) | 0.264 (18) | 0.362 (13) | 0.308 (15) | **0.531 (1)** | 0.520 (2) | 0.222 (19) | 0.270 (17) | 0.315 (14) | 0.392 (11) | 0.442 (5) | 0.429 (7) | 0.444 (4) | 0.384 (12) |
| Glass | 0.134 (11) | 0.153 (8) | 0.092 (19) | 0.197 (5) | 0.120 (14) | 0.136 (10) | 0.192 (6) | 0.121 (12) | 0.107 (17) | 0.117 (16) | 0.118 (15) | 0.172 (7) | 0.144 (9) | 0.120 (13) | 0.213 (2) | **0.253 (1)** | 0.209 (4) | 0.212 (3) | 0.102 (18) |
| HeartDisease | **0.598 (1)** | 0.541 (7) | 0.574 (2) | 0.525 (11) | 0.571 (3) | 0.566 (4) | 0.458 (16) | 0.441 (18) | 0.546 (6) | 0.536 (9) | 0.452 (17) | 0.514 (13) | 0.482 (14) | 0.420 (19) | 0.480 (15) | 0.523 (12) | 0.538 (8) | 0.532 (10) | 0.562 (5) |
| InternetAds | 0.331 (15) | 0.527 (5) | 0.366 (14) | 0.455 (10) | 0.482 (9) | 0.405 (11) | 0.296 (16) | 0.277 (18) | 0.202 (19) | 0.525 (6) | 0.503 (8) | 0.371 (13) | 0.295 (17) | 0.373 (12) | 0.576 (2) | 0.525 (7) | 0.534 (4) | 0.536 (3) | **0.579 (1)** |
| PageBlocks | 0.367 (16) | 0.465 (11) | 0.531 (5) | 0.409 (13) | 0.385 (15) | 0.390 (14) | 0.436 (12) | 0.262 (17) | 0.227 (18) | 0.551 (4) | 0.483 (7) | 0.192 (19) | 0.568 (2) | **0.575 (1)** | 0.564 (3) | 0.467 (9) | 0.480 (8) | 0.466 (10) | 0.494 (6) |
| PenDigits | 0.006 (13) | 0.005 (15) | 0.019 (5) | 0.006 (14) | 0.007 (11) | 0.007 (10) | 0.020 (4) | 0.014 (7) | **0.068 (1)** | 0.002 (19) | 0.002 (18) | 0.038 (2) | 0.013 (8) | 0.009 (9) | 0.015 (6) | 0.005 (17) | 0.007 (12) | 0.005 (16) | 0.021 (3) |
| Pima | 0.466 (11) | 0.516 (2) | 0.514 (3) | 0.436 (16) | 0.500 (6) | 0.488 (9) | 0.451 (13) | 0.409 (19) | 0.451 (12) | 0.410 (18) | 0.422 (17) | **0.527 (1)** | 0.438 (14) | 0.437 (15) | 0.474 (10) | 0.499 (7) | 0.509 (4) | 0.507 (5) | 0.491 (8) |
| Shuttle | 0.129 (7) | 0.069 (16) | 0.355 (4) | 0.090 (11) | 0.117 (8) | 0.114 (9) | 0.040 (17) | 0.175 (5) | 0.388 (3) | 0.022 (19) | 0.025 (18) | 0.160 (6) | **0.512 (1)** | 0.112 (10) | 0.084 (13) | 0.071 (15) | 0.090 (12) | 0.079 (14) | 0.396 (2) |
| SpamBase | 0.476 (9) | 0.480 (7) | 0.355 (18) | 0.533 (4) | 0.483 (6) | 0.439 (10) | 0.399 (14) | 0.349 (19) | 0.367 (17) | 0.404 (13) | 0.421 (12) | 0.377 (16) | 0.392 (15) | 0.431 (11) | 0.494 (5) | 0.479 (8) | 0.559 (2) | 0.557 (3) | **0.594 (1)** |
| Stamps | 0.249 (11) | 0.307 (8) | 0.333 (4) | 0.333 (5) | 0.333 (3) | 0.333 (2) | 0.256 (10) | 0.196 (13) | 0.113 (19) | 0.164 (16) | 0.123 (18) | 0.146 (17) | 0.174 (15) | 0.213 (12) | 0.189 (14) | 0.334 (2) | **0.345 (1)** | 0.309 (7) | 0.269 (9) |
| WBC | 0.556 (12) | 0.882 (3) | 0.875 (5) | 0.827 (9) | 0.864 (8) | 0.604 (11) | 0.274 (18) | 0.537 (13) | 0.368 (17) | 0.394 (16) | 0.501 (14) | 0.750 (10) | 0.111 (19) | 0.400 (15) | **0.895 (1)** | 0.877 (4) | 0.874 (6) | 0.885 (2) | 0.865 (7) |
| WDBC | 0.647 (11) | 0.647 (10) | 0.691 (7) | 0.716 (4) | 0.697 (5) | 0.684 (8) | 0.092 (19) | 0.551 (13) | 0.316 (16) | 0.630 (12) | 0.501 (15) | 0.512 (14) | 0.197 (18) | 0.275 (17) | **0.800 (1)** | 0.678 (9) | 0.760 (3) | 0.694 (6) | 0.768 (2) |
| WPBC | 0.233 (7) | 0.231 (9) | 0.232 (8) | 0.239 (5) | 0.231 (10) | 0.230 (11) | 0.243 (4) | 0.206 (19) | 0.266 (2) | 0.222 (17) | 0.207 (18) | 0.227 (14) | **0.275 (1)** | 0.216 (17) | 0.265 (3) | 0.227 (15) | 0.229 (13) | 0.225 (16) | 0.235 (6) |
| Waveform | 0.078 (8) | 0.061 (12) | 0.131 (4) | 0.055 (16) | 0.066 (9) | 0.081 (7) | 0.029 (19) | 0.033 (18) | 0.056 (15) | 0.058 (13) | 0.066 (10) | 0.141 (3) | 0.035 (17) | 0.063 (11) | 0.168 (2) | 0.056 (14) | 0.115 (6) | 0.118 (5) | **0.190 (1)** |
| Wilt | 0.048 (13) | 0.045 (17) | 0.053 (8) | 0.041 (19) | 0.043 (18) | 0.055 (6) | **0.283 (1)** | 0.089 (3) | 0.046 (14) | 0.054 (7) | 0.052 (9) | 0.083 (4) | 0.047 (15) | 0.049 (10) | 0.079 (5) | 0.045 (16) | 0.048 (12) | 0.046 (15) | 0.048 (11) |
| annthyroid | 0.195 (13) | 0.314 (7) | 0.204 (12) | 0.366 (5) | 0.285 (8) | 0.247 (10) | 0.161 (15) | 0.128 (17) | 0.120 (18) | 0.135 (16) | 0.114 (19) | 0.173 (14) | **0.655 (1)** | 0.243 (11) | 0.257 (9) | 0.336 (6) | 0.452 (3) | 0.395 (4) | 0.531 (2) |
| arrhythmia | 0.424 (14) | 0.479 (4) | 0.464 (7) | **0.502 (1)** | 0.445 (11) | 0.431 (13) | 0.313 (15) | 0.257 (19) | 0.309 (16) | 0.460 (8) | 0.450 (10) | 0.474 (5) | 0.291 (17) | 0.267 (18) | 0.459 (9) | 0.482 (2) | 0.474 (6) | 0.480 (3) | 0.475 (5) |
| breastw | 0.895 (9) | 0.969 (4) | 0.392 (18) | 0.967 (5) | 0.979 (2) | 0.941 (8) | 0.883 (10) | 0.782 (12) | 0.320 (19) | 0.686 (15) | 0.728 (14) | 0.748 (13) | 0.786 (11) | 0.569 (17) | 0.588 (16) | **0.979 (1)** | 0.955 (6) | 0.955 (7) | 0.972 (3) |
| glass | 0.076 (16) | 0.093 (10) | 0.083 (13) | 0.116 (7) | 0.085 (12) | 0.080 (14) | 0.195 (2) | 0.052 (18) | 0.041 (19) | **0.205 (1)** | 0.154 (5) | 0.171 (4) | 0.172 (3) | 0.078 (15) | 0.125 (6) | 0.116 (8) | 0.092 (11) | 0.104 (9) | 0.058 (17) |
| ionosphere | 0.636 (17) | 0.809 (6) | 0.799 (9) | 0.758 (13) | 0.803 (8) | 0.745 (14) | **0.949 (1)** | 0.599 (18) | 0.565 (19) | 0.735 (16) | 0.742 (15) | 0.914 (3) | 0.916 (2) | 0.809 (7) | 0.790 (11) | 0.910 (4) | 0.793 (10) | 0.813 (5) | 0.759 (12) |
| letter | 0.140 (7) | 0.087 (15) | 0.244 (4) | 0.088 (12) | 0.130 (8) | 0.220 (5) | 0.311 (2) | 0.094 (10) | 0.081 (19) | 0.100 (9) | 0.085 (16) | **0.383 (1)** | 0.269 (3) | 0.088 (14) | 0.209 (6) | 0.091 (11) | 0.085 (17) | 0.088 (13) | 0.082 (18) |
| lympho | 0.782 (10) | 0.944 (3) | 0.857 (6) | 0.976 (2) | 0.877 (5) | 0.822 (8) | 0.151 (18) | 0.519 (14) | 0.056 (19) | 0.897 (4) | 0.808 (9) | 0.640 (12) | 0.323 (16) | 0.403 (15) | 0.173 (17) | **1.000 (1)** | 0.720 (11) | 0.852 (7) | 0.556 (13) |
| mammography | 0.184 (9) | 0.221 (6) | 0.121 (12) | 0.221 (7) | 0.247 (3) | 0.198 (8) | 0.171 (11) | 0.100 (15) | 0.070 (19) | 0.073 (18) | 0.097 (16) | 0.105 (13) | 0.176 (10) | 0.102 (14) | 0.082 (17) | 0.238 (4) | 0.269 (2) | 0.235 (5) | **0.284 (1)** |
| mnist | 0.246 (16) | 0.265 (13) | 0.379 (5) | 0.262 (14) | 0.278 (12) | 0.290 (9) | 0.393 (3) | 0.225 (17) | 0.257 (19) | 0.309 (16) | 0.350 (7) | 0.450 (10) | **0.442 (1)** | 0.371 (6) | 0.246 (15) | 0.282 (11) | 0.246 (15) | 0.160 (19) | 0.160 (19) |
| musk | 0.548 (13) | **1.000 (1)** | 0.090 (17) | 1.000 (2) | 0.997 (4) | 0.882 (10) | 0.450 (15) | 0.856 (11) | 0.043 (18) | 0.998 (8) | 0.440 (16) | 0.025 (19) | 0.517 (14) | 0.688 (12) | 0.994 (9) | 1.000 (4) | 1.000 (5) | 1.000 (6) | 1.000 (7) |
| optdigits | 0.038 (12) | 0.051 (6) | 0.021 (18) | 0.043 (10) | 0.054 (5) | 0.050 (7) | 0.023 (16) | 0.018 (19) | 0.032 (13) | 0.070 (3) | 0.043 (11) | 0.021 (17) | 0.029 (14) | 0.025 (15) | 0.072 (2) | 0.055 (4) | 0.050 (8) | 0.049 (9) | **0.114 (1)** |
| pendigits | 0.195 (10) | 0.279 (3) | 0.044 (16) | 0.255 (7) | 0.260 (6) | 0.214 (9) | 0.035 (18) | 0.083 (13) | 0.036 (17) | 0.182 (11) | 0.228 (8) | 0.033 (19) | 0.048 (15) | 0.173 (12) | 0.066 (14) | 0.266 (5) | 0.289 (2) | 0.272 (4) | **0.359 (1)** |
| pima | 0.477 (11) | 0.500 (6) | 0.493 (7) | 0.465 (12) | 0.503 (5) | **0.521 (1)** | 0.405 (11) | 0.431 (13) | 0.398 (16) | 0.368 (18) | 0.382 (17) | 0.504 (4) | 0.431 (14) | 0.349 (19) | 0.490 (8) | 0.505 (3) | 0.480 (10) | 0.487 (9) | 0.516 (2) |
| satellite | 0.604 (9) | 0.660 (4) | 0.397 (18) | 0.660 (5) | 0.680 (2) | 0.613 (8) | 0.508 (15) | 0.574 (12) | 0.479 (16) | 0.527 (14) | 0.587 (10) | 0.332 (19) | 0.580 (11) | 0.540 (13) | 0.415 (17) | 0.658 (6) | 0.658 (7) | 0.671 (3) | **0.698 (1)** |
| satimage-2 | 0.709 (11) | 0.926 (4) | 0.142 (16) | 0.915 (7) | 0.944 (2) | 0.754 (10) | 0.100 (17) | 0.394 (12) | 0.063 (18) | **0.961 (1)** | 0.889 (9) | 0.033 (19) | 0.142 (15) | 0.380 (14) | 0.393 (13) | 0.921 (6) | 0.927 (3) | 0.924 (5) | 0.895 (8) |
| speech | 0.027 (3) | 0.018 (18) | 0.020 (11) | 0.027 (5) | 0.021 (10) | 0.028 (2) | 0.019 (13) | 0.023 (9) | 0.027 (4) | 0.019 (14) | 0.019 (12) | 0.024 (8) | 0.018 (17) | **0.038 (1)** | 0.019 (15) | 0.018 (16) | 0.026 (6) | 0.026 (7) | 0.017 (19) |
| thyroid | 0.464 (9) | 0.557 (7) | 0.335 (13) | 0.380 (12) | 0.497 (8) | 0.434 (10) | 0.309 (14) | 0.146 (16) | 0.049 (19) | 0.176 (15) | 0.118 (18) | 0.120 (17) | 0.729 (2) | 0.565 (6) | 0.410 (11) | 0.612 (4) | 0.651 (3) | 0.604 (5) | **0.746 (1)** |
| vertebral | 0.095 (12) | 0.096 (11) | 0.108 (7) | 0.092 (15) | 0.087 (18) | 0.090 (16) | 0.116 (5) | 0.144 (4) | 0.103 (7) | 0.156 (3) | 0.174 (2) | 0.099 (8) | 0.115 (6) | **0.193 (1)** | 0.080 (19) | 0.098 (10) | 0.094 (13) | 0.093 (14) | 0.099 (9) |
| vowels | 0.178 (9) | 0.138 (13) | 0.385 (4) | 0.106 (15) | 0.233 (7) | 0.195 (8) | 0.458 (3) | 0.067 (17) | 0.057 (19) | 0.140 (12) | 0.063 (18) | **0.615 (1)** | 0.362 (6) | 0.089 (16) | 0.369 (5) | 0.127 (14) | 0.155 (11) | 0.165 (10) | 0.491 (2) |
| wbc | 0.612 (6) | 0.608 (7) | 0.650 (3) | 0.582 (11) | 0.597 (8) | 0.616 (5) | 0.241 (18) | 0.415 (16) | 0.447 (14) | 0.596 (9) | 0.417 (15) | 0.378 (17) | 0.203 (19) | 0.472 (13) | 0.542 (12) | 0.591 (10) | 0.639 (4) | **0.668 (1)** | 0.663 (2) |
| wine | 0.246 (6) | 0.213 (12) | 0.290 (2) | 0.234 (8) | 0.255 (5) | 0.274 (3) | 0.093 (17) | 0.112 (15) | 0.081 (18) | 0.220 (9) | 0.108 (16) | 0.239 (7) | 0.256 (4) | 0.123 (14) | 0.000 (19) | 0.211 (13) | 0.215 (10) | 0.215 (11) | **0.366 (1)** |
| **Average** | 0.342 (10.56) | 0.398 (8.31) | 0.330 (9.49) | 0.392 (9.03) | 0.397 (8.00) | 0.369 (9.08) | 0.276 (12.18) | 0.275 (14.18) | 0.208 (14.62) | 0.347 (11.00) | 0.314 (12.82) | 0.302 (10.67) | 0.314 (10.67) | 0.293 (12.08) | 0.348 (9.08) | 0.408 (7.95) | 0.413 (7.08) | 0.414 (7.00) | **0.435 (6.23)** |
| **Std Dev** | 0.246 | 0.308 | 0.248 | 0.302 | 0.301 | 0.268 | 0.223 | 0.227 | 0.171 | 0.285 | 0.255 | 0.242 | 0.231 | 0.209 | 0.267 | 0.313 | 0.294 | 0.297 | 0.285 |

*Table A.14.* Single-family pool analysis. Restricted pools remove cross-family diversity.

| Candidate Pool | AP ↑ | Rank ↓ | Ens. Size |
|---|---|---|---|
| IForest-only (81) | $0.3552 \pm 0.0015$ | N/A | 2.0 |
| LOF-only (36) | $0.3369 \pm 0.0028$ | N/A | 2.0 |
| Full 8-family (297) | **$0.4308 \pm 0.0064$** | **$59.3 \pm 6.96$** | **2.2** |

*Table A.15.* Statistical significance testing: Paired Wilcoxon signed-rank test comparing `MetaEns` against baselines across 39 datasets (one-sided test, $\alpha = 0.05$).

| Ours | Baseline | $p$-value | $\triangle$AP | Win% |
|---|---|---|---|---|
| MetaEns | Singleton | 0.0000 | +0.0939 | 79.5% |
| MetaEns | IForest | 0.0151 | +0.0378 | 64.1% |
| MetaEns | LOF | 0.0022 | +0.1052 | 69.2% |
| MetaEns | Global Best | 0.0050 | +0.0438 | 71.8% |
| MetaEns | Mega Ensemble | 0.0096 | +0.0384 | 66.7% |
| MetaEns | Random Ensemble | 0.0002 | +0.0662 | 74.4% |
| MetaEns | RDA | 0.0001 | +0.1599 | 76.9% |
| MetaEns | DAGMM | 0.0000 | +0.1605 | 84.6% |
| MetaEns | DeepSVDD | 0.0000 | +0.2272 | 84.6% |
| MetaEns | ROBOD | 0.0000 | +0.1173 | 79.5% |
| MetaEns | LUNAR | 0.0004 | +0.1284 | 71.8% |
| MetaEns | DTE-C | 0.0010 | +0.1164 | 71.8% |
| MetaEns | TCCM | 0.0000 | +0.1379 | 76.9% |
| MetaEns | RandNet | 0.0013 | +0.0884 | 71.8% |
| MetaEns | LSCP | 0.0047 | +0.0876 | 66.7% |
| MetaEns | MetaOD | 0.0390 | +0.0275 | 61.5% |
| MetaEns | ELECT-1 | 0.0314 | +0.0222 | 66.7% |
| MetaEns | ELECT-10 | 0.0386 | +0.0213 | 61.5% |

on detector-score vectors, no architectural change is needed for image or text settings. We use ADBench extracted features: `CV_by_ResNet18` (512-dimensional ResNet-18 features) for the 15 MVTec-AD image categories (`bottle`, `cable`, `capsule`, `carpet`, `grid`, `hazelnut`, `leather`, `metal_nut`, `pill`, `screw`, `tile`, `toothbrush`, `transistor`, `wood`, and `zipper`), and `NLP_by_BERT` (768-dimensional BERT features) for the 5 text datasets. The compared methods are `MetaEns`, random singleton selection, mega ensemble, `ELECT-1`, `ELECT-10`, `ROBOD`, `LUNAR`, `DTE-C`, and `TCCM`, all under the same unsupervised evaluation protocol.

Table A.16 reports the headline AP results. Across all 20 datasets, `MetaEns` reaches $0.4914 \pm 0.0007$ AP, outperforming the strongest baseline, `LUNAR` ($0.4717 \pm 0.0045$). On MVTec-AD image data, `MetaEns` obtains $0.6230 \pm 0.0009$ AP, improving over `ELECT-1` ($0.5973 \pm 0.0035$). On the 5 text datasets, `LUNAR` slightly edges out `MetaEns` ($0.0977$ vs. $0.0966$ AP), likely because `LUNAR`'s graph-based neighborhood structure is more suited to high-dimensional dense text embeddings; nonetheless, `MetaEns` remains competitive without any modality-specific adaptation. Table A.17 further reports win/loss statistics and one-sided Wilcoxon tests,

confirming significant gains against all compared baselines over the 20 non-tabular datasets.

## A.18. Model Diversity Visualization

As a qualitative complement to Table A.18, we project the 297-model candidate pool into a two-dimensional space using `t-SNE` (van der Maaten & Hinton, 2008) based on prediction-correlation distance. The embedding shows that detectors naturally cluster by algorithmic family, since variants within the same family often tend to produce similar outlier rankings.

Fig. 3 visualizes four representative datasets (**Speech**, **WBC**,

*Table A.16.* Modality-transfer results on 20 non-tabular datasets.

| Setting | #Data | MetaEns AP ↑ | Best Base. AP ↑ | △AP |
|---|---|---|---|---|
| Overall | 20 | **0.4914 ± 0.0007** | LUNAR: 0.4717 ± 0.0045 | +0.0197 |
| Image (MVTec-AD) | 15 | **0.6230 ± 0.0009** | ELECT-1: 0.5973 ± 0.0035 | +0.0257 |
| Text | 5 | 0.0966 ± 0.0014 | LUNAR: 0.0977 ± 0.0021 | −0.0011 |

*Table A.17.* Dataset-level win/loss and one-sided Wilcoxon signed-rank tests for modality-transfer experiments over 20 non-tabular datasets.

| Baseline | Win/Loss | $p$-value |
|---|---|---|
| ELECT-1 | 14 / 6 | 0.0148 |
| ELECT-10 | 17 / 3 | 0.0047 |
| LUNAR | 15 / 5 | 0.0266 |
| ROBOD | 18 / 2 | $1.61 \times 10^{-4}$ |
| DTE-C | 18 / 2 | $6.7 \times 10^{-5}$ |
| TCCM | 18 / 2 | $1.05 \times 10^{-4}$ |
| Mega Ensemble | 18 / 2 | $5.1 \times 10^{-4}$ |
| Singleton | 19 / 1 | $6.7 \times 10^{-5}$ |

**Waveform**, and **Shuttle**). ELECT-10 selections often concentrate within a single family cluster, whereas MetaEns selects models spanning multiple clusters. This supports the quantitative finding that MetaEns constructs compact but complementary ensembles rather than simply adding many redundant high-ranked detectors.

### A.19. Effect of Expanded Model Pools

To evaluate MetaEns's scalability and robustness to model pool composition, we conduct additional experiments with an expanded pool of 310 models, incorporating neural network variants alongside the original 297 classical detectors.

**Expanded Pool Composition.** The expanded pool adds 13 neural network variants spanning 5 deep learning families: AutoEncoder (Chen et al., 2017) (4 architectural variants), Variational AutoEncoder (Xu et al., 2018) (3 variants), SO_GAAL (Liu et al., 2020) (2 configurations), MO_GAAL (Liu et al., 2020) (2 configurations), and DeepSVDD (Ruff et al., 2018) (2 variants). These models represent diverse deep learning paradigms, including reconstruction-based methods, adversarial approaches, and deep one-class classification. Table A.19 provides the complete specification of the 310-model expanded pool.

**Overall Performance Comparison.** Table A.20 presents the scalability analysis results comparing MetaEns performance on the original (297 models) versus expanded (310 models) pools. MetaEns demonstrates robust performance across both pool configurations, maintaining competitive efficacy with only a modest decrease in Average Precision (0.0186) and an increase in AP rank (13.0).

**Detailed Dataset-by-Dataset Results.** Table A.21 presents the complete performance comparison of all methods on the expanded 310-model pool. The table shows

*Table A.18.* Quantitative diversity of selected ensembles on the 39 tabular datasets. **Distinct Families** counts unique base algorithms in the ensemble; **Pairwise Jaccard** is the mean top-10% overlap between selected detectors (lower is more diverse); **Ens Size** is the average number of selected models. Mega Ensemble is included as a reference upper bound on family coverage since it uses all 297 detectors. Bold values indicate the best among selector-based methods (ELECT (Top-10) and MetaEns).

| Method | Distinct Families ↑ | Pairwise Jaccard ↓ | Ens Size |
|---|---|---|---|
| ELECT (Top-10) | 2.1 | 0.68 | 10 |
| Mega Ensemble (ref.) | 8.0 | 0.44 | 297 |
| MetaEns (Ours) | **2.2** | **0.36** | 2.2 |

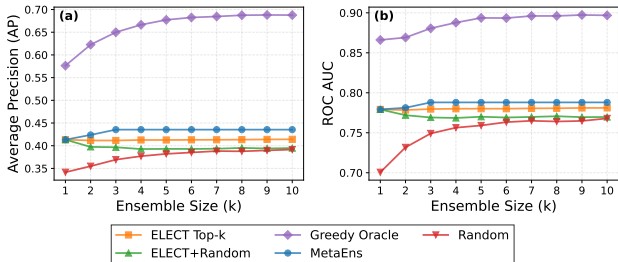

*Figure A.2.* Ensemble size impact on performance (AP) and robustness (ROC AUC). **(Left)** Average Precision vs ensemble size shows that MetaEns achieves rapid performance gains with small ensembles ($k \leq 3$), then plateaus, confirming the effectiveness of the family-risk regularizer in adaptive ensemble stopping. **(Right)** ROC AUC demonstrates consistent robustness across different ensemble sizes. Greedy Oracle (with oracle access to test labels) provides an upper bound, while Random selection serves as the lower bound. ELECT performs comparably to single models ($k = 1$) across ensemble sizes due to its ranking-based selection strategy.

Average Precision (AP) values and model ranks (1–310, where lower ranks indicate better performance) for each method across all 39 benchmark datasets, with best results for each dataset highlighted in bold.

**Results Analysis.** Beyond the scalability numbers above, two additional patterns emerge:

- **Discriminative Selection:** Despite several neural network models achieving competitive individual performance, MetaEns predominantly selects classical detectors, indicating that its meta-learned policy exploits cross-dataset structure rather than per-model strength.

- **Competitive Performance:** MetaEns achieves the best performance on 10 out of 39 datasets (25.6%) on the expanded pool, consistent with its competitive behavior on the original pool.

### A.20. Ensemble Size Impact Analysis

To understand how ensemble size affects performance, we analyze the relationship between the number of models and

*Table A.19.* Expanded candidate model pool specification with neural network variants ($M = 310$ total models).

| Family | Count | Hyperparameter Grid |
|---|---|---|
| kNN (Chehreghani, 2016) | 36 | method $\in$ {largest, mean, median}; $k \in \{1, 5, 10, 15, 20, 25, 50, 60, 70, 80, 90, 100\}$ |
| LOF (Breunig et al., 2000) | 36 | metric $\in$ {euclidean, manhattan, minkowski}; $k \in \{1, 5, 10, 15, 20, 25, 50, 60, 70, 80, 90, 100\}$ |
| IForest (Liu et al., 2008) | 81 | $n_{\text{estimators}} \in \{10, 20, 30, 40, 50, 75, 100, 150, 200\}$; max_samples $\in \{0.1, 0.2, \ldots, 0.9\}$ |
| HBOS (Goldstein & Dengel, 2012) | 40 | $n_{\text{bins}} \in \{5, 10, 20, 30, 40, 50, 75, 100\}$; tolerance $\in \{0.1, 0.2, 0.3, 0.4, 0.5\}$ |
| OCSVM (Schölkopf et al., 1999) | 36 | kernel $\in$ {linear, poly, rbf, sigmoid}; $\nu \in \{0.1, 0.2, \ldots, 0.9\}$ |
| LODA (Pevný, 2016) | 54 | $n_{\text{bins}} \in \{5, 10, 15, 20, 25, 30\}$; $n_{\text{cuts}} \in \{10, 20, 30, 40, 50, 75, 100, 150, 200\}$ |
| ABOD (Kriegel et al., 2008) | 7 | $n_{\text{neighbors}} \in \{3, 5, 10, 15, 20, 25, 50\}$ |
| COF (Tang et al., 2002) | 7 | $n_{\text{neighbors}} \in \{3, 5, 10, 15, 20, 25, 50\}$ |
| **Neural Network Extensions** | | |
| Autoencoder (Chen et al., 2017) | 4 | Architecture variants: [64,32,32,64], [128,64,64,128], [32,16,16,32], [64,32,16,16,32,64] |
| Variational Autoencoder (Xu et al., 2018) | 3 | Encoder-decoder pairs: ([64,32],[32,64]), ([128,64],[64,128]), ([32,16],[16,32]) |
| SO_GAAL (Liu et al., 2020) | 2 | Single-objective GAN: stop_epochs=20; lr $\in \{0.0005, 0.001\}$ |
| MO_GAAL (Liu et al., 2020) | 2 | Multi-objective GAN: stop_epochs=20; $k \in \{5, 20\}$ |
| DeepSVDD (Ruff et al., 2018) | 2 | Deep SVDD: pre-training $\in$ {True, False} |
| **Classical Total** | **297** | Traditional anomaly detection methods |
| **Neural Network Total** | **13** | Deep learning-based detection methods |
| **Grand Total** | **310** | Complete expanded model pool |

detection quality. Fig A.2 presents the Average Precision (AP) and ROC AUC as a function of ensemble size for several representative methods.

A key finding from our experiments is that `MetaEns` employs adaptive ensemble sizing, selecting an average of 2.2 models per dataset across the 39-dataset benchmark. This adaptive behavior contrasts sharply with fixed-size baselines: ELECT Top-10 always uses 10 models, while Random Ensemble uses a fixed best $k = 3$. The ability to adjust ensemble size based on dataset characteristics is a crucial advantage of our approach.

The performance comparison in Table 1 reveals important insights about ensemble construction strategies. ELECT Top-10, despite using 10 models, achieves only 0.4117 AP—a marginal 0.0048 improvement over ELECT (Top-1) with 0.4069 AP. This negligible gain from expanding to 10 models suggests that simple rank-based aggregation without diversity consideration leads to redundant model selection. In contrast, `MetaEns` achieves 0.4308 AP with an average of only 2.2 models, demonstrating that carefully selected small

ensembles can substantially outperform large ensembles of top-ranked models.

The Random Ensemble baseline with best $k = 3$ achieves 0.3759 AP, performing worse than both the single ELECT selector (0.4069 AP) and `MetaEns` (0.4308 AP). This indicates that expanding ensembles with randomly chosen partners actively degrades performance by introducing low-quality models that add noise to the ensemble prediction. This finding validates our hypothesis that partner selection must be guided by both quality and diversity considerations.

Our adaptive stopping mechanism enables dataset-specific ensemble construction: simple datasets benefit from smaller ensembles, while more complex ones may require additional models for complementary coverage. Fixed-size baselines cannot capture this heterogeneity.

*Table A.20.* `MetaEns` scalability analysis: Performance on original (297) vs. expanded (310) model pools. Results demonstrate method robustness across different pool compositions.

| Pool Configuration | Size | AP (Mean ± Std) ↑ | Rank ↓ | ROC-AUC ↑ | Ens Size |
|---|---|---|---|---|---|
| Classical Models Only | 297 | 0.4308 ± 0.0064 | 59.3 ± 6.9610 | 0.7867 ± 0.0045 | 2.2 |
| Classical + Neural Networks | 310 | 0.4122 ± 0.0068 | 72.3 ± 11.4018 | 0.7823 ± 0.0044 | 2.3 |

*Table A.21.* Detailed Performance Comparison Across 39 Benchmark Datasets on Expanded Neural Network Pool (310 models). For each method, we report Average Precision (AP) and rank in parentheses (lower rank is better, 1–19, among compared methods). Best results in **bold**. Per-dataset values are from a single representative seed (seed=42). Abbreviations: `Sgl` = `Singleton` (random single-model selection), `RE` = `Random Ensemble` ($k = 3$), `ELECT-1` = `ELECT` (Top-1), `ELECT-10` = `ELECT` (Top-10).

| Dataset | Sgl | IForest | LOF | GB | ME | RE | RDA | DAGMM | DeepSVDD | RandNet | ROBOD | LUNAR | DTE-C | TCCM | LSCP | MetaOD | ELECT-1 | ELECT-10 | MetaEns |
|---|---|---|---|---|---|---|---|---|---|---|---|---|---|---|---|---|---|---|---|
| ALOI | 0.039 (10) | 0.034 (16) | 0.074 (6) | 0.032 (19) | 0.036 (14) | 0.035 (15) | 0.039 (11) | 0.038 (12) | 0.049 (7) | 0.040 (9) | 0.042 (8) | 0.142 (4) | 0.033 (18) | 0.038 (13) | 0.077 (5) | 0.033 (17) | **0.164 (1)** | 0.158 (3) | 0.164 (2) |
| Annthyroid | 0.104 (12) | 0.113 (10) | 0.129 (8) | 0.139 (6) | 0.136 (7) | 0.129 (9) | 0.083 (16) | 0.090 (15) | 0.109 (11) | 0.066 (19) | 0.068 (18) | 0.093 (14) | 0.102 (13) | 0.171 (4) | 0.083 (17) | 0.155 (5) | 0.197 (2) | **0.218 (1)** | 0.197 (3) |
| Arrhythmia | 0.675 (14) | 0.765 (2) | 0.755 (5) | 0.751 (6) | 0.746 (9) | 0.737 (11) | 0.669 (15) | 0.637 (16) | 0.501 (19) | 0.747 (8) | 0.737 (12) | 0.750 (7) | 0.607 (17) | 0.591 (18) | 0.746 (10) | **0.768 (1)** | 0.761 (4) | 0.764 (3) | 0.726 (13) |
| Cardiotocography | 0.412 (8) | 0.437 (6) | 0.302 (16) | 0.473 (3) | 0.410 (9) | 0.400 (10) | 0.264 (18) | 0.362 (13) | 0.308 (15) | **0.531 (1)** | 0.520 (2) | 0.222 (19) | 0.270 (17) | 0.315 (14) | 0.392 (11) | 0.442 (5) | 0.429 (7) | 0.444 (4) | 0.384 (12) |
| Glass | 0.134 (10) | 0.153 (8) | 0.092 (19) | 0.197 (5) | 0.119 (14) | 0.131 (11) | 0.192 (6) | 0.121 (12) | 0.107 (17) | 0.117 (16) | 0.118 (15) | 0.172 (7) | 0.144 (9) | 0.120 (13) | 0.213 (2) | **0.253 (1)** | 0.209 (4) | 0.212 (3) | 0.102 (18) |
| HeartDisease | **0.598 (1)** | 0.541 (7) | 0.574 (2) | 0.525 (11) | 0.567 (3) | 0.566 (4) | 0.458 (16) | 0.441 (18) | 0.546 (6) | 0.536 (9) | 0.452 (17) | 0.514 (13) | 0.482 (14) | 0.420 (19) | 0.480 (15) | 0.523 (12) | 0.538 (8) | 0.532 (10) | 0.562 (5) |
| InternetAds | 0.339 (15) | 0.527 (5) | 0.366 (14) | 0.455 (10) | 0.476 (9) | 0.390 (11) | 0.296 (16) | 0.277 (18) | 0.202 (19) | 0.525 (6) | 0.503 (8) | 0.371 (13) | 0.295 (17) | 0.373 (12) | 0.576 (2) | 0.525 (7) | 0.534 (4) | 0.536 (3) | **0.579 (1)** |
| PageBlocks | 0.325 (16) | 0.465 (11) | 0.531 (5) | 0.409 (13) | 0.393 (14) | 0.368 (15) | 0.436 (12) | 0.262 (17) | 0.227 (18) | 0.551 (4) | 0.483 (7) | 0.192 (19) | 0.568 (2) | **0.575 (1)** | 0.564 (3) | 0.467 (9) | 0.480 (8) | 0.466 (10) | 0.494 (6) |
| PenDigits | 0.006 (13) | 0.005 (15) | 0.019 (5) | 0.006 (14) | 0.007 (10) | 0.007 (11) | 0.020 (4) | 0.014 (7) | **0.068 (1)** | 0.002 (19) | 0.002 (18) | 0.038 (2) | 0.013 (8) | 0.009 (9) | 0.015 (6) | 0.005 (16) | 0.007 (12) | 0.005 (17) | 0.021 (3) |
| Pima | 0.466 (11) | 0.516 (2) | 0.514 (3) | 0.436 (16) | 0.497 (8) | 0.488 (9) | 0.451 (12) | 0.409 (19) | 0.451 (13) | 0.410 (18) | 0.422 (17) | **0.527 (1)** | 0.438 (14) | 0.437 (15) | 0.474 (10) | 0.499 (7) | 0.509 (5) | 0.507 (6) | 0.512 (4) |
| Shuttle | 0.129 (7) | 0.069 (16) | 0.355 (4) | 0.090 (11) | 0.119 (8) | 0.114 (9) | 0.040 (17) | 0.175 (5) | 0.388 (3) | 0.022 (19) | 0.025 (18) | 0.160 (6) | **0.512 (1)** | 0.112 (10) | 0.084 (13) | 0.071 (15) | 0.090 (12) | 0.079 (14) | 0.436 (2) |
| SpamBase | 0.476 (9) | 0.480 (6) | 0.355 (18) | 0.533 (3) | 0.479 (7) | 0.439 (10) | 0.399 (14) | 0.349 (19) | 0.367 (17) | 0.404 (13) | 0.421 (12) | 0.377 (16) | 0.392 (15) | 0.431 (11) | 0.494 (5) | 0.499 (7) | 0.509 (5) | 0.507 (6) | 0.530 (4) |
| Stamps | 0.267 (10) | 0.307 (8) | 0.333 (3) | 0.333 (4) | 0.332 (5) | 0.265 (11) | 0.196 (13) | 0.113 (19) | 0.164 (16) | 0.123 (18) | 0.146 (17) | 0.174 (15) | 0.213 (12) | 0.189 (14) | 0.334 (2) | **0.345 (1)** | 0.309 (7) | 0.327 (6) | 0.269 (9) |
| WBC | 0.483 (14) | 0.882 (3) | 0.875 (5) | 0.827 (9) | 0.858 (8) | 0.605 (11) | 0.274 (18) | 0.537 (12) | 0.368 (17) | 0.394 (16) | 0.501 (13) | 0.750 (10) | 0.111 (19) | 0.400 (15) | **0.895 (1)** | 0.877 (4) | 0.874 (7) | 0.885 (2) | 0.875 (6) |
| WDBC | 0.647 (10) | 0.647 (11) | 0.691 (6) | 0.716 (4) | 0.686 (7) | 0.684 (8) | 0.092 (19) | 0.551 (13) | 0.316 (16) | 0.630 (12) | 0.501 (15) | 0.512 (14) | 0.197 (18) | 0.275 (17) | **0.800 (1)** | 0.678 (9) | 0.760 (3) | 0.694 (5) | 0.768 (2) |
| WPBC | 0.233 (7) | 0.231 (9) | 0.232 (8) | 0.239 (5) | 0.230 (10) | 0.229 (11) | 0.243 (4) | 0.206 (19) | 0.266 (2) | 0.229 (12) | 0.207 (18) | 0.227 (14) | **0.275 (1)** | 0.216 (17) | 0.265 (3) | 0.227 (15) | 0.229 (13) | 0.225 (16) | 0.235 (6) |
| Waveform | 0.078 (8) | 0.061 (12) | 0.131 (4) | 0.055 (16) | 0.067 (9) | 0.080 (7) | 0.029 (19) | 0.033 (18) | 0.056 (14) | 0.058 (13) | 0.066 (10) | 0.141 (3) | 0.035 (17) | 0.063 (11) | **0.168 (1)** | 0.056 (15) | 0.115 (6) | 0.118 (5) | 0.166 (2) |
| Wilt | 0.048 (11) | 0.045 (16) | 0.053 (8) | 0.041 (19) | 0.044 (18) | 0.055 (6) | **0.283 (1)** | 0.089 (3) | 0.046 (14) | 0.054 (7) | 0.052 (9) | 0.083 (4) | 0.147 (2) | 0.049 (10) | 0.079 (5) | 0.045 (17) | 0.048 (12) | 0.046 (15) | 0.048 (13) |
| annthyroid | 0.195 (13) | 0.314 (7) | 0.204 (12) | 0.366 (5) | 0.290 (8) | 0.247 (10) | 0.161 (15) | 0.128 (17) | 0.120 (18) | 0.135 (16) | 0.114 (19) | 0.173 (14) | **0.655 (1)** | 0.243 (11) | 0.257 (9) | 0.336 (6) | 0.452 (3) | 0.395 (4) | 0.568 (2) |
| arrhythmia | 0.424 (14) | 0.479 (4) | 0.464 (7) | **0.502 (1)** | 0.444 (11) | 0.431 (13) | 0.313 (15) | 0.257 (19) | 0.309 (16) | 0.460 (8) | 0.450 (10) | 0.442 (12) | 0.291 (17) | 0.267 (18) | 0.459 (9) | 0.482 (2) | 0.474 (6) | 0.480 (3) | 0.475 (5) |
| breastw | 0.895 (9) | 0.969 (4) | 0.392 (18) | 0.967 (5) | **0.979 (1)** | 0.940 (8) | 0.883 (10) | 0.782 (12) | 0.320 (19) | 0.686 (15) | 0.728 (14) | 0.748 (13) | 0.786 (11) | 0.569 (17) | 0.588 (16) | 0.979 (2) | 0.955 (6) | 0.955 (7) | 0.977 (3) |
| glass | 0.076 (16) | 0.093 (11) | 0.083 (14) | 0.116 (7) | 0.085 (13) | 0.095 (10) | 0.195 (2) | 0.052 (18) | 0.041 (19) | **0.205 (1)** | 0.154 (5) | 0.171 (4) | 0.172 (3) | 0.078 (15) | 0.125 (6) | 0.116 (8) | 0.092 (12) | 0.104 (9) | 0.058 (17) |
| ionosphere | 0.636 (17) | 0.809 (6) | 0.799 (9) | 0.758 (13) | 0.806 (8) | 0.752 (14) | **0.949 (1)** | 0.599 (18) | 0.565 (19) | 0.735 (16) | 0.742 (15) | 0.914 (3) | 0.916 (2) | 0.809 (7) | 0.790 (11) | 0.910 (4) | 0.793 (10) | 0.813 (5) | 0.771 (12) |
| letter | 0.140 (7) | 0.087 (15) | 0.244 (4) | 0.088 (12) | 0.134 (8) | 0.225 (5) | 0.311 (2) | 0.094 (10) | 0.081 (19) | 0.100 (9) | 0.085 (16) | **0.383 (1)** | 0.269 (3) | 0.088 (13) | 0.209 (6) | 0.091 (11) | 0.085 (17) | 0.088 (13) | 0.082 (18) |
| lympho | 0.782 (10) | 0.944 (3) | 0.857 (6) | 0.976 (2) | 0.877 (5) | 0.827 (8) | 0.151 (18) | 0.519 (14) | 0.056 (19) | 0.897 (4) | 0.808 (9) | 0.640 (12) | 0.323 (16) | 0.403 (15) | 0.173 (17) | **1.000 (1)** | 0.720 (11) | 0.852 (7) | 0.556 (13) |
| mammography | 0.184 (9) | 0.221 (6) | 0.121 (12) | 0.221 (7) | 0.252 (3) | 0.198 (8) | 0.171 (11) | 0.100 (15) | 0.070 (19) | 0.073 (18) | 0.097 (16) | 0.105 (13) | 0.176 (10) | 0.102 (14) | 0.082 (17) | 0.238 (4) | 0.269 (2) | 0.235 (5) | **0.284 (1)** |
| mnist | 0.246 (15) | 0.265 (13) | 0.379 (5) | 0.262 (14) | 0.274 (12) | 0.295 (9) | 0.393 (3) | 0.225 (17) | 0.207 (18) | 0.392 (4) | 0.333 (8) | 0.350 (7) | 0.412 (2) | **0.442 (1)** | 0.371 (6) | 0.246 (16) | 0.282 (10) | 0.282 (11) | 0.160 (19) |
| musk | 0.506 (14) | **1.000 (1)** | 0.090 (17) | 0.043 (10) | 1.000 (2) | 1.000 (3) | 0.818 (11) | 0.856 (10) | 0.043 (18) | 0.998 (8) | 0.070 (19) | 0.043 (11) | 0.025 (19) | 0.688 (12) | 0.994 (9) | 0.072 (15) | 0.055 (17) | 0.048 (9) | 0.114 (11) |
| optdigits | 0.037 (12) | 0.051 (6) | 0.071 (17) | 0.043 (10) | 0.053 (5) | 0.051 (7) | 0.023 (16) | 0.018 (18) | 0.032 (13) | 0.036 (11) | 0.182 (11) | 0.228 (8) | 0.033 (18) | 0.048 (15) | 0.173 (12) | 0.066 (14) | **0.289 (1)** | 0.272 (4) | 0.285 (2) |
| pendigits | 0.195 (10) | 0.279 (3) | 0.044 (16) | 0.255 (7) | 0.258 (6) | 0.211 (9) | 0.035 (18) | 0.083 (13) | 0.036 (17) | 0.182 (11) | 0.228 (8) | 0.033 (19) | 0.048 (15) | 0.173 (12) | 0.066 (14) | 0.266 (5) | 0.289 (1) | 0.272 (4) | 0.285 (2) |
| pima | 0.474 (11) | 0.500 (6) | 0.493 (7) | 0.465 (12) | 0.501 (5) | 0.520 (2) | 0.405 (15) | 0.431 (14) | 0.398 (16) | 0.368 (18) | 0.382 (17) | 0.504 (4) | 0.431 (13) | 0.349 (19) | 0.490 (8) | 0.505 (3) | 0.480 (10) | 0.487 (9) | **0.522 (1)** |
| satellite | 0.571 (5) | 0.660 (3) | 0.397 (18) | 0.660 (4) | **0.682 (1)** | 0.609 (7) | 0.508 (15) | 0.574 (10) | 0.479 (16) | 0.527 (14) | 0.587 (8) | 0.332 (19) | 0.580 (9) | 0.540 (12) | 0.415 (17) | 0.658 (5) | 0.658 (6) | 0.671 (2) | 0.535 (13) |
| satimage-2 | 0.709 (11) | 0.926 (5) | 0.142 (16) | 0.915 (8) | 0.944 (3) | 0.729 (10) | 0.100 (17) | 0.394 (12) | 0.063 (18) | 0.961 (2) | 0.889 (9) | 0.033 (19) | 0.142 (15) | 0.380 (14) | 0.393 (13) | 0.921 (7) | 0.927 (4) | 0.924 (6) | **0.965 (1)** |
| speech | 0.026 (4) | 0.018 (17) | 0.020 (11) | 0.027 (2) | 0.021 (10) | 0.026 (5) | 0.019 (13) | 0.023 (9) | 0.027 (3) | 0.019 (14) | 0.019 (12) | 0.024 (8) | **0.038 (1)** | 0.018 (16) | 0.019 (15) | 0.026 (6) | 0.018 (18) | 0.026 (7) | 0.017 (19) |
| thyroid | 0.464 (8) | 0.557 (6) | 0.335 (13) | 0.380 (12) | 0.486 (7) | 0.434 (9) | 0.309 (15) | 0.146 (16) | 0.049 (19) | 0.176 (15) | 0.118 (18) | 0.120 (17) | 0.193 (14) | **0.729 (1)** | 0.565 (5) | 0.410 (10) | 0.612 (3) | 0.651 (2) | 0.604 (4) | 0.406 (11) |
| vertebral | 0.095 (12) | 0.096 (11) | 0.088 (17) | 0.092 (15) | 0.087 (18) | 0.090 (16) | 0.116 (5) | 0.144 (4) | 0.103 (7) | 0.156 (3) | 0.174 (2) | 0.099 (9) | 0.115 (6) | **0.193 (1)** | 0.000 (19) | 0.098 (10) | 0.094 (13) | 0.093 (14) | 0.099 (8) |
| vowels | 0.169 (9) | 0.138 (13) | 0.385 (4) | 0.106 (15) | 0.228 (7) | 0.192 (8) | 0.458 (3) | 0.067 (17) | 0.057 (19) | 0.140 (12) | 0.063 (18) | 0.362 (6) | 0.089 (16) | 0.369 (5) | 0.410 (... ) | 0.089 (16) | 0.127 (14) | 0.155 (11) | 0.165 (10) | 0.491 (2) |
| wbc | 0.612 (6) | 0.608 (7) | 0.650 (3) | 0.582 (11) | 0.597 (8) | 0.616 (5) | 0.241 (18) | 0.415 (16) | 0.447 (14) | 0.596 (9) | 0.417 (15) | 0.378 (17) | 0.203 (19) | 0.472 (13) | 0.542 (12) | 0.349 (... ) | 0.612 (... ) | **0.668 (1)** | 0.639 (4) | 0.665 (2) |
| wine | 0.246 (6) | 0.213 (12) | 0.290 (2) | 0.234 (8) | 0.253 (5) | 0.272 (3) | 0.093 (17) | 0.112 (15) | 0.081 (18) | 0.220 (9) | 0.108 (16) | 0.239 (7) | 0.256 (4) | 0.123 (14) | 0.000 (19) | 0.211 (13) | 0.215 (10) | 0.215 (11) | **0.364 (1)** |
| **Average** | 0.337 (10.51) | 0.398 (8.26) | 0.330 (9.41) | 0.392 (8.87) | 0.396 (8.10) | 0.367 (9.13) | 0.276 (12.15) | 0.275 (14.10) | 0.208 (14.59) | 0.347 (10.97) | 0.314 (12.72) | 0.302 (10.69) | 0.314 (10.51) | 0.293 (12.03) | 0.348 (9.03) | 0.408 (8.05) | 0.413 (7.00) | 0.414 (6.97) | **0.422 (6.90)** |
| **Std Dev** | 0.243 | 0.308 | 0.248 | 0.302 | 0.300 | 0.264 | 0.223 | 0.227 | 0.171 | 0.285 | 0.255 | 0.242 | 0.231 | 0.209 | 0.267 | 0.313 | 0.294 | 0.297 | 0.283 |

