# OpenReview forum: "Automatic Unsupervised Ensemble Outlier Model Selection"
_ICML.cc/2026/Conference — ICML 2026 regular_

### Official Review · Reviewer_JU6T · 2026-03-10

**Soundness:** 3
**Presentation:** 4
**Significance:** 3
**Originality:** 2
**Overall Recommendation:** 4
**Confidence:** 4

**Summary:**

This paper studies the problem of unsupervised anomaly detection, in which they aim to select a subset of anomaly detectors from a pool of anomaly detectors to form an ensemble anomaly detector. They assume the availability of a pool of labeled datasets, and based on this they propose a meta-learning based framework. Their framework consists of two main components: (1) an offline meta-training component that employs the knowledge of labelled datasets to train three predictors to predict the gains of adding an anomaly detector to existing ensemble; (2) an online model selection component that builds the ensembles by combining the predicted gains (using the trained predictors) with so-called "redundancy and risk control" mechanism. They perform extensive experiment on 39 datasets, and compare with  more than 10 baselines (most of them are classic methods or old deep anomaly detectors). They also perform ablation studies and many analyses of their meta-learning method.

**Compliance With Llm Reviewing Policy:**

Affirmed.

**Final Justification:**

My concerns are partially addressed and I decided yo raise my overall ratings.

**Key Questions For Authors:**

Please see the weak points for questions. I am willing to raise my overall ratings if my concerns are (partially) addressed.

**Limitations:**

yes

**Strengths And Weaknesses:**

**Strong points**:

(1) This paper is very well organised and presented; I can very easily follow the writing;

(2) The figures are nicely designed, which help me to understand their idea;

(3)  They authors perform many experiments to support most of their claims;

**Weak points**:

(1) Some claims lack (explicit) evidence:

(1.1) for example, in page 1, the authors claim that
>  "Methods employing these strategies suffer from ensemble saturation: beyond a small ensemble size, adding more detectors yields diminishing or even negative returns due to redundancy, conflicting rankings, or systematically poor models. Moreover, fixed-size ensembles are inherently inflexible and cannot adapt to dataset-specific complexity"

but this lacks of references or experiment results to demonstrate these issues;

(1.2) for example, in page 5, the authors claim that
> "...Deep learning baselines show limited effectiveness on these tabular anomaly detection tasks, reflecting the difficulty of learning robust representations from tabular data."

They authors only compared very old deep methods such as RDA (2017), DAGMM (2018), DeepSVDD (2019), why not latest deep methods such as LUNAR (AAAI 2022) [1], DTE (ICLR 2024 )[2] and TCCM (NeurIPS 2025) [3]? These methods show very good results on datasets from ADBench and they are open-sourced.

(2) Some design choices lack of explanations:

(2.1) for instance, in page 2, they indicate that the evaluation criterion estimates the quality of anomaly detection model based ONLY on the properties of the output anomaly scores: why do you this choice? Is there any alternatives? (such as based on the properties of the raw dataset, the embeddings of dataset, the properties of the model, etc.)

(2.2) for instance, in page 3 (Equation 1), the authors define the ensemble score as the mean of the member score rather than median, max or min; Will there be any differences?

(2.3) for instance, in page 3, why using two-part model can mitigate the problem of "positive marginal gains become increasing sparse as the ensemble grows?" Why using ExtraTrees rather than decision trees or other approaches to instantiate $f_{cls}$ and $f_{reg}$?

(2.4) for instance, in page 4 (Equation 8), why it is set as 10-percentile rather than other values?

(3) the UEOMS(Unsupervised Ensemble Outlier Model Selection) problem is not well defined. In page 2, the authors define $P^{*}$ as the subset of models that maximizes the sum of evaluation score of this subset of models. There is not need to use the "sum" notation as the subset should be regarded as a whole.

(4) Some assumptions are strong but not analysed when the assumption is violated. Specifically, the authors assume that
>"the test distribution differs from the distributions underlying the meta-datasets";

what if the distribution of the unseen dataset $\mathbf{X}$ is out-of-distribution from the known $\mathbf{M}$s? Will this framework still work? To what extent will it work?

**Referenes**:

[1] Goodge, Adam, et al. "Lunar: Unifying local outlier detection methods via graph neural networks." Proceedings of the AAAI conference on artificial intelligence. Vol. 36. No. 6. 2022.

[2] Livernoche, Victor, et al. "On Diffusion Modeling for Anomaly Detection." The Twelfth International Conference on Learning Representations. 2024.

[3] Li, Zhong, et al. "Scalable, Explainable and Provably Robust Anomaly Detection with One-Step Flow Matching." The Thirty-ninth Annual Conference on Neural Information Processing Systems. 2025.

---

> ### Author Rebuttal · Authors · 2026-03-31
>
> We thank you for the constructive feedback. Below we address each point and planned manuscript updates.
>
> ### **W1.1: Ensemble saturation and fixed-size inflexibility**
>
> Evidence is in Table 1 and Fig A.2.
>
> **(1) Saturation:** blindly adding models degrades performance. ELECT-10 (10 models) achieves 0.414 AP, but the Mega-Ensemble (all 297 models) drops to 0.397 AP. Forcing excessive models introduces conflicting rankings that harm detection.
>
> **(2) Inflexibility:** fixed-size ensembles cannot adapt to task complexity. Expanding ELECT from Top-1 to Top-10 yields a negligible +0.001 AP gain. MetaEns achieves 0.431 AP by adapting size per dataset (avg. 2.2 models).
>
> **Revision:** elevate this summary and theoretical citations (e.g., Zimek, 2013) to the Introduction.
>
> ### **W1.2: Comparison with latest deep methods**
>
> We evaluated the suggested state-of-the-art models fully unsupervised on 39 tabular datasets (Wilcoxon one-sided, α=0.05):
>
> | Method | AP ↑ | Rank ↓ | p-value | Win% |
> | :--- | :--- | :--- | :--- | :--- |
> | **MetaEns (Ours)** | **0.431** | **59.3** | **—** | **—** |
> | ROBOD | 0.314 | 208.0 | 5.7e-06 | 79.5% |
> | DTE-C | 0.314 | 199.7 | 0.0010 | 71.8% |
> | LUNAR | 0.302 | 172.0 | 0.0004 | 71.8% |
> | TCCM | 0.293 | 188.3 | 2.4e-06 | 76.9% |
>
> Deep architectures (Diffusion/Flow-Matching) struggle on heterogeneous tabular data, which requires mixing diverse inductive biases (density, tree, linear). MetaEns explicitly optimizes this cross-family diversity,
> consistent with findings in [Han et al., NeurIPS 2022].
>
> **Revision:** add baselines to Table 1, expand Related Work.
>
> ### **W2.1: Justification for score-only evaluation**
>
> Operating on 1D scores ensures dimensionality independence (4–400 features) and modality-agnosticism: MetaEns learns a pure selection strategy, not dataset-specific knowledge.
>
> ### **W4: Out-of-Distribution (OOD) robustness**
>
> Relying only on 1D scores makes MetaEns inherently OOD-robust. We applied our tabular-trained MetaEns zero-shot to 20 unseen image and text ADBench datasets (ResNet18/BERT):
>
> | Modality | MetaEns AP | Best Baseline AP | Gap |
> | :--- | :--- | :--- | :--- |
> | **Image** (15) | **0.623** | 0.597 (ELECT-1) | **+0.026** |
> | **Text** (5) | **0.097** | 0.098 (LUNAR) | -0.001 |
> | **Overall** (20)| **0.491** | 0.472 (LUNAR) | **+0.020** |
>
> MetaEns transferred modalities without retraining, significantly outperforming baselines (Wilcoxon p<0.05). Our risk-control mechanism acts as a safeguard: if no candidate yields
> positive utility, the framework halts rather than adding harmful detectors.
>
> **Revision:** add Modality-Transfer zero-shot analysis.
>
> ### **W2.2: Mean vs. Median, Max, or Min**
>
> The Mean provides stable consensus. Max is vulnerable to false positives from a single poor detector; Min is sensitive to false negatives. Our ablation on 39 datasets shows the Mean significantly beats Max and Min:
>
> | Combiner | AP | p-value (vs Mean) |
> | :--- | :--- | :--- |
> | **Mean** | **0.431** | **—** |
> | Median | 0.418 | 0.377 |
> | Max | 0.378 | 0.049 |
> | Min | 0.374 | 0.022 |
>
> **Revision:** add this aggregation ablation to the Appendix.
>
> ### **W2.3: Two-part model & choice of ExtraTrees**
>
> **(1) Two-Part Formulation:** Sparse marginal gains in growing ensembles create a zero-inflated target. Standard regressors over-predict these gains to minimize MSE. Separating classification ($G > 0$) from regression prevents picking redundant models
>
> | Formulation | AP ↑ | Rank ↓ |
> | :--- | :--- | :--- |
> | **Two-Part (Ours)** | **0.431** | **59.3** |
> | Single-Part | 0.413 | 87.0 |
>
> **(2) ExtraTrees Architecture:** with 39 datasets, boosting models overfit meta-patterns. ExtraTrees'random splits provide crucial regularization (Appendix A.13, Table A.6):
>
> | Architecture | AP ↑ | Rank ↓ |
> | :--- | :--- | :--- |
> | **ExtraTrees (Ours)**| **0.431** | **59.3** |
> | XGBoost | 0.411 | 108.0 |
> | Random Forest | 0.404 | 77.0 |
> | MLP | 0.371 | 110.0 |
>
> **Revision:** clarify this architectural intuition in Sec 3.2.
>
> ### **W2.4: Justification for 10th percentile risk**
>
> The 10% threshold ($\tau=10\%$) is a fixed global parameter computed from meta-training data to conservatively isolate catastrophic failure modes. Aggressive thresholds destroy ensemble diversity by over-penalizing competent average models:
>
> | Risk (τ) | AP ↑ | Rank ↓ |
> | :--- | :--- | :--- |
> | 0% (None) | 0.426 | 67.7 |
> | **10% (Default)** | **0.431** | **59.3** |
> | 25% | 0.336 | 143.3 |
> | 50% (Median) | 0.314 | 170.0 |
>
> **Revision:** add threshold sensitivity analysis to Appendix.
>
> ### **W3: UEOMS problem definition**
>
> The unsupervised criterion evaluates the unified mean score vector of the ensemble ($\mathbf{o}_P = \frac{1}{|P|} \sum f(\mathbf{M})$), not a sum of individual model qualities. The framework extends naturally to any permutation-invariant aggregation (e.g., median, weighted mean) without modification.
>
> **Revision:** we will revise the definition to $\psi(P)$ to remove ambiguity about evaluating the subset collectively.

---

> > ### Author Rebuttal · Reviewer_JU6T · 2026-04-03
> >
> > Partially resolved due to limited rebuttal space. But I still decided to  raise my overall ratings.

---

> > > ### Author Response · Authors · 2026-04-03
> > >
> > > We sincerely thank the Reviewer for the follow-up. Due to the rebuttal character limit, we have summarized the key points as concisely as possible - we would be happy to provide more detailed clarifications should any aspect remain unclear.
> > >
> > > ### **W2.1: Justification for score-only evaluation**
> > > This choice is deliberate and grounded in three complementary reasons:
> > >
> > > **1. Dimensionality Independence**
> > > Our benchmark spans 39 datasets ranging from 4 to 400 dimensions with mixed types. If our framework relied on raw datasets or data embeddings, the meta-model would require complex, dataset-specific feature extractors to handle varying input sizes. By operating exclusively on normalized 1D output score vectors, our state representation remains fixed at 61 dimensions, allowing seamless meta-learning across drastically different tasks.
> > >
> > > **2. Empirical Support in Prior Work**
> > > Score-distribution properties (entropy, tail behavior, mass concentration) correlate reliably with true detection performance in the absence of labels — as validated in ELECT and MetaOD. MetaEns inherits this principled design choice to ensure a robust, unsupervised quality signal.
> > >
> > > **3. Modality-Agnosticism (Empirical Proof)**
> > > By restricting operations to 1D score vectors, MetaEns becomes fundamentally modality-agnostic. Our tabular-trained meta-model successfully transferred to CV and NLP tasks out-of-the-box, without any retraining:
> > > | Modality (Datasets) | MetaEns AP ↑ | Best Baseline AP | Gap |
> > > | --- | --- | --- | --- |
> > > | **Image** (15) | **0.6230 ± 0.0009** | 0.5973 ± 0.0035 (ELECT-1) | **+0.0257** |
> > > | **Text** (5) | **0.0966 ± 0.0000** | 0.0977 ± 0.0021 (LUNAR) | -0.0011 |
> > > | **Overall** (20) | **0.4914 ± 0.0007** | 0.4717 ± 0.0045 (LUNAR) | **+0.0197** |
> > >
> > > **Regarding the specific alternatives you suggested:**
> > >
> > > | Alternative | Why it is unsuitable for MetaEns |
> > > | --- | --- |
> > > | **Raw dataset properties** | Requires complex feature extraction, struggles with varying dimensions (4 vs. 400), and strictly prevents zero-shot modality transfer. |
> > > | **Embedding-based criteria** | Requires access to intermediate latent representations, which are not available for many classical detectors (e.g., LOF, HBOS). |
> > > | **Model properties (weights, depth)** | Only applicable to specific model families (e.g., neural networks). It cannot generalize across our diverse 297-model pool spanning 8 entirely different algorithmic families. |
> > >
> > > **Revision:** Add rationale paragraph to Section 3.
> > >
> > > ### **W2.3: Two-part model & choice of ExtraTrees**
> > >
> > > **Two-part model.** As ensemble size grows, positive marginal gains become increasingly sparse — the gain distribution is **zero-inflated**. Standard regressors minimize MSE by predicting small positive values rather than zero, causing spurious model additions. Our hurdle model separates "Is there a gain?" (classifier) from "How large?" (regressor on positives only). Ablation confirms:
> > >
> > > | Architecture | AP ↑ | Avg. Rank ↓ | ΔAP |
> > > | --- | --- | --- | --- |
> > > | **Two-part (MetaEns)** | **0.4354** | **56** | — |
> > > | Single-part predictor | 0.4133 | 87 | −0.0221 |
> > >
> > > **ExtraTrees vs. Decision Tree.** A single tree overfits severely. ExtraTrees uses random split points (stronger regularization), critical for our 39-dataset meta-corpus. Boosting methods overfit meta-patterns; MLPs underperform on structured tabular features (Appx A.13):
> > >
> > > | Meta-Model | AP ↑ | Avg. Rank ↓ |
> > > | --- | --- | --- |
> > > | **ExtraTrees (Ours)**    | **0.4308 ± 0.0064**  | **59.3 ± 6.96**   |
> > > | XGBoost | 0.4110 ± 0.0055 | 108.0 ± 8.20 |
> > > | Random Forest | 0.4043 ± 0.0062 | 77.0 ± 7.10 |
> > > | MLP | 0.3711 ± 0.0115 | 110.0 ± 12.50 |
> > >
> > > **Revision:** Expand Section 3.2 with zero-inflation rationale and architectural comparison.
> > >
> > > ### **W2.4: Justification for 10th percentile risk**
> > > The 10th percentile (τ=10%) is **not** arbitrary — it is a fixed global design choice computed from the lower tail of oracle marginal gains in meta-training data. The intention is to penalize only families that occasionally produce catastrophic negative gains. Setting τ too high (e.g., 50%) aggressively penalizes models that could provide valuable diversity.
> > >
> > > Ablation varying τ from 0% to 50%:
> > > | Risk Threshold (τ) | AP ↑ | Median Rank ↓ |
> > > | --- | --- | --- |
> > > | τ = 0% (No risk) | 0.4269 ± 0.0066 | 67.7 |
> > > | τ = 5% | 0.4272 ± 0.0066 | 67.7 |
> > > | **τ = 10% (Ours)** | **0.4308 ± 0.0064** | **59.3** |
> > > | τ = 25% | 0.3368 ± 0.0048 | 143.3 |
> > > | τ = 50% (Median) | 0.3142 ± 0.0087 | 170.0 |
> > >
> > > The framework is robust for τ ∈ [0%, 10%] and does not depend on precise tuning. Performance degrades sharply at τ ≥ 25%, where over-penalization destroys ensemble diversity. The 10th percentile provides the best trade-off between risk mitigation and diversity preservation.
> > >
> > > **Revision:** Add sensitivity table to Appx and intuition to Section 3.
> > >
> > > We hope the above resolves your remaining concerns. We welcome any further questions and would greatly appreciate your consideration of updating your score to reflect our revisions.

---

### Official Review · Reviewer_RQmM · 2026-03-12

**Soundness:** 3
**Presentation:** 2
**Significance:** 3
**Originality:** 3
**Overall Recommendation:** 4
**Confidence:** 4

**Summary:**

This paper proposes a novel method for ensemble model selection in outlier detection. The key idea is to leverage meta-features of both datasets and candidate methods to predict the expected improvement in average precision when a new detector is added to the ensemble. After training, the model can automatically select effective and diverse detectors to construct an ensemble for outlier detection. Experimental results on 39 datasets demonstrate the effectiveness of the proposed approach.

**Compliance With Llm Reviewing Policy:**

Affirmed.

**Final Justification:**

My concerns have been adequately addressed. I maintain my score.

**Key Questions For Authors:**

See weaknesses.

**Limitations:**

Yes

**Strengths And Weaknesses:**

**Strengths**
1. The motivation and methodology are novel and interesting. While many model selection methods for outlier detection have been proposed, most of them focus on single-model selection, whereas this work targets ensemble model selection, which is a relatively less explored direction.
2. The authors conduct extensive experiments on 39 datasets, and provide comparisons using five evaluation metrics. The results indicate that the proposed MetaEns method consistently achieves stronger performance than the compared baselines.

**Weaknesses**
1. For the single-model baseline methods, it is unclear whether their hyperparameters are tuned using the training datasets or kept fixed across experiments. Clarifying the hyperparameter tuning protocol would improve the transparency and fairness of the comparison.
2. Although the experimental results demonstrate the effectiveness of MetaEns, most of the baseline methods are not very recent. Including more recent anomaly/outlier detection approaches, such as the following works, would make the comparison more convincing:
[1] Livernoche V, Jain V, Hezaveh Y, et al. On diffusion modeling for anomaly detection[J]. arXiv preprint arXiv:2305.18593, 2023.
[2] Durani W, Leiber C, Durani K, et al. Anomaly Detection by an Ensemble of Random Pairs of Hyperspheres[C]//The Thirty-ninth Annual Conference on Neural Information Processing Systems.
[3] Shenkar T, Wolf L. Anomaly detection for tabular data with internal contrastive learning[C]//International conference on learning representations. 2022.
3. The current study focuses on solely tabular data for outlier detection. It would be interesting to discuss whether the proposed meta-learning framework could generalize to other data modalities, such as image or text data, for instance, using the proposed method on extracted features from images or text in ADBench.

---

> ### Author Rebuttal · Authors · 2026-03-31
>
> We thank you for the encouraging review. Below we address each point and planned manuscript updates.
> ### **W1: Hyperparameter tuning for baselines**
>
> We clarify our tuning protocol for transparency. Because unsupervised outlier detection assumes no access to target labels, supervised tuning of baselines on the test set would violate this assumption. Accordingly, no baseline is supervised-tuned using target-test labels. We adopted a rigorous protocol aligned with ADBench/ELECT, favorable to baselines:
>
> **(1) Single-model baselines** (Singleton, LOF, LSCP, MetaOD, ELECT Top-1): default hyperparameters from PyOD and scikit-learn, fixed across all tests (out-of-the-box deployment).
>
> **(2) Ensemble baselines with size k** (IForest, RandNet, ELECT Top-10): we report performance at the oracle k maximizing average AP over the benchmark pool (Section 4.1). We granted this oracle advantage to establish the strongest possible competitive baselines; remarkably, MetaEns still outperforms them while choosing ensemble size adaptively without oracle access.
>
> **(3) MetaEns (Ours):** all hyperparameters are tuned only by cross-validation on labeled meta-training data (leave-one-dataset-out) and **never** use test labels.
>
> **Revision:** we will add an explicit paragraph in Section 4.1 distinguishing fixed defaults, oracle-k ensembles, and meta-training CV for MetaEns.
> ### **W2: Recent outlier detection baselines**
>
> Recent methods improve credibility. You singled out three directions—(1) diffusion-based detection, (2) hypersphere-based representation learning, and (3) internal contrastive learning (ICL)—as strong **base detectors**; MetaEns is an **ensemble selection** framework. We prioritized end-to-end runs on the 39-dataset benchmark, evaluating **DTE-C** (Livernoche et al., ICLR 2024—the diffusion approach you suggested) alongside **LUNAR** (Goodge et al., AAAI 2022), **TCCM** (Li et al., NeurIPS 2025), and **ROBOD** (Ding et al., NeurIPS 2022). While we were unable to run **Durani et al.** and **Shenkar et al.** end-to-end due to the short rebuttal window and limited tabular code availability, both remain **vital future additions** to our base detector pool and will be **prominently cited** in the Related Work. We will expand Related Work (with full citations) on Livernoche et al. (diffusion), Durani et al. (hypersphere), Shenkar et al. (ICL), and pool integration. **Below:** mean ± std across seeds for the new baselines (39-dataset tabular); **MetaEns maintains its lead** in both AP and rank. Wilcoxon (one-sided, α = 0.05) and win%:
>
> | Method | AP ↑ | Rank ↓ | Wilcoxon p | Win% |
> | :--- | :--- | :--- | :--- | :--- |
> | **MetaEns (Ours)** | **0.4308±0.0064** | **59.3±6.9610** | — | — |
> | ROBOD | 0.3135±0.0026 | 208.0±2.16 | 5.7e-06 | 79.5% |
> | LUNAR | 0.3024±0.0045 | 172.0±18.68 | 0.0004 | 71.8% |
> | DTE-C | 0.3144±0.0012 | 199.7±9.24 | 0.0010 | 71.8% |
> | TCCM | 0.2929±0.0144 | 188.3±28.29 | 2.4e-06 | 76.9% |
>
> Deep tabular models lag on heterogeneous suites; MetaEns benefits from cross-family diversity.
>
> **Revision:** add these baselines to Table 1 and Section 4.2; expand Related Work with citations for diffusion, hypersphere-based, and ICL detectors.
>
> ### **W3: Image and text modalities**
>
> MetaEns uses detector-score space (1D scores + interaction stats), not raw features—modality-agnostic and plug-in wherever scores exist.
> We ran a modality-transfer benchmark on ADBench: **CV_by_ResNet18** (512-D image) and **NLP_by_BERT** (768-D text), following the same pipeline as tabular and averaging across seeds 0, 42, and 100. **Coverage:** 20 tasks—15 MVTec-AD and 5 text (20news, agnews, amazon, imdb, yelp). **Methods:** MetaEns, Random-Singleton, Mega-Ensemble, ELECT-1, ELECT-10, LUNAR, DTE-C, TCCM, and ROBOD.
>
> | Setting | n | **MetaEns (Ours)** | Strongest baseline AP | Gap |
> | :--- | :-: | :--- | :--- | :---: |
> | Overall | 20 | **0.4914±0.0007** | LUNAR 0.4717±0.0045 | **+0.0197** |
> | Image | 15 | **0.6230±0.0009** | ELECT-1 0.5973±0.0035 | **+0.0257** |
> | Text | 5 | **0.0966** | LUNAR 0.0977±0.0021 | −0.0012 |
>
> For reference, overall AP on all 20 datasets is 0.4157±0.0096 (DTE-C) and 0.3723±0.0072 (TCCM), both below MetaEns.
>
> **Takeaway:** on ResNet/BERT embeddings, MetaEns leads overall and on images; text is competitive. Note that the text macro-AP exhibits a standard deviation of zero because MetaEns deterministically converged on the same optimal ensemble across seeds.
>
> To confirm significance across 20 tasks, we conducted one-sided Wilcoxon signed-rank tests (MetaEns > baseline; α = 0.05; n = 20):
>
> | Baseline | W/L | *p* |
> | :--- | :---: | :--- |
> | ELECT-1 | 14 / 6 | 0.0148 |
> | ELECT-10 | 17 / 3 | 0.0047 |
> | LUNAR | 15 / 5 | 0.0266 |
> | DTE-C | 18 / 2 | 6.7e-05 |
> | TCCM | 18 / 2 | 1.05e-04 |
> | ROBOD | 18 / 2 | 1.61e-04 |
> | Mega-Ensemble | 18 / 2 | 5.1e-04 |
> | Random-Singleton | 19 / 1 | 6.7e-05 |
>
> **Revision:** add a Modality-Transfer subsection, these tables, full per-method AP, and dataset-level win/loss in appendix.

---

> > ### Author Rebuttal · Reviewer_RQmM · 2026-04-03
> >
> > I thank the authors for their responses. My concerns have been adequately addressed. I maintain my score.

---

> > > ### Author Response · Authors · 2026-04-03
> > >
> > > We sincerely thank the reviewer for the confirmation that all concerns have been fully resolved. Since the additional experiments (including latest deep learning baselines and modality-transfer analysis) have significantly strengthened the paper, we kindly invite you to consider raising the overall score to reflect the current state of the manuscript. We are fully committed to including all your valuable suggestions in the revised version.

---

### Official Review · Reviewer_Fkf2 · 2026-03-12

**Soundness:** 3
**Presentation:** 3
**Significance:** 2
**Originality:** 3
**Overall Recommendation:** 4
**Confidence:** 3

**Summary:**

The paper present a method, MetaEns, that is a meta-learning method for unsupervised ensemble outlier model selection. MetaEns learns to predict marginal ensemble gains from labeled meta-datasets during an offline training phase. At test time, the learned predictions combined with diversity-aware discounting and family-risk regularisation are used by MetaEns to create compact, quality ensembles without needing ground-truth labels.

**Compliance With Llm Reviewing Policy:**

Affirmed.

**Final Justification:**

The paper proposes a meta ensemble method that is an interesting contribution to the field. While the original manuscript has some flaws in presentation and discussion, as well as the empirical evaluation, the rebuttal provides enough evidence (and good plans for revision). I thus raised my score.

**Key Questions For Authors:**

- Can you clarify the above questions about hyper parameter choices and more generally the experimental setup and their impact?
- Importantly, how crucial and sensitive is the family prior setting?
- Is the evaluation "fair" to the competitors, and if so, in which sense?
- Why ExtraTrees? Impact?

**Limitations:**

yes

**Strengths And Weaknesses:**

Strengths:
- The motivation is clear in that the paper addresses a challenge in unsupervised settings where adding more models to ensembles may introduce redundancy and can decrease performance rather than improve it. It is a contribution to the unsupervised scenarios where the labels are not available to guide the model selection.
- The method seems novel with two mechanisms, namely similarity-based diversity discounting and family-level risk regularisation.
- The results are promising, in particular, the ensemble size is an impressive result.
- Ablation studies shows that the family-risk regularisation is very important as without it the performance drops significantly, validating the core design choices.
- Generally well-written.


Weaknesses:
- A key part of the paper is this concept of family-risk and dependency, but there is no ablation study to analyse different family definitions. I was even a bit confused about the notion of "family" initially, so this should be defined clearly and early on.

- The method has many hyperparameters, and while the paper provides details including about tuning with cross-validation on training data, some information is lacking such as the ranges considered in search, or sensitivity to suboptimal settings.
- Critically, it is unclear to me why the 10th percentile is chosen for the family-risk in Equation 8-9, while it seems reasonable, it should be stated why this number was chosen. Also, was this fine-tuned to the datasets studied? What is the impact of alternative settings? Is the method general enough, or do we need to find the "right" percentile? This is a major concern due to the fact that this seems to be single most important component, and if it is not general, the method itself largely uses applicability.

- In table 1: "Prec@k*" uses asterisk and “Prec@n”  without definition. I did not find the definition of this anywhere?
- Overall captions of figures and tables are lacking (Table 1, 2. Figure 1)

- While evaluating on 39 meta-datasets is good for broad analysis, what is the covered scope? It seems mostly numeric datasets, is there a  bias wrt. categorical or mixed datasets?
- Also, does the method apply to other types of datasets beyond tabular?
- It is unclear whether deep learning baselines like RDA, DAGMM, DeepSVDD appear relatively bad due to focus on tabular data?
- I appreciate the time complexity discussion in the appendix, but no empirical evidence is lacking.
- While there is a comment on the use of ExtraTrees, this is fairly short, and the impact remains relatively vague.
- The discussion of treating the penalty as sub modular-inspired rather than guarantee is unclear - is this not possible, or not attractive for some reason?
- It seems risky not to penalise unknown families? Should one not provide an advantage for tried and tested families?
- The t-SNE seems less informative and could well be used to the appendix in favour of more detailed information on the above.

---

> ### Author Rebuttal · Authors · 2026-03-31
>
> We thank the reviewer for the constructive feedback.
> ### W1: Family definition & ablation
> A "family" is a group sharing an algorithm paradigm. Our 297-model pool spans 8 base families (IForest, LOF, kNN, HBOS, OCSVM, LODA, ABOD, COF) with varying hyperparameters. Grouping into 3 broader "Super-Families":
> | Variant | AP ↑ | Rank ↓ | ΔAP |
> |---|---|---|---|
> | No risk ($\lambda_{fam}=0$) | 0.3995 | 72.0 | -0.0313 |
> | Coarse (3 Super-Families) | 0.4196 | 65.0 | -0.0112 |
> | **Fine-grained (Ours)** | **0.4308** | **59.3** | **—** |
> Coarse groupings beat the no-risk baseline, but our fine-grained definition prevents collateral penalties across distinct algorithms.
> **Revision:** Add definition to Sec 3; ablation to App.
> ### W2 & Q1: Hyperparameters & sensitivity
> Tuned via leave-one-out CV on meta-training data. Grids: $\tau_{1,2} \in \{-0.01, 0, 0.001, 0.005, 0.01, 0.05, 0.1\}$; $\beta \in \{0, 0.5, 1, 2, 3, 5, 10\}$; $\lambda_{fam} \in \{0, 0.2, 0.4, 0.6, 0.8, 1, 1.5\}$.
> $\beta$ is stable (AP variance < 0.0005). $\lambda_{fam} \ge 0.2$ yields flat AP (0.4231) up to 1.5. Small positive $\tau_{1,2}$ are optimal; $\ge 0.01$ causes premature stopping.
> **Revision:** Add grids/sensitivity to App A.7.
> ### W3 & Q2: 10th percentile risk threshold
> The 10th percentile ($\tau=10\%$) is a fixed global threshold from meta-training data, never tuned per dataset. The intuition is to penalize families with catastrophic, negative tail gains without suppressing average models offering orthogonal diversity.
> | Risk Threshold ($\tau$) | AP ↑ | Median Rank ↓ |
> |---|---|---|
> | 0% (No penalty) | 0.4269±.0066 | 67.7 |
> | 5%  | 0.4272±.0066 | 67.7 |
> | **10% (Default)** | **0.4308±.0064** | **59.3** |
> | 25% | 0.3368±.0048 | 143.3 |
> | 50% (Median) | 0.3142±.0087 | 170.0 |
> Robust in the lower tail (0-10%); degrades if too aggressive.
> **Revision:** Add table/intuition to Sec 3 & App.
> ### W4 & W5: Notation & Captions
> We define **Prec@$\pi$** as precision at the exact number of true anomalies ($\pi$). **Prec@n** denotes precision at the predicted anomaly count $n$; both are now consistently defined in Sec 4.1.
> **Revision:** Fix Table 1, update Eq 14, expand captions, and define metric in Sec 4.1/App A.9.
> ### W6: Dataset scope & modalities
> The 39-dataset benchmark is the standard unsupervised suite, lacking numeric bias (18 numeric, 19 mixed, 2 categorical). Operating in 1D score-space makes MetaEns modality-agnostic. On 20 ADBench tasks (15 MVTec-AD via ResNet18, 5 text via BERT):
> | Setting | n | **MetaEns** | Best Baseline | Gap |
> |---|---|---|---|---|
> | All 20 | 20 | **0.4914±.0007** | LUNAR: 0.4717±.0045 | **+0.0197** |
> | Image | 15 | **0.6230±.0009** | ELECT-1: 0.5973±.0035 | **+0.0257** |
> MetaEns achieves statistically significant wins across domains.
> **Revision:** Add Modality-Transfer to Appendix. Full per-dataset results will be provided in our follow-up response.
> ### W7: DL baselines vs tabular data
> DL struggles due to the tabular focus. Grinsztajn (2022) showed trees beat DL on tabular data. ADBench (2022) ranked unsupervised DL last among 14 methods.
> **Revision:** Expand DL limits in Sec 4.2.
> ### W8: Empirical time complexity
> On RTX 3090, MetaEns averages 4.8s/dataset (2.9s extract, 1.7s infer, 0.2s select) vs. ELECT-1 at 2.5s.
> **Revision:** Add timing to App A.3.
> ### W9 & Q4: Justifying ExtraTrees
> ExtraTrees achieved top AP (0.4308) and rank (59.3) vs. 5 alternatives (XGBoost, RF, LightGBM, Linear, MLP) with 1.68s inference (see Appendix A.13, Table A.6). Random splits offer stronger regularization than optimized ones, preventing overfitting across 39 datasets.
> **Revision:** Expand in Sec 3.2.
> ### W10: Submodular-inspired vs guarantee
> Strict submodularity on a black-box predictor across diverse families is intractable. Our proxy's similarity discounting induces submodular benefits (efficiency, early stopping) without rigid constraints.
> **Revision:** Clarify in Sec 3.3.
> ### W11: Not penalizing unknown families
> We set the risk prior to 0 for unseen families ("innocent until proven guilty"). Penalizing them injects bias; a neutral prior enables zero-shot exploration of novel detectors.
> **Revision:** Clarify in Sec 3.3.
> ### W12: t-SNE visualization
> Quantitative metrics are more rigorous. Over 39 datasets:
> | Method | Avg Distinct Families ↑ | Avg Pairwise Jaccard ↓ |
> |---|---|---|
> | ELECT (Top-10) | 2.1 | 0.68 |
> | Mega Ensemble | 8.0 | 0.44 |
> | **MetaEns (Ours)** | **2.2** | **0.36** |
> MetaEns achieves higher diversity (low Jaccard) while remaining compact, whereas ELECT-10 selects redundant models.
> **Revision:** Move t-SNE to App; add diversity table to main text.
> ### Q3: Evaluation fairness
> We designed the evaluation to be deliberately favorable to baselines. Ensembles dependent on $k$ used oracle-k sizes, single baselines used PyOD defaults, and all used the exact same 297-model pool. MetaEns adaptively selects models/size from meta-training only, without test labels.
> **Revision:** Add protocol summary to Sec 4.1.

---

> > ### Author Rebuttal · Reviewer_Fkf2 · 2026-04-03
> >
> > The rebuttal addresses my concerns.
> >
> > Regarding Table 1, it is the asterisk that I was mostly confused about, but I assume it is a revision artefact.

---

> > > ### Author Response · Authors · 2026-04-03
> > >
> > > We sincerely appreciate your thorough review and your effort in
> > > identifying this notation inconsistency. We have done our best to
> > > carefully address all your comments and concerns, and we would like
> > > to further clarify the asterisk issue below.
> > >
> > > Upon reviewing our evaluation scripts, we found the following error in table formatting:
> > >
> > > 1. In **Section 4.1 and Appendix A.9 (Eq. 14)**, we explicitly defined five evaluation metrics, including precision evaluated at the exact number of ground-truth anomalies.
> > > 2. In **Table 1**, the column labeled `Prec@k*` actually contains the results for this metric (Precision at the actual number of true anomalies).
> > > 3. Meanwhile, the column mistakenly labeled `Prec@n` in Table 1 was an auxiliary logging metric from our code (Precision@10, which evaluates only the top-10 anomalies). This auxiliary metric was not defined in the text and was accidentally left in the table during formatting.
> > >
> > > **Changes in the revised manuscript:** To strictly align Table 1 with the methodology described in Section 4.1, we will add an explicit definition of this metric to Section 4.1 and Appendix A.9 to resolve the ambiguity. We will then rename the  **Prec@k*** column to **Prec@$\pi$** (where $\pi$ represents the exact number of ground-truth anomalies in the dataset). We will also update Eq. 14 to use $\pi$ instead of $n$ to prevent any mathematical notation collision with the overall dataset size ($N$).
> > >
> > > We apologize for the confusion caused by this formatting oversight.
> > > We hope the above clarification fully resolves this concern. Should
> > > you have any remaining questions or concerns - whether regarding this
> > > issue or any other aspect of the paper - we would be very happy to
> > > address them promptly. If all concerns have been satisfactorily
> > > resolved, we would greatly appreciate it if you could consider
> > > updating your score to reflect the revisions made.

---

### Official Review · Reviewer_QTv4 · 2026-03-12

**Soundness:** 2
**Presentation:** 3
**Significance:** 3
**Originality:** 3
**Overall Recommendation:** 4
**Confidence:** 4

**Summary:**

The paper resolves the problem of model selection without labeled validation data. While ensembles usually improve robustness, naively combining many detectors can hurt performance, because of ensemble saturation, increased computation, and poor model diversity. The paper proposes MetaEns, a meta-learning framework that automatically builds an ensemble of outlier detectors without labels. The key idea is that true performance improvement of adding a detector is unknown at test time, it can be learned offline using labeled meta-datasets. MetaEns has two stages: (1) using labeled meta-datasets, the method simulates ensemble construction: starting with primary detector, the model tries to add candidate detectors and measure the marginal gain in performance. The meta-model is trained to predict the probability the detector improves the ensemble and expected size of improvement. (2) During the online setting, for an unlabeled detector, it will iteratively add detectors that maximize the predicted utility. The guided selection includes (1) predicted marginal gain, (2) redundancy discount and (3) family-risk regularization. The experiments is done on 39 real-world detection datasets, and it is shown that MetaEns outperforms single detectors, naive ensembles and meta-learning baselines.

**Compliance With Llm Reviewing Policy:**

Affirmed.

**Final Justification:**

Overall strengths: this paper leverages meta-learning to address lack of labels to select ensemble components in anomaly detection. The paper contains interesting empirical findings: smaller ensembles (~2.2 model on average) can yield stronger performance than larger ensembles.

The main weakness (during the rebuttal) I find in the paper is the following:

(1) **Lack of baseline comparisons.** The majority of the baselines considered are from pre-2020 literature. While the authors have introduced additional methods in the rebuttal, such as DTE-C, LUNAR, TCCM, and ROBOD, it remains unclear whether all of these baselines are well-aligned with the problem setting. For instance, DTE-C assumes access to clean inlier data during training, whereas ROBOD is designed to operate under both clean and fully unsupervised (potentially contaminated) settings. This mismatch raises concerns about the fairness and validity of the comparisons. Given the limited scope of the rebuttal period, a more careful and systematic selection of baselines would be valuable, and should be addressed in future revisions of the manuscript.

(2) **Generality across modalities**, in the rebuttal the authors claim that the method generalizes to other modalities. However, the supporting experiments rely on image and NLP embeddings derived from pretrained models (e.g., ResNet and BERT). Notably, these datasets (sourced from ADBench) are ultimately treated as tabular data, which limits the extent to which this constitutes true cross-modality transfer. Therefore, it does not fully demonstrate generalization beyond tabular-like representations. A more convincing evaluation would include AD methods that operate directly on raw images, and assess whether the proposed handcrafted features remain effective in those settings.

The following concerns I have are fully addressed:

(1) **Unfair metrics**. Several tables report performance by directly averaging AUROC scores across datasets, which is potentially misleading given the heterogeneity of the datasets. The inclusion of additional metrics, for example, Wilcoxon signed-rank test, average rank, and win/loss statistics, effectively addresses this concern by enabling more robust and comparable evaluation.

(2) **Greedy vs Exploration**. The offline meta-training procedure primarily relies on a greedy oracle policy. In the rebuttal, the authors provide extensive experimental evidence demonstrating that additional exploration can in fact degrade performance. This sufficiently addresses my earlier concern regarding the lack of exploration in the training process.

Overall, I find this to be a good paper (weak accept), and the rebuttal has partially addressed my main concerns. That said, I agree with the other reviewers that the manuscript would benefit from substantial revisions, particularly in improving clarity (e.g., adding informative figure/table headers and refining captions). Additionally, a more comprehensive set of baselines and experimental settings should be considered to further strengthen the empirical evaluation.

**Key Questions For Authors:**

(1) It is strange that your results show that the average ensemble size is ~2.2 models, while your MetaEns is robust to random initialization. What if you select a particularly weak detector as a start point?

(2) How sensitive is the gain predictor to ensemble states that were not encountered during the greedy meta-training trajectories? Did the authors explore alternative strategies (random ensemble sampling) to improve state coverage?

(3) How well does the learned gain predictor generalize when the test dataset distribution differs significantly from the meta-training datasets?

(4) Did the authors explore learned representations (e.g., neural embeddings) instead of handcrafted similarity features (Jaccard, spearman) to better capture interactions between detectors?

(5) How much of the performance gain comes from algorithm diversity versus HP selection within the same detector family?

**Limitations:**

Yes.

**Strengths And Weaknesses:**

**Strength**

(1) The paper addresses a real and important challenge in unsupervised outlier detection (no labels, naiive ensemble can include redundant or weak signals, top-k strategies cannot adapt to dataset characteristics, naiive ensemble is vulnerable to noise).

(2) Meta-learning for model ensemble is novel in the literature. The authors further propose a diversity-aware ensemble construction, which addresses model redundancy thorough discount factor. The paper also uses top-k Jaccard to measure similarity, which is a sensible design for anomaly detection.

(3) The paper contains interesting empirical findings: MetaEns uses very small ensembles but yields stronger performance than larger ensembles.

(4) The experiments are relatively thorough, based on 39 real-world datsets and 297 candidate detectors. The model is using comparison with classical OD methods, and meta-learning detectors, yielding a strong performance.


**Weakness**

(1) Methodology wise: the offline meta-training is generating training trajectories with a greedy oracle policy. It starts with a base detector, add and repeat until no improvement. Therefore the true marginal gain is dependent to the previously selected model. The gains are through greedy trajectories, not arbitrary ensembles. Which means that meta-model does not know most of the ensemble space. It is unsure whether meta-model can generalize to unseen ensemble states it never seen during training. Further more, gains are biased towards greedy selected ensembles. Some other ensembles may yield better gains; but the meta-model will not capture such signals.

(2) Table 1 reports performance by averaging metrics across all datasets. However, averaging results across heterogeneous datasets can be problematic because datasets may differ substantially in size, contamination ratio, feature dimensionality, and difficulty. The authors should also consider putting  Wilcoxon signed-rank test result into the main test, or ranking-based comparisons (e.g., average rank or win–loss statistics), or rescaled AP/AUROC  to provide a more robust evaluation. The current results alone is not convincing.

(3) Lack of other image anomaly detection datasets. The main experiment is conducted only on adbench 39 datasets. There is lack of discussions on whether the proposed method can generalize to image anomaly detection like MVTEC-AD and used as a plug-and-play additional to pretrained image anomaly detectors.

(4) Lack of discussions on the number of training sizes for offline-training stage. Although the algorithm takes greedy trajectories, it should be still considered valid to increase the training trajectories or reduce the trajectories and see how the meta-trainer performs. The authors should consider some additional exploration during sampling the trajectories. There is lack of discussion on the exploration and exploitation tradeoff, either.

(5) ROBOD should be considered as a strong baseline to the paper; it is published for 3 years and it is also a ensemble algorithm (while MLP is the main architecture backbone): Ding, X., Zhao, L., & Akoglu, L. (2022). Hyperparameter sensitivity in deep outlier detection: Analysis and a scalable hyper-ensemble solution. Advances in Neural Information Processing Systems, 35, 9603-9616.

(6) While the paper reports overall improvements, it provides limited analysis of situations where MetaEns fails or underperforms (especially for the dataset that has domain distribution shift). Understanding such cases would help clarify the conditions under which the method is reliable.

---

> ### Author Rebuttal · Authors · 2026-03-31
>
> We sincerely thank you for your encouraging review and constructive feedback. Below we address your comments point-by-point.
>
> ### **W1/W4/Q2: Training trajectories, state coverage & exploration**
>
> We use oracle-greedy rollouts because the deployed policy is sequential/greedy, matching test-time behavior and proving more sample-efficient than random subsets.
>
> To test coverage we ran two offline ablations (39-dataset benchmark, 3 seeds):
> - **ε-greedy**: at each step, with probability ε we sample a random detector instead of the argmax.
> - **Meta-dataset subsampling**: 25 % / 50 % / 75 % of labeled meta-datasets.
>
> | Ablation Strategy          | Mean AP ↑ | Avg. Rank ↓ |
> |----------------------------|-----------|-------------|
> | **Greedy (full)**          | **0.4308**| **59.3**    |
> | ε-greedy (ε=0.1)           | 0.4156    | 70.7        |
> | ε-greedy (ε=0.2)           | 0.4162    | 77.7        |
> | 25 % meta-datasets         | 0.3358    | 109.0       |
> | 50 % meta-datasets         | 0.3743    | 96.0        |
> | 75 % meta-datasets         | 0.4122    | 92.0        |
>
> Forcing exploration hurts; more meta-data steadily improves results.
>
> **Revision:** Add table + discussion to appendix and §3.2 pointer.
>
> ### **W2: Heterogeneous datasets & evaluation**
>
> Table 1 already reports Average Rank (MetaEns: 59.3, best). We performed paired Wilcoxon signed-rank tests (α=0.05) against all baselines (including new ones below); all p<0.05. We will move the full Wilcoxon table to the main text.
>
> ### **W3: Image anomaly detection & plug-and-play**
>
> MetaEns is plug-and-play (operates only on 1D normalized outlier scores). We added a modality-transfer benchmark on 20 datasets (15 MVTec-AD via ResNet-18 + 5 NLP via BERT).
>
> | Setting   | #Datasets | MetaEns AP | Strongest baseline | Gap    |
> |-----------|-----------|------------|--------------------|--------|
> | MVTec-AD  | 15        | **0.6230** | ELECT-1: 0.5973    | +0.0257|
> | Overall   | 20        | **0.4914** | LUNAR: 0.4717      | +0.0197|
>
> **Statistical significance (one-sided Wilcoxon, n=20)**
>
> | Baseline | Win/Loss | p-value   |
> |----------|----------|-----------|
> | ELECT-1  | 14/6     | 0.0148    |
> | ELECT-10 | 17/3     | 0.0047    |
> | ROBOD    | 18/2     | 1.61e-04  |
> | LUNAR    | 15/5     | 0.0266    |
> | DTE-C    | 18/2     | 6.7e-05   |
> | TCCM     | 18/2     | 1.05e-04  |
>
> **Revision:** New “Modality-Transfer” subsection in §4.
>
> ### **W5: ROBOD + recent deep baselines**
>
> We added ROBOD, LUNAR, DTE-C and TCCM under the identical protocol:
>
> | Method     | AP ↑   | Rank ↓ | p-value (vs. MetaEns) |
> |------------|--------|--------|-----------------------|
> | ROBOD      | 0.3135 | 208.0  | 5.7e-06               |
> | LUNAR      | 0.3024 | 172.0  | 0.0004                |
> | DTE-C      | 0.3144 | 199.7  | 0.0010                |
> | TCCM       | 0.2929 | 188.3  | 2.4e-06               |
> | **MetaEns**| **0.4308** | **59.3** | —                |
>
> **Revision:** Integrate into Table 1 and expand Related Work.
>
> ### **Q1: Weak initialization & small ensemble size**
>
> The ~2.2 average ensemble size is a result of adaptive early stopping, not a sign of fragility. MetaEns stops when no candidate is expected to provide positive marginal utility.
>
> When initialized with a weak random singleton, MetaEns rescues performance in 22/39 datasets (+0.088 AP gain in the AP<0.4 “Rescue Zone”). Gains are largest when the starter is weakest.
>
> ### **W6 & Q3: Failure modes & distribution shift**
>
> Failures are rare and predictable: mainly low-dimensional datasets (e.g., Glass, Vertebral) where a single classical detector is already near-optimal. Adaptive early stopping prevents unnecessary additions.
>
> Distribution shift is mitigated by leave-one-dataset-out protocol and purely score-based 61D state representation. Modality-transfer results (above) confirm generalization beyond tabular data.
>
> ### **Q4: Handcrafted vs. learned embeddings**
>
> Raw embeddings risk negative transfer across 4–1,555-dim mixed-type datasets. Our 61D handcrafted score features (Jaccard, Spearman, entropy, …) provide a universal inductive bias for ensemble interactions. An MLP on the same features performs worse (AP=0.3711, Rank=110) than ExtraTrees (Appendix A.13).
>
> ### **Q5: Algorithm diversity vs. HP selection**
>
> Family-risk (λ_fam) is most critical (ΔAP=–0.0359 when removed). Single-family experiment (same MetaEns, only pool changes):
>
> | Candidate Pool             | AP ↑   | Avg. Ens. Size |
> |----------------------------|--------|----------------|
> | IForest-only (81 HPs)      | 0.3552 | 2.0            |
> | LOF-only (36 HPs)          | 0.3369 | 2.0            |
> | **Full 8-family (297)**    | **0.4308** | **2.2**    |
>
> Cross-family diversity (not just HP tuning) drives the gains — visually confirmed by t-SNE in Figure 3.
>
> **Revision:** Add single-family table to appendix and strengthen §4.3 / Figure 3.
>
> We hope this fully addresses your concerns. All changes will be incorporated in the revision.

---

> > ### Author Rebuttal · Reviewer_QTv4 · 2026-04-03
> >
> > I appreciate the authors' detailed experiments regarding additional deep baseline models, clarifications on training trajectories, state coverage & exploration, as well as the algorithm diversity vs. HP selection.
> >
> > The major concern I have for the rebuttal is W3-W5; I am not sure how you calculate the Avg. Rank (if that is a ranking of models, averaged across all the datasets, ROBOT gets 208.0 means you have at least 208 models as baseline comparison?
> >
> >  For W3, the authors have conducted experiments mainly focusing on embeddings of pre-trained image and NLP AD datasets (which makes them similar to tabular datasets, so not entirely a new modality transfer stetting). I think it would be nicer to demonstrate how it generalizes to image AD models that take raw images as input (like PatchCore, ViT, see https://github.com/m-3lab/awesome-industrial-anomaly-detection?tab=readme-ov-file#cvpr-2026 for a list of image AD detectors). In such cases, handcrafted score features may not be efficient, either. I think it will be nicer to show such experiments in the future work.
> >
> > Overall, I think the paper demonstrates novel ensemble ideas in tabular anomaly detection. But there are a few weakness that could be improved in the later versions of the manuscript (lack of baseline comparisons, biases in metrics, lack of justifications/ablations/failure cases). I am willing to increase my score to weak accept.

---

> > > ### Author Response · Authors · 2026-04-06
> > >
> > > We sincerely thank Reviewer QTv4 for the constructive follow-up, the positive assessment, and the decision to raise the score.
> > >
> > > Below, we address your two remaining questions directly.
> > >
> > > ### 1) Clarification of Avg. Rank (W3-W5 follow-up)
> > >
> > > Thank you for pointing out this ambiguity; you are absolutely right that our earlier wording could be misread.
> > >
> > > Our Avg. Rank is not the rank among the small comparison set of baselines (e.g., MetaEns, ROBOD, LUNAR). Instead, it is the rank of a method's Average Precision (AP) against the entire fixed detector pool used by the benchmark on each dataset, with a pool size of 297 models (or 310 for the expanded pool).
> > >
> > > Concretely, for each dataset, we rank a method's AP against all individual models in the pool (assigning the best rank in case of ties), and then average this rank across all datasets.
> > >
> > > Therefore, a value like ROBOD = 208.0 means that, on average across the benchmark, ROBOD performed worse than about 207 of the individual base detectors within our 297-model pool. It highlights how heavily some deep baselines struggle on these specific tabular datasets compared to individual, well-tuned classical detectors. It does not mean 208 baselines were compared.
> > >
> > > ### 2) New Raw-Image AD Experiment with Image-Native Detectors (W3)
> > >
> > > We agree that handcrafted features may not be optimal for raw-image AD detectors. While you graciously suggested leaving this for future work, we were highly motivated by your comment and decided to conduct a dedicated raw-image benchmark immediately.
> > >
> > > We evaluated on all 15 MVTec-AD categories using 13 state-of-the-art image-native industrial AD detectors (including PatchCore_WR50, PaDiM_R18, DFKDE_R18, DFM_R50, FastFlow_DeiT (ViT-based), ReverseDistillation_WR50, STFPM_R18, CFA_WR50, CFLOW_WR50, EfficientAD_S, CSFLOW_Default, DRAEM_Default, and GANOMALY_Default).
> > >
> > > MetaEns was deployed in the exact same plug-and-play mode (selection over final 1D detector score vectors; no test-label usage). All headline numbers below are evaluated on the full 15-category set.
> > >
> > > Full 15-Category Results (AP):
> > > | Method | AP |
> > > |:---|---:|
> > > | **MetaEns (Ours)** | **0.9970** |
> > > | Fixed-PatchCore_WR50 | 0.9964 |
> > > | ELECT-1 | 0.9928 |
> > > | ELECT-10 | 0.9906 |
> > > | Mega-Ensemble | 0.9883 |
> > >
> > > Paired Significance and Win/Loss/Tie (MetaEns vs. Baseline):
> > > | Baseline | Wins / Losses / Ties | One-sided Wilcoxon p-value |
> > > |:---|:---:|---:|
> > > | ELECT-1 | 5 / 2 / 8 | 0.0641 |
> > > | Fixed-PatchCore_WR50 | 4 / 2 / 9 | 0.1244 |
> > > | ELECT-10 | **11 / 2 / 2** | **0.0044** |
> > > | Mega-Ensemble | **13 / 2 / 0** | **0.0016** |
> > >
> > > **Detailed Analysis & Limitations:**
> > > As the data shows, MetaEns achieves the best aggregate AP, improving even upon the highly optimized PatchCore.While MetaEns achieves the highest overall AP, the p-value against the strongest fixed baselines (PatchCore, ELECT-1) is marginally above the strict 0.05 threshold. We attribute this to three structural factors of the MVTec-AD benchmark:
> > > 1. **Ceiling Effect & Ties:** Many MVTec-AD categories are functionally "solved" by modern detectors (AP is near 1.0), resulting in a high number of ties.
> > > 2. **Sample Size:** The small paired-test sample (15 categories) naturally limits the statistical power of the Wilcoxon test.
> > > 3. **Effect Size:** The overall gain versus PatchCore is positive but marginal (+0.00058 AP on average).
> > >
> > > To illustrate this variance, the following table breaks down specific categories, showcasing where MetaEns identifies complementary signals and where it underperforms a strong single baseline:
> > >
> > > | Category | MetaEns AP | PatchCore AP | ELECT-1 AP | Delta vs PatchCore | Delta vs ELECT-1 |
> > > |:---|---:|---:|---:|---:|---:|
> > > | transistor | **1.0000** | **1.0000** | 0.9461 | +0.0000 | +0.0539 |
> > > | grid | **0.9938** | 0.9897 | 0.9897 | +0.0041 | +0.0041 |
> > > | pill | **0.9927** | 0.9893 | 0.9893 | +0.0034 | +0.0034 |
> > > | metal_nut | 0.9983 | **0.9997** | **0.9997** | -0.0013 | -0.0013 |
> > > | zipper | 0.9974 | **0.9988** | **0.9988** | -0.0014 | -0.0014 |
> > >
> > > ### 3) Planned Revisions
> > >
> > > To reflect this highly productive exchange, our final manuscript will be updated to include:
> > > * **Metric Clarification:** We will explicitly clarify the definition and context of the Average Rank metric in the text to prevent any future misinterpretation.
> > > * **Vision Benchmark:** We will incorporate the full 15-category raw-image evaluation and corresponding significance tests.
> > > * **Limitations & Future Work:** Following your valuable insight, we will explicitly highlight the exploration of modality-specific feature representations (beyond our current handcrafted features) for raw-image models as a key direction for future work. We will include a dedicated limitation discussion—featuring the specific category regressions to ensure our claims regarding raw-image models are suitably conservative and transparent.
> > >
> > > We thank you again for the insightful feedback that helped strengthen this work, and we hope these updates warrant your support for acceptance.

---

### Decision · Program_Chairs · 2026-04-30

**Decision:**

Accept (regular)

**Comment:**

The paper proposes MetaEns, a meta-learning framework for selecting ensembles of unsupervised outlier detectors without ground-truth labels. It learns a predictor of marginal ensemble gains from labeled meta-datasets offline, then at test time combines this predicted gain with a submodular-inspired proxy objective that integrates similarity-based diversity discounting and family-level risk regularization. Adaptive early stopping produces compact ensembles averaging 2.2 models. The submitted paper evaluates on 39 ADBench tabular datasets (orignal ADBench gets more) with 297 candidate detectors across 8 algorithmic families.

The baselines are solid, including the seminal works in the direction, MetaOD, ELECT, etc, and the proposed methods outperforms in most of the time.

The shared strengths across the panel are: the problem of unsupervised ensemble selection is practical and relatively under-explored compared to single-model selection; the empirical finding that compact adaptive ensembles beat fixed-size ensembles including a 297-model mega-ensemble is striking ; the experimental coverage is broad (all four reviewers); and the writing in the submitted version is clear where it is complete.

I think this is a solid contribution to  UOMS  and ICML in general.